# SELF-SUPERVISED LEARNING FROM STRUCTURAL INVARIANCE

**Yipeng Zhang**[1,2†]  **Hafez Ghaemi**[1,2,3]  **Jungyoon Lee**[1,2]  **Shahab Bakhtiari**[1,2]
**Eilif B. Muller**[1,2,3,5]  **Laurent Charlin**[1,2,4,5]
[1]Mila - Québec AI Institute   [2]Université de Montréal   [3]CHU Sainte-Justine
[4]HEC Montréal   [5]CIFAR AI Chair

## ABSTRACT

Joint-embedding *self-supervised learning* (SSL), the key paradigm for unsupervised representation learning from visual data, learns from invariances between semantically-related data pairs. We study the one-to-many mapping problem in SSL, where each datum may be mapped to multiple valid targets. This arises when data pairs come from naturally occurring generative processes, e.g., successive video frames. We show that existing methods struggle to flexibly capture this conditional uncertainty. As a remedy, we introduce a latent variable to account for this uncertainty and derive a variational lower bound on the mutual information between paired embeddings. Our derivation yields a simple regularization term for standard SSL objectives. The resulting method, which we call AdaSSL, applies to both contrastive and distillation-based SSL objectives, and we empirically show its versatility in causal representation learning, fine-grained image understanding, and world modeling on videos.[1]

## 1 INTRODUCTION

Over the last decade, joint-embedding *self-supervised learning* (SSL) has become the dominant approach in representation learning from unlabeled visual data (Chen et al., 2020a; Zbontar et al., 2021; Grill et al., 2020; Radford et al., 2021; Assran et al., 2023). The intuition behind SSL is to obtain semantically-related data pairs, often called *positive pairs*, and encourage their representations to be similar, with proper regularization to prevent the encoder collapsing to a constant function (Wang & Isola, 2020; Garrido et al., 2023a; Zhuo et al., 2023). The hope is that invariance to the differences between positive pairs leads to generalizable representations.

However, positive pairs are typically built with handcrafted augmentations (e.g., cropping, color jittering), which perturb pixels while preserving semantics. Such augmentations cannot precisely mimic changes in natural factors of variation that drive real-world distribution shifts (Ibrahim et al., 2023). For example, brightness jitter across pixels does not reproduce how lighting varies across objects or environments (e.g., from indoor to outdoor). Consequently, augmentations may fail to induce the right invariances (Ibrahim et al., 2023; 2022; Bouchacourt et al., 2021), discard fine-grained information (Chen et al., 2020a; Zhang et al., 2024), incur additional computation burden (Bordes et al., 2023), and require modality-specific heuristics (Balestriero et al., 2023), ultimately harming downstream performance.

One alternative is to exploit naturally-paired data—e.g., nearby video frames (Klindt et al., 2021; Bardes et al., 2024; Sermanet et al., 2018), image–caption pairs (Radford et al., 2021), class labels (Khosla et al., 2020), or embeddings from other models (Sobal et al., 2025a; Feizi et al., 2024)—which (better) reflect real-world variations. From the lens of *causal representation learning* (CRL) (Yao et al., 2025; Reizinger et al., 2025), positive pairs $(\mathbf{x}, \mathbf{x}^+)$ are deterministically mapped from latent factors sampled according to $(\mathbf{z}, \mathbf{z}^+) \sim p(\mathbf{z})p(\mathbf{z}^+ \mid \mathbf{z})$, and we aim to recover $\mathbf{z}$ from $\mathbf{x}$. Unlike augmentations that perturb the observations directly, natural positive pairs differ according to structured changes in latent factors of the *data generating process* (DGP). Modeling these latent

---

[†]Correspondence to `yipeng.zhang@mila.quebec`.
[1]Code is available at `https://github.com/SkrighYZ/AdaSSL`.

changes can potentially improve generalization (Ibrahim et al., 2022; Dittadi et al., 2021; Kaur et al., 2023) and visual understanding (Awal et al., 2024; Garrido et al., 2025; Lippe et al., 2023).

Despite the aforementioned benefits, modeling natural pairs for SSL remains challenging. When learning with augmented pairs, we typically assume the semantic latent factors are invariant, so the latent conditional distribution $p(\mathbf{z}^+ \mid \mathbf{z})$ is highly concentrated near $\mathbf{z}$. However, natural pairs induce complex $p(\mathbf{z}^+ \mid \mathbf{z})$. For example, in world modeling (Ha & Schmidhuber, 2018b;a; Hafner et al., 2025; Assran et al., 2025), the present state $\mathbf{z}$ may lead to multiple plausible futures $\{\mathbf{z}^+\}$ (e.g., a car may turn left or right), making the conditional distribution inherently multimodal. For image–caption pairs, caption details vary with image complexity, producing *heteroscedastic* noise. SSL methods that fail to capture this uncertainty often discard information not shared between the pair, leading to degraded performance (Chen et al., 2020a; Radford et al., 2021; Jing et al., 2022; Yuksekgonul et al., 2023; Trusca et al., 2024; Zhang et al., 2024). We argue that leveraging the structure of $p(\mathbf{z}^+ \mid \mathbf{z})$ enables SSL to learn more diverse and generalizable features—a principle we call **SSL from structural invariance**.

Recent works enable contrastive SSL to learn $p(\mathbf{z}^+ \mid \mathbf{z})$ that has constant, anisotropic noise (Kügelgen et al., 2021; Zimmermann et al., 2021; Rusak et al., 2025). In this work, we contribute a solution to model more complex conditional distributions with unknown shapes. We are inspired by *joint-embedding predictive architectures* (JEPAs) (LeCun, 2022; Garrido et al., 2024; Assran et al., 2025), which use a latent variable to capture the prediction uncertainty. However, in contrast to prior work (Devillers & Lefort, 2023; Garrido et al., 2024; Ghaemi et al., 2024; Dangovski et al., 2022), we do not assume access to this variable and infer it purely from (the structure hidden in) positive pairs. For contrastive learning, this latent variable allows us to derive a new tractable bound on the *mutual information* (MI) between paired embeddings—a core motivation for SSL (Linsker, 1988; Tschannen et al., 2020; Oord et al., 2018)—and we empirically show its compatibility with distillation-based methods. We name our method **Adaptive SSL (AdaSSL)** to highlight that it can adapt to different conditional distributions.

We evaluate AdaSSL in controlled settings with numerical data, natural images, and videos. On numerical data, we show that existing SSL methods lack the ability to model non-trivial conditionals, and AdaSSL achieves better performance both in- and out-of-distribution (OOD). On images, AdaSSL consistently recovers fine-grained features better than baselines and learns more disentangled representations. On videos, AdaSSL captures stochastic object accelerations that baselines discard without sacrificing class accuracy.

In summary, our main contributions show that:

- Naturally paired data that differ in their data-generating factors can help SSL methods to recover these factors better and lead to improved generalization performance.
- Existing SSL methods cannot capture the non-trivial conditional distributions induced by natural pairs, and we propose two variants of AdaSSL to address this limitation.
- AdaSSL consistently outperforms baselines on a variety of benchmarks, including weakly-supervised CRL, fine-grained image understanding, and world modeling.

## 2 BACKGROUND AND MOTIVATION

In this section, we lay out our problem (§2.1) and discuss the limitations of existing SSL methods in addressing it (§2.2, §2.3). We then present a theoretical result showing the importance of modeling heteroscedastic noise (§2.4), which motivates our method.

### 2.1 PROBLEM FORMULATION

In CRL, representation learning is viewed as learning to *invert* the DGP such that the latent factors responsible for generating the data are recovered (Zimmermann et al., 2021; Kügelgen et al., 2021; Rusak et al., 2025; Reizinger et al., 2025). We assume a data pair $(\mathbf{x}, \mathbf{x}^+)$ conforms to the following generative process:

$$\mathbf{z} \sim p(\mathbf{z}), \quad \mathbf{z}^+ \mid \mathbf{z} \sim p(\mathbf{z}^+ \mid \mathbf{z}), \quad \mathbf{x} = g(\mathbf{z}), \quad \mathbf{x}^+ = g(\mathbf{z}^+), \tag{1}$$

where $g : \mathcal{Z} \to \mathcal{X}$ is an unknown mixing function that produces the observations $\mathbf{x}, \mathbf{x}^+ \in \mathcal{X}$ based on the latent factors $\mathbf{z}, \mathbf{z}^+ \in \mathcal{Z}$.

Under Eq. 1, the variability in $\mathbf{x}$ is entirely captured by the latent factors $\mathbf{z}$ since $g$ is deterministic. A representation that encodes the full latent variability is therefore important for supporting a broad range of downstream tasks, each relying on some (a priori unknown) semantic structure in $\mathbf{z}$ (Bengio et al., 2013a). When SSL models fail to accurately model $p(\mathbf{z}^+ \mid \mathbf{z})$, they may discard features that contribute to the unmodeled variance, because doing so removes them from the target and thereby reduces the prediction error.

Formally, our goal is to learn a function $f : \mathcal{X} \to \mathbb{R}^{d_f}$ that encodes the data $\mathbf{x} \in \mathcal{X}$ into an embedding space $\mathbb{R}^{d_f}$ such that we can predict, or *identify*, a subset[2] of the latent factors that are useful for downstream tasks from $f(\mathbf{x})$ with a simple function, e.g., an affine transformation. We denote this subset of latent factors as "content factors" $\mathbf{c} := \mathbf{z}_{\mathbb{I}}$ for $\mathbb{I} \subseteq [d_z]$, and the other (less relevant) factors as "style" factors $\mathbf{s} := \mathbf{z}_{[d_z] \setminus \mathbb{I}}$ following Kügelgen et al. (2021).

## 2.2 PRELIMINARIES: CONTRASTIVE SSL

Contrastive SSL methods assume the content factors $\mathbf{c}$ to be roughly unperturbed under the conditional law $p_{\mathbf{z}^+ \mid \mathbf{z}}$, and use an objective that encourage $f(\mathbf{x})$ and $f(\mathbf{x}^+)$ to be similar. To prevent representation collapse where $f$ becomes a constant function, contrastive objectives use another term to encourage the representations to have high entropy (Chen et al., 2020a; Zbontar et al., 2021; Bardes et al., 2022; Wang & Isola, 2020). In this work, we focus on sample-contrastive methods based on InfoNCE (Oord et al., 2018; Chen et al., 2020a), and observe the duality between dimension- and sample-contrastive methods (Garrido et al., 2023a; Balestriero & LeCun, 2022). The InfoNCE loss has the form:

$$\mathcal{L}_{\text{InfoNCE}} = \mathbb{E}_{\{(\mathbf{x}^{(i)}, \mathbf{x}^{+(i)})\}_{i=1}^K \overset{\text{iid}}{\sim} p(\mathbf{x}, \mathbf{x}^+)} \left[ \frac{1}{K} \sum_{i=1}^K - \log \frac{e^{s(\mathbf{x}^{(i)}, \mathbf{x}^{+(i)})/\tau}}{\frac{1}{K} \sum_{j=1}^K e^{s(\mathbf{x}^{(i)}, \mathbf{x}^{+(j)})/\tau}} \right], \quad (2)$$

where $\tau$ is a temperature parameter and $s(\cdot, \cdot)$ is a similarity function over pairs. Intuitively, InfoNCE encourages the similarity function to assign a high score for positive pairs and a low score for pairs that does not come from the true joint. The similarity function often adopts a simple form on the normalized embeddings, i.e., $s(\mathbf{x}, \mathbf{y}) = \psi(\mathbf{x})^\top \psi(\mathbf{y})$ where $\psi(\cdot) = \frac{f(\cdot)}{\|f(\cdot)\|_2}$. The simplicity of the similarity function allows features to be easily extracted from the embedding space because they are used to discriminate between data points linearly during training (Tschannen et al., 2020).

It has been shown that when the marginal $p(\mathbf{z}^+)$ is uniform, the similarity function implicitly models the log conditional: $s^\star(\mathbf{x}, \mathbf{x}^+) \propto \log p(\mathbf{z}^+ \mid \mathbf{z})$ (Zimmermann et al., 2021). With a dot-product similarity, the hypothesis class of $p_{\mathbf{z}^+ \mid \mathbf{z}}$ reduces to von Mises-Fisher (vMF) distributions, where $\tau$ controls the concentration strength. Since vMF distribution does not account for anisotropic noise, Rusak et al. (2025) introduces a diagonal matrix $\mathbf{\Lambda}$ that weighs the concentration along each dimension: $s(\mathbf{x}, \mathbf{y}) = -(\psi(\mathbf{x}) - \psi(\mathbf{y}))^\top \mathbf{\Lambda} (\psi(\mathbf{x}) - \psi(\mathbf{y}))$. **Nevertheless, it remains unclear how to flexibly model an arbitrary conditional distribution $p(\mathbf{z}^+ \mid \mathbf{z})$ while keeping the similarity function simple enough to allow efficient feature extraction.**

## 2.3 PRELIMINARIES: DISTILLATION-BASED SSL

Distillation-based SSL methods do not use explicit regularization to prevent representation collapse. Typically, they use asymmetric encoders: an online branch predicts target representations, with a stop-gradient on the target (Grill et al., 2020; Chen & He, 2021). We illustrate our findings with BYOL (Grill et al., 2020), the backbone of many recent successful distillation-based methods (Guo et al., 2022; Assran et al., 2025):

$$\mathcal{L}_{\text{BYOL}} = \left\| \eta(\psi(\mathbf{x})) - \psi_{\text{EMA}}(\mathbf{x}^+) \right\|_2^2, \quad (3)$$

where $\psi_{\text{EMA}}$ is the exponential moving average of the parameters defining $\psi$ over time and $\eta(\cdot)$ is an MLP predictor. **Intuitively, the predictor accounts for cases where $\mathbb{E}[\mathbf{z}^+ \mid \mathbf{z}] \neq \mathbf{z}$; but it remains unclear how it can capture complex noise structures in $p(\mathbf{z}^+ \mid \mathbf{z})$—which may be heteroscedastic or even multimodal—without conditioning on additional information.**

---

[2]Although full latent recovery is often the goal in theory, invariance to certain style factors can help generalization (Deng et al., 2022) and prevent shortcut solutions in SSL (Chen et al., 2020a) under finite data.

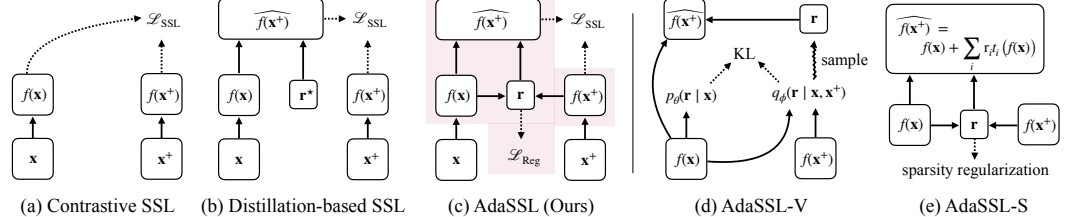

Figure 1: Visual comparison of models. Boxes denote vectors and arrows denote functions. We use $f$ to denote both encoders although they may use different parameters in practice. (a) Contrastive SSL typically uses a symmetric architecture. (b) Distillation-based SSL uses a predictor to predict the embeddings of one branch from the other, optionally with the help of some supervision $\mathbf{r}^\star$ related to the difference between the inputs. (c) Our method, AdaSSL, extends SSL by modeling the latent variable $\mathbf{r}$. (d) AdaSSL-V learns a variational distribution, $q_\phi(\mathbf{r} \mid \mathbf{x}, \mathbf{x}^+)$, and uses an MLP as predictor. (e) AdaSSL-S regularizes the sparsity of $\mathbf{r}$ and uses a modular predictor.

## 2.4 UNAVOIDABLE HETEROSCEDASTICITY

Before presenting our solution to these questions, we first provide a theoretical result that underscores the importance of modeling heteroscedasticity between paired embeddings.

**Proposition 2.1.** *Let $\mathbb{S}^{d_f} \subset \mathbb{R}^{d_f+1}$ denote the $d_f$-dimensional unit sphere. Let $g : \mathbb{R}^{d_z} \to \mathbb{R}^{d_x}$ be $C^1$ diffeomorphic to its image, and let $f : \mathbb{R}^{d_x} \to \mathbb{S}^{d_f}$ be $C^1$ almost everywhere. Define $h := f \circ g : \mathbb{R}^{d_z} \to \mathbb{S}^{d_f}$. Assume further that the random vectors $\mathbf{z}, \mathbf{z}^+ \in \mathbb{R}^{d_z}$ are sampled as $\mathbf{z} \sim p_{\mathbf{z}}, \mathbf{z}^+ = \mathbf{z} + \varepsilon, \varepsilon \sim p_\varepsilon$, where $p_{\mathbf{z}}$ is not a point mass and $\varepsilon$ is independent of $\mathbf{z}$, $\mathbb{E}[\varepsilon] = 0$, and $\mathrm{Cov}(\varepsilon) \succ 0$. Suppose that for $p_{\mathbf{z}}$-almost every $\mathbf{z}$ we have $\mathrm{rank}\, Dh(\mathbf{z}) = d_z$. Write $H = h(\mathbf{z})$ and $H^+ = h(\mathbf{z}^+)$. Then the conditional law $p_{H^+|H}(h(\mathbf{z}^+) \mid h(\mathbf{z}))$ is necessarily heteroscedastic: its conditional variance depends on $h(\mathbf{z})$ for $p_{\mathbf{z}}$-almost every $\mathbf{z}$.*

Proposition 2.1 shows that heteroscedasticity between paired embeddings emerges from the geometric mismatch between the embedding space and the ground-truth latent space, regardless of the encoding function or embedding dimensionality (proof in §C.1). Intuitively, mapping the flat latent space $\mathbb{R}^{d_z}$ onto a curved manifold such as $\mathbb{S}^{d_f}$ distorts local neighborhoods differently depending on the location of $h(\mathbf{z})$, causing input-dependent variance in $p(h(\mathbf{z}^+) \mid h(\mathbf{z}))$. Here, we explicitly show the case of projecting from unbounded latent space $\mathbb{R}^{d_z}$ to normalized embedding space $\mathbb{S}^{d_f}$ and discuss the reverse scenario in §C.2.

In SSL, standard similarity functions and predictors implicitly assume that positive pairs exhibit the same noise scale (§2.2, §2.3), but Proposition 2.1 shows that this cannot hold when the (unknown) geometry of $\mathcal{Z}$ and the embedding space differ. Consequently, common similarity functions such as the dot product fail to capture this conditional variance, since they aggregate the variability uniformly across all embedding directions and data pairs. We show this empirically in §4.2.

## 3 METHOD

We now present our method, which addresses the aforementioned challenges by modeling uncertainty with a *latent-variable model*. In §3.1, we introduce our overall objective. We then discuss two variants of AdaSSL in §3.2 and §3.3, which optimize this objective in distinct ways.

### 3.1 MODELING UNCERTAINTY WITH A LATENT VARIABLE

Learning a representation that maximally preserves the mutual information (MI) between paired embeddings is useful for representation learning and motivates SSL (Linsker, 1988; Tschannen et al., 2020; Oord et al., 2018). Contrastive SSL optimizes a lower bound on the mutual information $I(f(\mathbf{x}); f(\mathbf{x}^+))$ (Oord et al., 2018) but, as discussed in Sec. 2.2, relies on strong assumptions on the latent conditional $p(\mathbf{z}^+ \mid \mathbf{z})$.

To relax these assumptions, we introduce an auxiliary latent variable $\mathbf{r}$ to parameterize the conditional uncertainty in $p(\mathbf{z}^+ \mid \mathbf{z})$. This induces a latent-variable model $p(\mathbf{r}, \mathbf{z}^+ \mid \mathbf{z}) = p(\mathbf{r} \mid \mathbf{z})p(\mathbf{z}^+ \mid$

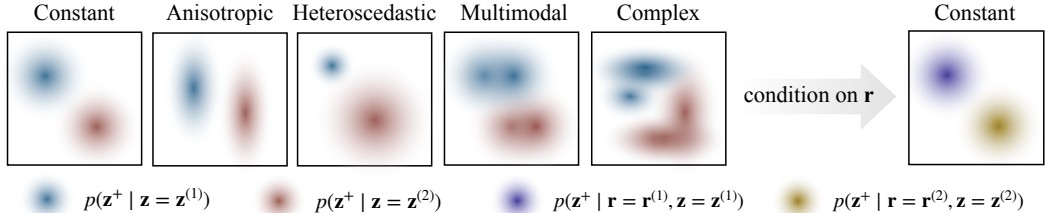

Figure 2: Illustration of different types of noise structure in $p(\mathbf{z}^+ \mid \mathbf{z})$. Here, "constant" refers to isotropic, homoscedastic noise. Conditioning on a latent variable $\mathbf{r}$ can transform the noise into a simpler form. For example, a car may turn left or right, producing a bimodal conditional distribution; conditioning on the driver's intention removes the irrelevant mode.

$\mathbf{r}, \mathbf{z}$), where $p(\mathbf{r} \mid \mathbf{z})$ acts as a conditional prior. In Fig. 1, we visually compare our method to existing approaches. Intuitively, $\mathbf{r}$ should contain information about $\mathbf{z}^+$ that cannot be solely predicted from $\mathbf{z}$. For example, $\mathbf{r}$ may represent camera movement, agent actions, or temporal gaps—factors that modify the underlying latents before they are mapped to observations through $g$. Consequently, modeling $p(\mathbf{z}^+ \mid \mathbf{z}, \mathbf{r})$ may require a simpler model (e.g., $s(\cdot, \cdot)$ in Eq. 2 or $\eta(\cdot)$ in Eq. 3) compared to modeling the full $p(\mathbf{z}^+ \mid \mathbf{z})$, as illustrated in Fig. 2.

With this in mind, we can rearrange $I(f(\mathbf{x}); f(\mathbf{x}^+))$ into a form that is easier to model. Specifically, by the chain rule of MI,

$$I\big(f(\mathbf{x}); f(\mathbf{x}^+)\big) = I\big(f(\mathbf{x}), \mathbf{r}; f(\mathbf{x}^+)\big) - I\big(\mathbf{r}; f(\mathbf{x}^+) \mid f(\mathbf{x})\big). \tag{4}$$

Intuitively, optimizing the first term on the RHS involves modeling $p(\mathbf{z}^+ \mid \mathbf{z}, \mathbf{r})$ instead of $p(\mathbf{z}^+ \mid \mathbf{z})$. This can be achieved through an SSL objective that encourages the model to use information in $\mathbf{r}$ to reduce the uncertainty in predicting $f(\mathbf{x}^+)$ from $f(\mathbf{x})$. However, without constraints, $\mathbf{r}$ could encode $f(\mathbf{x}^+)$ directly, increasing the second term and creating a shortcut. This motivates the general form of our objective:

$$\mathcal{L}_{\text{AdaSSL}} = \mathcal{L}_{\text{SSL}}((\mathbf{x}, \mathbf{r}), \mathbf{x}^+) + \beta \mathcal{L}_{\text{Reg}}(\mathbf{r}), \tag{5}$$

where the SSL term is any standard SSL loss (e.g., $\mathcal{L}_{\text{InfoNCE}}$) and the regularizer $\mathcal{L}_{\text{Reg}}(\mathbf{r})$ limits this degenerate behavior by discouraging overly informative $\mathbf{r}$. The hyperparameter $\beta$ controls the strength of regularization per standard practice (Higgins et al., 2017; Locatello et al., 2020). This objective matches the conceptual framework depicted in Fig. 13 of LeCun (2022).

## 3.2 AdaSSL-V and a lower bound on $I(f(\mathbf{x}), f(\mathbf{x}^+))$

We first estimate the posterior of $\mathbf{r}$ with a variational distribution $q_\phi(\mathbf{r} \mid \mathbf{x}, \mathbf{x}^+)$ (Sohn et al., 2015). The joint then becomes $\tilde{p}(\mathbf{x}, \mathbf{x}^+, \mathbf{r}) := p(\mathbf{x}, \mathbf{x}^+)q(\mathbf{r} \mid \mathbf{x}, \mathbf{x}^+)$. The informational-theoretical properties of contrastive learning allow us to optimize a lower bound on $I(\mathbf{x}, \mathbf{x}^+)$[3]. Specifically, the first term in Eq. 4 is bounded by InfoNCE (Oord et al., 2018) by treating $(\mathbf{x}, \mathbf{r})$ as a single variable:

$$I_{\tilde{p}}(\mathbf{x}, \mathbf{r}; \mathbf{x}^+) \geq -\mathcal{L}_{\text{InfoNCE}} = \mathbb{E}_{\{(\mathbf{x}^{(i)}, \mathbf{x}^{+(i)}, \mathbf{r}^{(i)})\}_{i=1}^K \overset{\text{iid}}{\sim} \tilde{p}} \left[ \frac{1}{K} \sum_{i=1}^K \log \frac{e^{s(\mathbf{x}^{(i)}, \mathbf{x}^{+(i)}, \mathbf{r}^{(i)})/\tau}}{\frac{1}{K} \sum_{j=1}^K e^{s(\mathbf{x}^{(i)}, \mathbf{x}^{+(j)}, \mathbf{r}^{(i)})/\tau}} \right]. \tag{6}$$

**Similarity function.** As discussed in §2.2, our goal is to have a similarity function that is flexible yet simple. With $\mathbf{r}$ as a latent variable, we still use the dot-product similarity on embeddings:

$$s(\mathbf{x}, \mathbf{x}^+, \mathbf{r}) = \psi_1(\mathbf{x}, \mathbf{r})^\top \psi_2(\mathbf{x}^+), \quad \text{where } \psi_1(\mathbf{x}, \mathbf{r}) = \frac{t(f(\mathbf{x}), \mathbf{r})}{\|t(f(\mathbf{x}), \mathbf{r})\|_2}, \psi_2(\mathbf{x}^+) = \frac{f(\mathbf{x}^+)}{\|f(\mathbf{x}^+)\|_2}. \tag{7}$$

Specifically, we *edit* $f(\mathbf{x})$ with the help of $\mathbf{r}$ and an editing function $t(\cdot, \cdot)$ such that it lies in the vicinity of $f(\mathbf{x}^+)$. We parameterize $t$ with a linear projection or two-layer MLPs for AdaSSL-V.

We derive a bound for the second term of Eq. 4 in §B:

$$-I_{\tilde{p}}(\mathbf{r}; \mathbf{x}^+ \mid \mathbf{x}) \geq -\mathcal{L}_{\text{Reg}} = -\mathbb{E}_{p(\mathbf{x}, \mathbf{x}^+)}\big[D_{\text{KL}}(q_\phi(\mathbf{r} \mid \mathbf{x}, \mathbf{x}^+) \| p_\theta(\mathbf{r} \mid \mathbf{x}))\big]. \tag{8}$$

---

[3]For brevity, we use the notation $I(\mathbf{x}; \mathbf{x}^+)$ in this section, but in practice we optimize $I(f(\mathbf{x}); f(\mathbf{x}^+)) \leq I(\mathbf{x}; \mathbf{x}^+)$ because our method operates on paired embeddings.

Thus, by introducing a variational distribution on $\mathbf{r}$, we obtain a tractable lower bound on $I(\mathbf{x}; \mathbf{x}^+)$. In practice, we parameterize $q_\phi$ and $p_\theta$ using lightweight MLPs on top of the embeddings $f(\mathbf{x})$ and $f(\mathbf{x}^+)$, modeling both as factorized Gaussians. Plugging the terms into Eq. 5, we get

$$\mathcal{L}_{\text{AdaSSL}-\text{V}} = \mathcal{L}_{\text{SSL}}\left(\mathbb{E}_{q_\phi}\psi_1(\mathbf{x}, \mathbf{r}), \psi_2(\mathbf{x}^+)\right) + \beta D_{\text{KL}}(q_\phi(\mathbf{r} \mid \mathbf{x}, \mathbf{x}^+)\|p_\theta(\mathbf{r} \mid \mathbf{x})). \qquad (9)$$

We call this variant of our method **AdaSSL-V**(variational). For InfoNCE, The first term is explicilty stated in Eq. 6. For BYOL, we replace the input to the predictor $\eta(\cdot)$ in Eq. 3 with $\psi_1(\mathbf{x}, \mathbf{r})$.

*Remark.* Although AdaSSL-V is only theoretically justified for contrastive SSL, one can use a distillation-based SSL loss as well because they still encourage $\mathbf{r}$ to aid prediction of $f(\mathbf{x}^+)$.

### 3.3 ADASSL-S AND SPARSE MODULAR EDITS

Natural transitions usually correspond to sparse changes in the latent factors, an inductive bias widely adopted in the CRL literature (Ahuja et al., 2022; Klindt et al., 2021; Lippe et al., 2023). Therefore, we hypothesize that we can implement Eq. 5 by predicting $\mathbf{r}$ and regularizing its sparsity. **AdaSSL-S**(parse) realizes this idea. Instead of variational approximation, we predict $\mathbf{r}$ deterministically from $f(\mathbf{x})$ and $f(\mathbf{x}^+)$: $\mathbf{r} = m(f(\mathbf{x}), f(\mathbf{x}^+))$, where $m$ is an MLP followed by `tanh` activation. We then regularize the sparsity of $\mathbf{r}$:

$$\mathcal{L}_{\text{AdaSSL}-\text{S}} = \mathcal{L}_{\text{SSL}}\left(\psi_1(\mathbf{x}, \mathbf{r}), \psi_2(\mathbf{x}^+)\right) + \beta \|\mathbf{r}\|_0, \qquad (10)$$

where the $L_0$ penalty[4] is made differentiable through the Gumbel-Sigmoid estimator (§E.1). Inspired by Ibrahim et al. (2022); Hu et al. (2022), we use a modular editing function for AdaSSL-S:

$$t(f(\mathbf{x}), \mathbf{r}) = f(\mathbf{x}) + \sum_{i=1}^{d_r} \mathbf{r}_i t_i(f(\mathbf{x})) = f(\mathbf{x}) + \sum_{i=1}^{d_r} \mathbf{r}_i (\mathbf{B}_i \mathbf{A}_i f(\mathbf{x}) + b_i), \qquad (11)$$

where $d_r$ is the dimensionality of $\mathbf{r}$. Each $t_i(\cdot)$ is an affine transformation parameterized by a rank-1 matrix $\mathbf{B}_i \mathbf{A}_i$ and a scalar offset $b_i$. This design is motivated by the assumption that differences between the paired embeddings lie in a low-dimensional latent subspace, where edits are applied.

In §4.4, we will show that AdaSSL-S works well with contrastive learning, but requires additional care for distillation-based methods in some setups.

## 4 EXPERIMENTS

We evaluate AdaSSL in various settings to show it learns diverse and generalizable representations. We start with two benchmarks where we have access to the ground-truth data generating factors. First, we generate numerical data where we systematically increase the complexity of $p(\mathbf{z}^+ \mid \mathbf{z})$, and show AdaSSL mitigates the limitations of existing SSL methods (§4.2). Second, we use 3D-rendered images from 3DIdent (Zimmermann et al., 2021) and show AdaSSL identifies latent factors better than all baselines (§4.3).

For other benchmarks, we do not have access to the ground-truth data generating factors. Instead, we use proxy tasks such as fine-grained classification to evaluate the quality of the learned representations. On natural images from CelebA (Liu et al., 2015), AdaSSL captures fine-grained features and generalizes to OOD data (§4.4). On large-scale image data from iNat-2021 (Van Horn et al., 2021), we show that AdaSSL is more robust to noisy data pairings than vanilla SSL (§D.2). Finally, in §4.4 and §D.5, we show that AdaSSL outperforms baselines on modeling the uncertainty in video transitions on an extended Moving-MNIST dataset (Srivastava et al., 2015; Drozdov et al., 2024).

Throughout this section, we refer to data pairs that differ in the underlying latent factors as *natural pairs* and those constructed using augmented views of the same image as *standard pairs*. We show that natural pairs enable us to learn better representations in these settings.

### 4.1 OVERVIEW OF EXPERIMENTAL PROTOCOL

**Baselines.** Our experiments in §4.2, §4.3 focus on contrastive SSL. As discussed in §2.2, InfoNCE (Chen et al., 2020a; Oord et al., 2018) and AnInfoNCE (Rusak et al., 2025) are the con-

---

[4]We slightly abuse the notation and use $\|\mathbf{r}\|_0$ to denote the number of non-zero elements of $\mathbf{r}$.

Table 1: Linear regression $R^2$ on unimodal $p(\mathbf{z}^+ \mid \mathbf{z})$. All experiments share the same $\Sigma$ and the mixing function $g$ for each trial. Although all models achieve good performance on the training set $p(\mathbf{z})$, a flexible model is crucial to achieving good OOD performance. Values below 0.7 are dimmed.

| $\mathrm{Var}(\mathbf{c}^+ \mid \mathbf{c})$ | Model | MODEL SPACE: UNBOUNDED | | | MODEL SPACE: HYPERSPHERE | | |
|---|---|---|---|---|---|---|---|
| | | $p(\mathbf{z})$ | $\mathcal{N}(0,5\cdot\mathbf{I})$ | $\mathcal{N}(0,5\cdot\mathbf{I})_{\mathrm{OOD}}$ | $p(\mathbf{z})$ | $\mathcal{N}(0,5\cdot\mathbf{I})$ | $\mathcal{N}(0,5\cdot\mathbf{I})_{\mathrm{OOD}}$ |
| - | Identity | $0.7410_{\pm 0.0943}$ | $0.5103_{\pm 0.0374}$ | $0.1243_{\pm 0.0883}$ | $0.7410_{\pm 0.0943}$ | $0.5103_{\pm 0.0374}$ | $0.1243_{\pm 0.0883}$ |
| 0 | InfoNCE | $0.9912_{\pm 0.0051}$ | $0.9614_{\pm 0.0060}$ | $0.8924_{\pm 0.0590}$ | $0.8657_{\pm 0.1462}$ | $0.8004_{\pm 0.0764}$ | $0.2683_{\pm 0.2626}$ |
| 1 | InfoNCE | $0.9943_{\pm 0.0031}$ | $0.9731_{\pm 0.0070}$ | $0.9564_{\pm 0.0074}$ | $0.9785_{\pm 0.0178}$ | $0.9104_{\pm 0.0154}$ | $0.6944_{\pm 0.0657}$ |
| | H-InfoNCE$_{\mathrm{Affine}}$ | $0.9956_{\pm 0.0019}$ | $0.9736_{\pm 0.0080}$ | $0.9592_{\pm 0.0072}$ | $0.9953_{\pm 0.0021}$ | $0.9645_{\pm 0.0065}$ | $0.9154_{\pm 0.0100}$ |
| Anisotropic | InfoNCE | $0.9968_{\pm 0.0013}$ | $0.9764_{\pm 0.0055}$ | $0.9668_{\pm 0.0056}$ | $0.9509_{\pm 0.0358}$ | $0.7755_{\pm 0.1385}$ | $0.3523_{\pm 0.2323}$ |
| | AnInfoNCE | $0.9962_{\pm 0.0019}$ | $0.9753_{\pm 0.0068}$ | $0.9627_{\pm 0.0088}$ | $0.9613_{\pm 0.0418}$ | $0.8403_{\pm 0.0299}$ | $0.4022_{\pm 0.2316}$ |
| | H-InfoNCE$_{\mathrm{Affine}}$ | $0.9963_{\pm 0.0019}$ | $0.9685_{\pm 0.0032}$ | $0.9510_{\pm 0.0023}$ | $0.9970_{\pm 0.0017}$ | $0.9537_{\pm 0.0149}$ | $0.9018_{\pm 0.0035}$ |
| Heteroscedastic (affine+activation) | InfoNCE | $0.8553_{\pm 0.0532}$ | $0.2664_{\pm 0.0984}$ | $-0.1891_{\pm 0.2545}$ | $0.7851_{\pm 0.0920}$ | $0.2690_{\pm 0.1024}$ | $0.0209_{\pm 0.1110}$ |
| | AnInfoNCE | $0.8447_{\pm 0.0611}$ | $0.2745_{\pm 0.1052}$ | $-0.2277_{\pm 0.3284}$ | $0.7563_{\pm 0.1276}$ | $0.2563_{\pm 0.1092}$ | $0.0070_{\pm 0.1230}$ |
| | H-InfoNCE$_{\mathrm{Affine}}$ | $0.9826_{\pm 0.0060}$ | $0.9482_{\pm 0.0165}$ | $0.8666_{\pm 0.0741}$ | $0.9426_{\pm 0.0222}$ | $0.6276_{\pm 0.1084}$ | $0.3106_{\pm 0.1218}$ |
| | H-InfoNCE$_{\mathrm{MLP}}$ | $0.9892_{\pm 0.0023}$ | $0.9610_{\pm 0.0098}$ | $0.9149_{\pm 0.0348}$ | $0.9856_{\pm 0.0075}$ | $0.9288_{\pm 0.0175}$ | $0.7633_{\pm 0.0576}$ |

trastive baselines that account for isotropic and anisotropic noise in $p(\mathbf{z}^+ \mid \mathbf{z})$, respectively. AnInfoNCE learns directional weights of the similarity function, $\boldsymbol{\Lambda}$. For a fair comparison, we also use a learnable scalar weight $\lambda$ for other methods in §4.2 and §4.4 and find it beneficial. On natural images in §4.4, we experiment with both InfoNCE and BYOL (Grill et al., 2020). We also compare with Ibrahim et al. (2022), which models the change between latent factors as Lie group transformations; we denote this method as LieSSL. For the CRL benchmark in §4.3, we include classic disentanglement methods, including $\beta$-VAE (Higgins et al., 2017) and AdaGVAE (Locatello et al., 2020). For the video experiments in §4.4, we use BYOL as our base SSL method.

**H-InfoNCE.** In addition to existing baselines, we introduce H-InfoNCE, which extend AnInfoNCE to account for heteroscedastic noise by predicting $\boldsymbol{\Lambda}_{\mathbf{x}}$ from $f(\mathbf{x})$ with an affine function (H-InfoNCE$_{\mathrm{Affine}}$) or an MLP (H-InfoNCE$_{\mathrm{MLP}}$); it replaces $\boldsymbol{\Lambda}$ in AnInfoNCE's similarity function with this conditional $\boldsymbol{\Lambda}_{\mathbf{x}}$ (see Table 8). Additionally, H-InfoNCE uses another MLP predictor to predict $f(\mathbf{x}^+)$ from $f(\mathbf{x})$, similar to distillation-based SSL, except for in Table 1, where we ensure $\mathbb{E}[\mathbf{z}^+ \mid \mathbf{z}] = \mathbf{z}$. Note that all InfoNCE, AnInfoNCE, and H-InfoNCE assume unimodal $p(\mathbf{z}^+ \mid \mathbf{z})$ while AdaSSL relaxes this assumption by using a latent-variable model.

**Experimental setup.** We use a five-layer MLP as $f$ for the numerical experiments in §4.2, a ResNet-18 *encoder* followed by a two-layer MLP *projector* as $f$ for the image experiments in §4.3 and §4.4, a ResNet-50 encoder followed by a two-layer MLP projector as $f$ in §D.2, and a five-layer 3D CNN followed by a three-layer MLP projector as $f$ for videos. We train all models from random initializations. For evaluation in §4.2 and §4.3, we follow Zimmermann et al. (2021) by training an affine probe on top of the *embeddings* produced by the frozen $f$ on the training data. For evaluation in §4.4, we train an affine probe on both the embeddings (output of the frozen projector) and the output of the frozen encoder, which we refer to as *representations*. We then evaluate the probes' performance on the test set following standard practice (Chen et al., 2020a; Grill et al., 2020). All results are reported as the mean and standard deviation over at least three random seeds. Additional experimental details and data visualizations can be found in §E.

## 4.2 NUMERICAL DATA

In this section, we study the effect of complexity of the conditional variance in $p(\mathbf{z}^+ \mid \mathbf{z})$. Specifically, we sample correlated latents $\mathbf{c} \sim \mathcal{N}(0, \Sigma)$, where $\Sigma$ is sampled from an Inverse-Wishart distribution (mean $\mathbf{I}$, with typically nonzero correlations), and sample $\mathbf{c}^+$ from different conditional distributions $p(\mathbf{c}^+ \mid \mathbf{c})$. Style latents are sampled independently: $\mathbf{s}, \mathbf{s}^+ \sim \mathcal{N}(0, \mathbf{I})$, yielding $\mathbf{z} = [\mathbf{c}, \mathbf{s}]$ and $\mathbf{z}^+ = [\mathbf{c}^+, \mathbf{s}^+]$. Denote the training latent distribution as $p(\mathbf{z})$. A random invertible MLP parameterizing $g$ (details in §E.2) maps these latents to observations $\mathbf{x}, \mathbf{x}^+$ via Eq. 1. We then train $f$ on the $(\mathbf{x}, \mathbf{x}^+)$ pairs under the SSL algorithms and freeze it.

Next, we train a linear regressor to predict $\mathbf{c}$ from the frozen embeddings, $f(\mathbf{x}) = f(g([\mathbf{c}, \mathbf{s}]))$. We perform three evaluations by varying the training and testing distributions *of the regressor*:

- $p(\mathbf{z})$: both train and test use $\mathbf{z} \sim p(\mathbf{z})$.
- $\mathcal{N}(0, 5 \cdot \mathbf{I})$: both train and test use $\mathbf{z} \sim \mathcal{N}(0, 5 \cdot \mathbf{I})$. This tests embeddings under covariate shift.
- $\mathcal{N}(0, 5 \cdot \mathbf{I})_{\mathrm{OOD}}$: train on $\mathbf{z} \sim p(\mathbf{z})$ but test on $\mathbf{z} \sim \mathcal{N}(0, 5 \cdot \mathbf{I})$. This corresponds to a practical scenario where distribution shifts happen after deployment when both the encoder and regressor

Table 2: Identifiability results on 3DIdent. AdaSSL achieves the best disentanglement and $R^2$ scores. "—ǁ—" denotes "same as above".

| Model | Pairing | DCI disent. (↑) | $R^2$ (↑) |
|---|---|---|---|
| $\beta$-VAE$_{\beta=1}$ | - | $0.2076 \pm 0.0243$ | $0.6649 \pm 0.0307$ |
| $\beta$-VAE$_{\beta=16}$ | - | $0.1883 \pm 0.0191$ | $0.6672 \pm 0.0216$ |
| $\beta$-VAE$_{\beta=100}$ | - | $0.3352 \pm 0.0468$ | $0.6691 \pm 0.0342$ |
| AdaGVAE$_{\beta=1}$ | Natural | $0.4098 \pm 0.0413$ | $0.6436 \pm 0.0343$ |
| AdaGVAE$_{\beta=16}$ | —ǁ— | $0.3800 \pm 0.0131$ | $0.6511 \pm 0.0141$ |
| AdaGVAE$_{\beta=100}$ | —ǁ— | $\underline{0.4582} \pm 0.0154$ | $0.6213 \pm 0.0143$ |
| InfoNCE | Standard | $0.1447 \pm 0.0032$ | $0.3382 \pm 0.0074$ |
| AnInfoNCE | —ǁ— | $0.1349 \pm 0.0007$ | $0.3704 \pm 0.0113$ |
| InfoNCE | Natural | $0.1178 \pm 0.0073$ | $0.8184 \pm 0.0047$ |
| AnInfoNCE | —ǁ— | $0.2772 \pm 0.0184$ | $0.8243 \pm 0.0002$ |
| AdaSSL-V$_{Additive}$ | —ǁ— | $\mathbf{0.4661} \pm 0.0467$ | $0.8857 \pm 0.0012$ |
| AdaSSL-V$_{Linear}$ | —ǁ— | $0.2756 \pm 0.0266$ | $\mathbf{0.9331} \pm 0.0077$ |
| AdaSSL-V$_{MLP}$ | —ǁ— | $0.1027 \pm 0.0048$ | $0.8948 \pm 0.0017$ |
| AdaSSL-S | —ǁ— | $0.1777 \pm 0.1009$ | $\underline{0.9309} \pm 0.0096$ |

Figure 3: AdaSSL-V performs controllable retrieval. From the query image $\mathbf{x}$, we sample $\tilde{\mathbf{r}}$ from different dimensions of the learned prior $p_\theta(\mathbf{r} \mid \mathbf{x})$ which correspond to interpretable changes in the edited image $t(f(\mathbf{x}), \tilde{\mathbf{r}})$.

are frozen. This setting requires not only a robust regressor but also that the representation be well aligned across the two distributions (Ruan et al., 2022).

Following prior works (Zimmermann et al., 2021; Kügelgen et al., 2021), we vary the latent space assumptions (unbounded or hypersphere) and model flexibility (InfoNCE, AnInfoNCE, or H-InfoNCE) by changing the similarity function.

We first construct a unimodal conditional, where we expect H-InfoNCE to suffice. We sample $\mathbf{c}^+$ following $c_i^+ \mid \mathbf{c} \sim \mathcal{N}(c_i^+; c_i, \sigma(\mathbf{c})_i^2)$, with $\sigma(\mathbf{c})$ either $0$, isotropic, anisotropic, or heteroscedastic, where $\sigma(\cdot)$ is a random affine function followed by `softplus` activation.

Table 1 leads to two main observations. First, models achieve high performance when both their embedding space and model flexibility match the true conditional $p(\mathbf{c}^+ \mid \mathbf{c})$; otherwise we see a decrease in performance, which corroborates the findings of Zimmermann et al. (2021). Notably, we see a clear performance drop with InfoNCE and AnInfoNCE with normalized embedding space. H-InfoNCE improves the performance by a large margin by capturing heteroscedasticity, consistent with Proposition 2.1. Second, while latent correlations help all models perform well on in-distribution data $p(\mathbf{z})$, flexible models' improved performance is more pronounced under OOD evaluation. Under heteroscedastic noise, the encoders learned with InfoNCE and AnInfoNCE fall short, even trailing the identity function. Interestingly, when $\mathrm{Var}(\mathbf{c}^+ \mid \mathbf{c}) = 0$, generalization performance of InfoNCE is weaker than the best models in each block, supporting our hypothesis that naturally varying pairs that have independent variation along latent factors can help generalization.

In §D.1, we show that H-InfoNCE also improves upon baselines on more complex (heteroscedastic and multimodal) distributions of $p(\mathbf{c}^+ \mid \mathbf{c})$, and AdaSSL provides further improvements.

## 4.3 SYNTHETIC IMAGES

We next show AdaSSL can be used to recover *all* data-generating factors from natural pairs on 3DIdent (Zimmermann et al., 2021), a dataset of realistically rendered images of a teapot varying in ten data generating factors such as position, spotlight, and hue. Following Locatello et al. (2020), we generate natural pairs by first drawing two samples from the marginal latent distribution. Then, each latent coordinate is replaced with some probability by the corresponding coordinate from the other sample (details in §E.3). We evaluate (a) disentanglement in the learned embeddings with the DCI disentanglement score (Eastwood & Williams, 2018), and (b) the recovery of latent factors, i.e., "empirical" identifiability, up to affine transformations with $R^2$.

Table 2 shows that $\beta$-VAE and AdaGVAE fail to identify the latent factors, though AdaGVAE achieves decent disentanglement. InfoNCE with augmentations performs worse, likely because augmentation invariance conflicts with the CRL objective. SSL baselines using natural pairs achieve good latent recovery but yield more entangled latent factors. We hypothesize that AdaSSL's regularization encourages efficient encodings of $\mathbf{r}$, akin to $\beta$-VAE (Higgins et al., 2017; Burgess et al., 2018). Since $\mathbf{r}$ is modeled as factorized Gaussians, some disentanglement in $\mathbf{r}$ is expected. To verify this, we vary the complexity of the editing function $t$ (additive, linear, nonlinear) denoted by sub-

Table 3: Linear $F_1$ scores on representations (encoder output) and embeddings (projector output) trained on CelebA, under weak or strong augmentations. AdaSSL+GT, a soft performance upper bound, uses the ground-truth attribute difference as $\mathbf{r}$.

|  | Model | Pairing | WEAK AUGMENTATION Repr. | Emb. | STRONG AUGMENTATION Repr. | Emb. |
|---|---|---|---|---|---|---|
| CONTRASTIVE | InfoNCE | Standard | $0.2698_{\pm 0.0030}$ | $0.1295_{\pm 0.0051}$ | $0.5965_{\pm 0.0004}$ | $0.5694_{\pm 0.0011}$ |
| | AnInfoNCE | —"— | $0.2534_{\pm 0.0064}$ | $0.1822_{\pm 0.0036}$ | $\underline{0.5967}_{\pm 0.0015}$ | $\mathbf{0.5742}_{\pm 0.0030}$ |
| | InfoNCE | Natural | $0.5473_{\pm 0.0027}$ | $0.3747_{\pm 0.0051}$ | $0.5784_{\pm 0.0008}$ | $0.4941_{\pm 0.0035}$ |
| | AnInfoNCE | —"— | $0.5413_{\pm 0.0010}$ | $0.4249_{\pm 0.0032}$ | $0.5789_{\pm 0.0008}$ | $0.4987_{\pm 0.0033}$ |
| | LieSSL | —"— | $0.5029_{\pm 0.0061}$ | $0.4525_{\pm 0.0098}$ | $0.5926_{\pm 0.0036}$ | $0.5685_{\pm 0.0015}$ |
| | H-InfoNCE$_{\mathrm{MLP}}$ | —"— | $0.5521_{\pm 0.0042}$ | $0.4559_{\pm 0.0058}$ | $0.5789_{\pm 0.0016}$ | $0.5138_{\pm 0.0023}$ |
| | AdaSSL-V | —"— | $\mathbf{0.5784}_{\pm 0.0025}$ | $\mathbf{0.4794}_{\pm 0.0015}$ | $\mathbf{0.6014}_{\pm 0.0008}$ | $\underline{0.5706}_{\pm 0.0034}$ |
| | AdaSSL-S | —"— | $\underline{0.5676}_{\pm 0.0049}$ | $\underline{0.4581}_{\pm 0.0016}$ | $0.5911_{\pm 0.0014}$ | $0.5654_{\pm 0.0007}$ |
| | AdaSSL+GT | —"— | $0.6818_{\pm 0.0011}$ | $0.6840_{\pm 0.0019}$ | $0.6779_{\pm 0.0003}$ | $0.6832_{\pm 0.0011}$ |
| DISTILLATION | BYOL | Standard | $0.2989_{\pm 0.0025}$ | $0.1832_{\pm 0.0037}$ | $0.5368_{\pm 0.0013}$ | $\underline{0.5043}_{\pm 0.0037}$ |
| | BYOL | Natural | $\underline{0.5465}_{\pm 0.0018}$ | $\underline{0.4263}_{\pm 0.0019}$ | $\underline{0.5608}_{\pm 0.0004}$ | $0.5019_{\pm 0.0013}$ |
| | AdaSSL-V | —"— | $\mathbf{0.5816}_{\pm 0.0035}$ | $\mathbf{0.5067}_{\pm 0.0051}$ | $\mathbf{0.5702}_{\pm 0.0017}$ | $\mathbf{0.5302}_{\pm 0.0018}$ |
| | AdaSSL-S | —"— | $0.4520_{\pm 0.0130}$ | $0.3014_{\pm 0.0107}$ | $0.5334_{\pm 0.0057}$ | $0.4772_{\pm 0.0051}$ |
| | AdaSSL+GT | —"— | $0.5948_{\pm 0.0042}$ | $0.5730_{\pm 0.0016}$ | $0.5984_{\pm 0.0024}$ | $0.5872_{\pm 0.0003}$ |

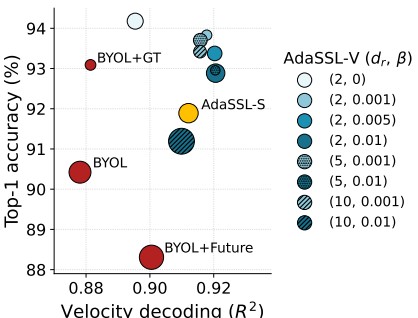

Figure 4: Velocity decoding and digit recognition from representations on stochastic Moving-MNIST. Marker radius indicates standard deviation.

scripts. With simpler $t$, the dimensions of $f(\mathbf{x})$ must align more directly with those of $\mathbf{r}$, making any disentanglement in $\mathbf{r}$ more visible in $f(\mathbf{x})$. Indeed, simpler $t$ leads to more disentagled embeddings while consistently outperforming baselines on regression. In particular, AdaSSL-V$_{\mathrm{Linear}}$ and AdaSSL-S, which both use linear editing functions, achieve the highest $R^2$.

To better understand the learned $\mathbf{r}$, we visualize its effect by retrieving nearest neighbor of a query image $\mathbf{x}$ after editing it with samples $\tilde{\mathbf{r}}$ (Fig. 3). Given evidence of disentanglement, we expect sampling specific latent dimensions to induce meaningful changes in the edited embeddings $t(f(\mathbf{x}), \tilde{\mathbf{r}})$. Concretely, we sample $\tilde{\mathbf{r}}_i \sim p_\theta(\mathbf{r}_i \mid \mathbf{x})$ for $i \in \mathbb{L} \subseteq [d_r]$ for some set of latent indices $\mathbb{L}$, and fix all others to their expectations. Fig. 3 shows results for three different $\mathbb{L}$'s. We find that we can retrieve objects that differ in position, spotlight, and color, while leaving most other factors unchanged, though orientation remains entangled with other factors. Finally, when sampling from the full prior, we retrieve images that differ sparsely in latent factors, consistent with the training DGP.

Together, these results highlight SSL as a promising path for CRL given its efficiency (no reconstruction) and demonstrated scalability to high-dimensional images.

## 4.4 NATURAL IMAGES AND VIDEOS

**Natural images.** Although we do not have access to the ground-truth data generating factors of natural images, we perform experiments on CelebA (Liu et al., 2015) which contains celebrity images with annotated facial attributes. We obtain real-world natural pairs by matching different photos of the same celebrity, which differ sparsely in their facial attributes (§E.4). We then train models on paired images and evaluate with affine probes on 40 facial attributes of unseen identities, inducing a natural distribution shift. Results in Table 3 show that standard pairing rely on strong augmentations to work well. However, using natural pairs largely reduces the gap, and AdaSSL-V is the only method that consistently improves upon the standard pairing baselines across all settings. This exposes the weakness of vanilla SSL objectives under complex conditionals in natural pairs. We still observe a gap between AdaSSL and AdaSSL+GT, indicating room for improvement in future work.

Interestingly, while AdaSSL-S consistently improves InfoNCE on natural pairs, this benefit is not observed with BYOL. We hypothesize that explicitly regularizing the embeddings' information content (as in the denominator of Eq. 2) pressures the model to utilize $\mathbf{r}$ effectively. In contrast, distillation-based methods lack this direct tension and may require additional regularization on $\mathbf{r}$ to achieve the same effect. We provide empirical support of this hypothesis in §D.4. We show AdaSSL scales to larger data and backbones in §D.2.

**World modeling on videos.** In sections above, we have shown AdaSSL models $p(\mathbf{z}^+ \mid \mathbf{z})$ well. Since modeling this transition distribution is central to world modeling on videos, we test AdaSSL on it. We hypothesize that inability to model uncertainty drives the model to discard variant factors. We introduce uncertainty by injecting random changes in velocity between two segments of Moving-MNIST (Srivastava et al., 2015; Drozdov et al., 2024), which are then used as positive pairs. We use BYOL (Grill et al., 2020) as the SSL method for this experiment, whose predictions can condition on a future segment (BYOL+Future) similar to Liu et al. (2025) or the ground-truth

change in velocity (BYOL+GT). Fig. 4 shows that AdaSSL captures both the invariant factor, digit, and the variant factor, velocity, better than baselines. Ablation on AdaSSL-V shows its robustness to the dimensionality of **r** under proper regularization. We include empirical analysis on AdaSSL's world modeling capabilities in §D.5 and additional experimental details in §E.5.

## 5    RELATED WORK

**Self-supervised learning.**    SSL in the latent space has evolved from solving hand-crafted *pretext tasks* (Noroozi & Favaro, 2016; Doersch et al., 2015; Dosovitskiy et al., 2014; Gidaris et al., 2018) to learning semantic-preserving representations from invariance to augmentations (Oord et al., 2018; Wu et al., 2018; Gutmann & Hyvärinen, 2010; Chen et al., 2020b; Caron et al., 2020; Wu et al., 2018; He et al., 2020; Radford et al., 2021; Caron et al., 2021; Zbontar et al., 2021; Bardes et al., 2022; Ermolov et al., 2021; Chen & He, 2021; Grill et al., 2020; Assran et al., 2023; Baevski et al., 2022; Caron et al., 2020; He et al., 2016). Studies have also explored the relationship between invariant representations and variational inference (Bizeul et al., 2024; Sinha & Dieng, 2021). Beyond invariance, equivariant representations preserve transformation information (Hinton et al., 2011). In SSL, this is achieved by providing augmentation parameters to the predictor (Garrido et al., 2023b; Ghaemi et al., 2024; Devillers & Lefort, 2023; Garrido et al., 2024; Park et al., 2022), or using subspaces for different invariances (Xiao et al., 2021; Eastwood et al., 2023). However, these approaches are tied to chosen augmentations and break down when the sources of uncertainty are unknown. Alternatively, one can exploit the invariance between observation pairs that are transformed similarly (Shakerinava et al., 2022), or model transformation with Lie groups (Ibrahim et al., 2022). Unlike prior work, our method does not require transformation labels, handles multiple varying factors, and provides a simple, theoretically justified objective that is compatible with standard SSL methods across diverse settings.

**Causal representation learning.**    Much research examines recovering data-generating factors and their causal relations (Hyvarinen & Morioka, 2016; Schölkopf et al., 2021; von Kügelgen et al., 2023; Ahuja et al., 2023; Brehmer et al., 2022; Locatello et al., 2020; Lachapelle et al., 2022; Lippe et al., 2023; Klindt et al., 2021; Ahuja et al., 2022; Lippe et al., 2022; Yao et al., 2025). While offering theoretical guarantees, these methods often rely on strong assumptions or probabilistic generative models, limiting scalability. SSL has been connected to CRL (Zimmermann et al., 2021; Kügelgen et al., 2021; Rusak et al., 2025; Yao et al., 2024), where studies focus on identifying the content factors that follow simple conditionals (§2.2). This work relaxes these assumptions by allowing structured variation between paired latents and demonstrates strong performance on weakly-supervised CRL, a step towards understanding and advancing SSL (Reizinger et al., 2025).

**World modeling with SSL.**    Unlike image-based SSL that rely on augmentations, video world models with SSL learn the transition dynamics of videos, often by predicting target frames given some context (Sermanet et al., 2018; Feichtenhofer et al., 2021; Bardes et al., 2024; Assran et al., 2025; Schwarzer et al., 2021; Guo et al., 2022). Through the process, the model learns useful representations for downstream tasks such as video understanding. A key challenge is that uncertainty grows with the temporal gap between positive pairs, forcing models to fix temporal resolution (Feichtenhofer et al., 2021; Bardes et al., 2024), which may limit their ability to learn features at different levels of abstractions (Zacks & Tversky, 2001) because the model can discard variant factors. Introducing a latent variable **r**, as we do, can reduce the uncertainty and learn more diverse features (§4.4). Finally, although we focus on improving SSL in the latent space, we note there are successful approaches that predict in the observation space (Schmidt & Jiang, 2024; Tong et al., 2022; Feichtenhofer et al., 2022; Jang et al., 2024; Bruce et al., 2024; Yang et al., 2024).

## 6    CONCLUSION

In this work, we reveal the limitation of SSL methods when trained on naturally paired data and introduce AdaSSL, which learns a latent variable that captures the uncertainty in prediction targets. Our approach consistently outperforms existing methods across all benchmarks. We believe this is a promising step in expanding the capability of SSL methods, leading to potentially fruitful advancements in learning generalizable representations, identifying latent factors from high-dimensional images, and world modeling with uncertainty.

ACKNOWLEDGMENTS

We appreciate the constructive feedback from the anonymous reviewers. We also thank Siddarth Venkatraman, Michael Chong Wang, Emiliano Penaloza, and Omar Salemohamed for insightful discussions, Anirudh Buvanesh and Lucas Maes for precise pointers to literature, and Mehran Shakerinava for proofreading. Additionally, YZ would like to thank Xiaofeng Zhang and Dhanya Sridhar for helpful feedback during the early development of the idea.

YZ, HG, EBM, and LC acknowledge the generous support of the CIFAR AI Chair program. Additionally, YZ is supported by the AI Scholarship from Université de Montréal. HG is supported by the UNIQUE Centre (unique.quebec). EBM is supported by NSERC Discovery Grants (RGPIN-2022-05033). SB is supported by NSERC Discovery Grants (RGPIN-2023-03875) and the Canada Excellence Research Chairs (CERC) Program. This research was enabled in part by compute resources provided by Mila (mila.quebec) and the Digital Research Alliance of Canada (alliancecan.ca).

ETHICS STATEMENT

This work uses the CelebA dataset (Liu et al., 2015), which consists of publicly available celebrity face images collected from the internet. We use it solely for non-commercial academic research in facial attribute learning, under its research-only license. Our use does not involve face generation or manipulation, and all experiments are conducted strictly for evaluating algorithmic performance, in line with the intended use of the dataset.

REPRODUCIBILITY STATEMENT

We comprehensively detail our experimental setup in §E. Code to reproduce our results is available at https://github.com/SkrighYZ/AdaSSL.

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

# APPENDIX

## A LIMITATIONS AND FUTURE WORK

In this section, we discuss the limitations of our work and suggest future directions.

First, while we perform experiments on controlled settings with ResNet-18/50, it would be interesting to explore AdaSSL's behavior with ViT backbones and on uncurated data, such as web-scale image-caption pairs and natural videos. On such data, one can investigate the types of uncertainty (e.g., object-level versus global, kinematic versus intrinsic) that AdaSSL prioritizes to learn. Two recent works, while potentially incorporating stronger inductive bias, share a similar conceptual motivation to AdaSSL on these specific settings (Lavoie et al., 2024; Hoang & Ren, 2025).

Second, the world model $t$ in AdaSSL must reason about how the latent cause $\mathbf{r}$ transforms the environment, i.e., $p(\mathbf{z}^+ \mid \mathbf{z}, \mathbf{r})$, a task that naturally scales in difficulty with the complexity of the underlying transition. However, we hypothesize that operating in the latent space makes this transition distribution considerably easier to model than in observation space. Future work could explore more structured transition models to better capture how latent factors drive world dynamics.

Third, a natural extension of AdaSSL is action-free representation learning on videos, where the model discovers an implicit action space from naturally occurring changes. This learned action space can then support downstream tasks. For example, one may train a decoder on the frozen world model and prior over $\mathbf{r}$ for controllable video generation. We can also learn a projection between actions and the learned latent space to enable *planning* (Sobal et al., 2025b; Assran et al., 2025). We perform a preliminary step of this idea where we evaluate the world model's capability to predict future trajectories with a given action in §D.5.2.

Finally, we do not provide theoretical guarantees for identifiability and leave it as future work.

## B DERIVATION OF EQ. 8

$$-I_{\tilde{p}}(\mathbf{r}; \mathbf{x}^+ \mid \mathbf{x})$$
$$= -\mathbb{E}_{\tilde{p}}\left[\log \frac{q(\mathbf{r} \mid \mathbf{x}, \mathbf{x}^+)}{\tilde{p}(\mathbf{r} \mid \mathbf{x})}\right]$$
$$= -\mathbb{E}_{\tilde{p}}\left[\log \frac{q(\mathbf{r} \mid \mathbf{x}, \mathbf{x}^+)}{\tilde{p}(\mathbf{r} \mid \mathbf{x})} + \log p(\mathbf{r} \mid \mathbf{x}) - \log p(\mathbf{r} \mid \mathbf{x})\right]$$
$$= -\mathbb{E}_{\tilde{p}}\left[\log \frac{q(\mathbf{r} \mid \mathbf{x}, \mathbf{x}^+)}{p(\mathbf{r} \mid \mathbf{x})} + \log \frac{p(\mathbf{r} \mid \mathbf{x})}{\tilde{p}(\mathbf{r} \mid \mathbf{x})}\right]$$
$$= -\mathbb{E}_{p(\mathbf{x},\mathbf{x}^+)}\big[D_{\mathrm{KL}}(q(\mathbf{r} \mid \mathbf{x}, \mathbf{x}^+) \| p(\mathbf{r} \mid \mathbf{x}))\big] + \mathbb{E}_{\tilde{p}}\left[\log \frac{\tilde{p}(\mathbf{r} \mid \mathbf{x})}{p(\mathbf{r} \mid \mathbf{x})}\right]$$
$$= -\mathbb{E}_{p(\mathbf{x},\mathbf{x}^+)}\big[D_{\mathrm{KL}}(q(\mathbf{r} \mid \mathbf{x}, \mathbf{x}^+) \| p(\mathbf{r} \mid \mathbf{x}))\big] + \int \int \int p(\mathbf{x})p(\mathbf{x}^+ \mid \mathbf{x})q(\mathbf{r} \mid \mathbf{x}, \mathbf{x}^+) \log \frac{\tilde{p}(\mathbf{r} \mid \mathbf{x})}{p(\mathbf{r} \mid \mathbf{x})} d\mathbf{r} d\mathbf{x}^+ d\mathbf{x}$$
$$= -\mathbb{E}_{p(\mathbf{x},\mathbf{x}^+)}\big[D_{\mathrm{KL}}(q(\mathbf{r} \mid \mathbf{x}, \mathbf{x}^+) \| p(\mathbf{r} \mid \mathbf{x}))\big] + \int \int p(\mathbf{x})\left(\int p(\mathbf{x}^+ \mid \mathbf{x})q(\mathbf{r} \mid \mathbf{x}, \mathbf{x}^+)d\mathbf{x}^+\right) \log \frac{\tilde{p}(\mathbf{r} \mid \mathbf{x})}{p(\mathbf{r} \mid \mathbf{x})} d\mathbf{r} d\mathbf{x}$$
$$= -\mathbb{E}_{p(\mathbf{x},\mathbf{x}^+)}\big[D_{\mathrm{KL}}(q(\mathbf{r} \mid \mathbf{x}, \mathbf{x}^+) \| p(\mathbf{r} \mid \mathbf{x}))\big] + \int \int p(\mathbf{x})\tilde{p}(\mathbf{r} \mid \mathbf{x}) \log \frac{\tilde{p}(\mathbf{r} \mid \mathbf{x})}{p(\mathbf{r} \mid \mathbf{x})} d\mathbf{r} d\mathbf{x}$$
$$= -\mathbb{E}_{p(\mathbf{x},\mathbf{x}^+)}\big[D_{\mathrm{KL}}(q(\mathbf{r} \mid \mathbf{x}, \mathbf{x}^+) \| p(\mathbf{r} \mid \mathbf{x}))\big] + \mathbb{E}_{p(\mathbf{x})\tilde{p}(\mathbf{r}|\mathbf{x})}\left[\log \frac{\tilde{p}(\mathbf{r} \mid \mathbf{x})}{p(\mathbf{r} \mid \mathbf{x})}\right]$$
$$= -\mathbb{E}_{p(\mathbf{x},\mathbf{x}^+)}\big[D_{\mathrm{KL}}(q(\mathbf{r} \mid \mathbf{x}, \mathbf{x}^+) \| p(\mathbf{r} \mid \mathbf{x}))\big] + \mathbb{E}_{p(\mathbf{x})}[D_{\mathrm{KL}}(\tilde{p}(\mathbf{r} \mid \mathbf{x}) \| p(\mathbf{r} \mid \mathbf{x}))]$$
$$\geq -\mathbb{E}_{p(\mathbf{x},\mathbf{x}^+)}\big[D_{\mathrm{KL}}(q(\mathbf{r} \mid \mathbf{x}, \mathbf{x}^+) \| p(\mathbf{r} \mid \mathbf{x}))\big].$$

## C THEORY

**Lemma C.1.** *Let $A \in \mathbb{R}^{m \times n}$ and let $\Sigma \in \mathbb{R}^{n \times n}$ be symmetric positive definite. Then*
$$\mathrm{range}(A\Sigma A^\top) = \mathrm{range}(A).$$

*Proof.* For any $x \in \mathbb{R}^m$, we have

$$x^\top (A\Sigma A^\top)x = (A^\top x)^\top \Sigma (A^\top x).$$

Since $\Sigma$ is symmetric positive definite, the right-hand side is zero if and only if $A^\top x = 0$. Thus,

$$\ker(A\Sigma A^\top) = \ker(A^\top).$$

Taking orthogonal complements yields

$$\mathrm{range}(A\Sigma A^\top) = \mathrm{range}(A).$$

$\square$

*Remark.* This is a standard linear algebra fact; we include it here for completeness.

**Proposition C.1.** *Let $\mathbb{S}^k \subset \mathbb{R}^{k+1}$ denote the $k$-dimensional unit sphere. Let $g : \mathbb{R}^d \to \mathbb{R}^{d'}$ be $C^1$ diffeomorphic to its image, and let $f : \mathbb{R}^{d'} \to \mathbb{S}^k$ be $C^1$ almost everywhere. Define $h := f \circ g : \mathbb{R}^d \to \mathbb{S}^k$. Assume further that the random vectors $z, z^+ \in \mathbb{R}^d$ are sampled as*

$$z \sim p_Z, \qquad z^+ = z + \varepsilon, \quad \varepsilon \sim p_\varepsilon,$$

*where $p_Z$ is not a point mass and $\varepsilon$ is independent of $z$, $\mathbb{E}[\varepsilon] = 0$, and $\mathrm{Cov}(\varepsilon) \succ 0$.*

*Suppose that for $p_Z$-almost every $z$ we have $h(z) \in \mathbb{S}^k$ and $\mathrm{rank}\, Dh(z) = d$. Write $H = h(z)$ and $H^+ = h(z^+)$. Then the conditional law*

$$p_{H^+|H}(h(z^+) \mid h(z)),$$

*is necessarily heteroscedastic: its conditional variance depends on $h(z)$ for $p_Z$-almost every $z$.*

*Proof.* Fix $z$ where $h$ is $C^1$ and $\mathrm{rank}\, Dh(z) = d$. For $\sigma > 0$ small, define $z^+ = z + \sigma\varepsilon$ with $\varepsilon \sim p_\varepsilon$. A first-order Taylor expansion and the delta method give

$$h(z + \sigma\varepsilon) = h(z) + Dh(z)\,\sigma\varepsilon + o(\sigma),$$

which implies

$$\mathrm{Cov}[h(z + \sigma\varepsilon) \mid z] = \sigma^2 Dh(z)\,\Sigma\,Dh(z)^\top + o(\sigma^2).$$

If the conditional covariance were homoscedastic at leading order, there exists a fixed positive semidefinite matrix $C$ such that

$$Dh(z)\,\Sigma\,Dh(z)^\top \equiv C \quad \text{for } p_Z\text{-almost every } z.$$

Let $W := \mathrm{range}(C)$. By Lemma C.1 and $\Sigma \succ 0$ we have

$$\mathrm{range}\left(Dh(z)\right) = \mathrm{range}\left(Dh(z)\Sigma Dh(z)^\top\right) = \mathrm{range}(C) = W,$$

so $\mathrm{range}(Dh(z)) \equiv W$ is the same $d$-dimensional subspace for $p_Z$-almost every $z$. Because $h(z) \in \mathbb{S}^k$ we have $\|h(z)\|^2 \equiv 1$, so differentiating yields

$$h(z)^\top Dh(z) = 0,$$

i.e. $\mathrm{range}(Dh(z)) \subset h(z)^\perp$. Since $\mathrm{range}(Dh(z)) = W$ for almost every $z$, we obtain $W \subset h(z)^\perp$ almost everywhere, hence $h(z) \in W^\perp$ for almost every $z$.

Pick any nonzero $w \in W$. Then $w^\top h(z) = 0$ for almost every $z$, and differentiating gives $w^\top Dh(z) = 0$ for almost every $z$, i.e. $w \perp \mathrm{range}(Dh(z)) = W$. Thus $W \subset W^\perp$, which forces $W = \{0\}$. This contradicts $\mathrm{rank}\, Dh(z) = d > 0$. Therefore the hypothesis that $Dh(z)\Sigma Dh(z)^\top$ is constant in $z$ is false, so the leading-order conditional covariance must depend on $z$ for $p_Z$-almost every $z$. $\square$

*Remark.* The above argument establishes heteroscedasticity at *leading order* in the noise scale $\sigma$, which rigorously shows that the conditional covariance depends on $z$ for sufficiently small $\sigma$. For larger $\sigma$, higher-order terms in the Taylor expansion of $h$ become significant and the exact conditional covariance may be more complicated; nevertheless, the local Jacobian $Dh(z)$ still transforms the noise differently at different points, so the conditional variance remains intuitively location-dependent, even if no simple closed-form expression exists.

**Proposition C.2** (Tangent-space variant of Proposition C.1). *Let $\mathbb{S}^k \subset \mathbb{R}^{k+1}$ denote the $k$-dimensional unit sphere, and $U \subset \mathbb{S}^k$ an open set. Let $g : \mathbb{S}^k \to \mathbb{S}^{k'}$ be $C^1$ diffeomorphic to its image, and let $f : \mathbb{S}^{k'} \to \mathbb{R}^d$ be $C^1$ almost everywhere. Define $h := f \circ g : U \to \mathbb{R}^d$. We assume that $h$ is nondegenerate, i.e., $h(U)$ is not contained in any proper affine subspace of its intrinsic dimension. Suppose that for almost every $z \in U$, the derivative $Dh(z) : T_z\mathbb{S}^k \to \mathbb{R}^d$ has full rank, i.e. $\operatorname{rank} Dh(z) = k$. Assume further that the conditional distribution of $z^+ \in \mathbb{S}^k$ given $z$ is locally Gaussian in the tangent space*

$$p(z^+ \mid z) \propto \exp\left( -(z^+ - z)^\top \Lambda(z^+ - z)\right),$$

*with a constant positive definite diagonal matrix $\Lambda$.*

*Define $H = h(z)$ and $H^+ = h(z^+)$. Then for generic nondegenerate $C^1$ maps $h$, the conditional law*

$$p_{H^+|H}(h(z^+) \mid h(z)),$$

*is heteroscedastic for almost every $z \in U$.*

*Proof.* We construct $z^+$ by a small Gaussian step in $\mathbb{R}^{k+1}$ and normalization:

$$z^+ = \frac{z + \varepsilon}{\|z + \varepsilon\|}, \quad \varepsilon \sim \mathcal{N}(0, \Lambda^{-1}).$$

A first-order approximation for small $\varepsilon$ gives

$$z^+ - z = P_z\varepsilon + O(\|\varepsilon\|^2),$$

where $P_z = I - zz^\top$ is the projector to the tangent space, and the pushforward density on the sphere matches

$$p(z^+ \mid z) \propto \exp\left( -(z^+ - z)^\top \Lambda(z^+ - z)\right)$$

up to higher-order terms.

Fix $z \in U$ where $h$ is $C^1$ and $\operatorname{rank} Dh(z)$ has full rank. A Euclidean Taylor expansion gives

$$h(z^+) = h(z) + Dh(z)(z^+ - z) + O(\|z^+ - z\|^2).$$

Substituting $z^+ - z \approx P_z\varepsilon$

$$h(z^+) = h(z) + Dh(z)P_z\varepsilon + R(z),$$

where $R(z)$ collects higher-order terms, and the leading-order conditional covariance is

$$\operatorname{Cov}(h(z^+) \mid z) = Dh(z)\Sigma_z^{\mathrm{tan}}Dh(z)^\top + R(z), \quad \Sigma_z^{\mathrm{tan}} = P_z\Lambda^{-1}P_z,$$

with $R(z)$ continuous and symmetric.

Suppose that $\operatorname{Cov}(h(z^+) \mid z)$ were constant across $z \in U$. With $\Sigma_z^{\mathrm{tan}} \succ 0$, the range of the leading term $\operatorname{range}(Dh(z)\Sigma_z^{\mathrm{tan}}Dh(z)^\top) = \operatorname{range}(Dh(z))$ would have to be the same subspace $W \subset \mathbb{R}^d$ for almost every $z \in U$.

For any differentiable curve $z(t) \subset U$ through points where $Dh(z(t))$ has full rank, we can write

$$\frac{d}{dt}h(z(t)) = Dh(z(t))\dot{z}(t) \in W$$

Integrating along all such curves in $U$ gives

$$h(U) \subset h(z_0) + W,$$

for some base point $z_0$. This would imply that the image $h(U)$ is contained in a fixed affine subspace $W \subset R^d$, contradicting the nondegeneracy assumption on $h$. Therefore, a constant pushforward covriance can only occur in the trivial case of no noise ($\Sigma_z^{\mathrm{tan}} = 0$, or $\Lambda^{-1} = 0$) or in a highly specific algebraic cancellation between $Dh(z)$ and $\Sigma_z^{\mathrm{tan}}$. For generic nondegenerate $C^1$ maps $h$ and almost every $z \in U$, the conditional covariance is therefore heteroscedastic. $\square$

*Remark.* This is analogous to Proposition C.1, but with domain and codomain swapped; the argument relies on the Jacobian of the map and the local Gaussian structure in the tangent space.

**Proposition C.3** (Extension of Proposition C.1). *Let $g : \mathbb{R}^d \to \mathbb{R}^{d'}$ be a $C^2$ with a local diffeomorphism and $f : \mathbb{R}^{d'} \to \mathcal{M}$ be $C^2$ almost everywhere. Define $h := f \circ g : \mathbb{R}^d \to \mathcal{M}$ where $\mathcal{M}$ is a Riemannian manifold with strictly positive sectional curvature on a nonempty open set. Assume further that the random vectors $z, z^+ \in \mathbb{R}^d$ are sampled as*

$$z \sim p_Z, \qquad z^+ = z + \varepsilon, \quad \varepsilon \sim p_\varepsilon,$$

*where $p_Z$ is not a point mass and $\varepsilon$ is independent of $z$, $\mathbb{E}[\varepsilon] = 0$, and $\mathrm{Cov}(\varepsilon) \succ 0$.*

*Suppose that for $p_Z$-almost every $z$ we have $h(z) \in \mathcal{M}$ and $\mathrm{rank}\, Dh(z) = d$. Write $H = h(z)$ and $H^+ = h(z^+)$. Then the conditional law*

$$p_{H^+ | H}(h(z^+) \mid h(z)),$$

*is necessarily heteroscedastic: its conditional variance depends on $h(z)$ for $p_Z$-almost every $z$.*

*Proof.* Following the same reasoning as in Theorem C.1, homoscedasticity at leading order would require a constant positive semidefinite matrix $C$ such that

$$Dh(z) \, \Sigma \, Dh(z)^\top \equiv C \quad \text{for } p_Z\text{-almost every } z.$$

Since $\Sigma \succ 0$, the above condition is equivalent to requiring that

$$\langle u, v \rangle_\Sigma := u^\top \Sigma v = \langle Dh(z)u, Dh(z)v \rangle_{\mathbb{R}^{k+1}} \quad \forall u, v \in \mathbb{R}^d, \text{ for a.e. } z$$

i.e., $h$ is a local Riemannian isometry from the flat space $(\mathbb{R}^d, \langle \cdot, \cdot \rangle_\Sigma)$ to the positively curved manifold $(\mathcal{M}, g_\mathcal{M})$. However, local isometries preserve sectional curvature (Gauss' Theorema Egregium), so no such local isometry from an open subset of $\mathbb{R}^d$ to an open subset of $\mathcal{M}$ exists. Hence, the homoscedasticity condition cannot hold.

Therefore, for all sufficiently small $\sigma > 0$, the conditional covariance

$$\mathrm{Cov}[h(z + \sigma\varepsilon) \mid z] = \sigma^2 Dh(z) \, \Sigma \, Dh(z)^\top + o(\sigma^2)$$

depends on $z$, and the conditional distribution of $h(z^+)$ given $h(z)$ is necessarily heteroscedastic for $p_Z$-almost every $z$. $\qquad \square$

# D   ADDITIONAL RESULTS AND ANALYSIS

## D.1   COMPLEX $p(\mathbf{z}^+ \mid \mathbf{z})$ ON NUMERICAL DATA

In this experiment, we design a DGP where $p(\mathbf{c}^+ \mid \mathbf{c})$ is both multimodal and heteroscedastic. We hypothesize that natural pairs usually differ sparsely in $\mathbf{c}$, and the differed factors are sometimes conditioned on a latent variable. Therefore, we randomly select some dimensions of $\mathbf{c}^+$ and $\mathbf{c}$ to be shared, while the rest follow Gaussians conditioned on a latent variable $\boldsymbol{\kappa}$, i.e., $c_i, c_i^+ \mid \boldsymbol{\kappa} \sim \mathcal{N}\big(\mu(\boldsymbol{\kappa})_i, \sigma(\boldsymbol{\kappa})_i^2\big)$. See §E.2 for details.

Table 4: Linear regression $R^2$ on complex $p(\mathbf{c}^+ \mid \mathbf{c})$. All models normalize embeddings and AdaSSL outperforms baselines.

| Model | $p(\mathbf{z})$ | $\mathcal{N}(0, 5 \cdot \mathbf{I})$ | $\mathcal{N}(0, 5 \cdot \mathbf{I})_{\mathrm{OOD}}$ |
|---|---|---|---|
| InfoNCE | $0.5210_{\pm 0.1611}$ | $0.5024_{\pm 0.0850}$ | $0.0395_{\pm 0.3141}$ |
| AnInfoNCE | $0.5446_{\pm 0.1745}$ | $0.5578_{\pm 0.1271}$ | $0.1652_{\pm 0.2261}$ |
| H-InfoNCE$_{\mathrm{MLP}}$ | $\underline{0.8750}_{\pm 0.0658}$ | $0.7784_{\pm 0.0915}$ | $0.5471_{\pm 0.2480}$ |
| AdaSSL-V | $0.8609_{\pm 0.0740}$ | $\mathbf{0.8656}_{\pm 0.0195}$ | $\mathbf{0.6638}_{\pm 0.0956}$ |
| AdaSSL-S | $\mathbf{0.9187}_{\pm 0.0174}$ | $\underline{0.8472}_{\pm 0.0292}$ | $\underline{0.6325}_{\pm 0.0737}$ |

Table 4 shows that InfoNCE and AnInfoNCE struggle to recover the latent factors, especially on the hardest OOD evaluation, $\mathcal{N}(0, 5 \cdot \mathbf{I})_{\mathrm{OOD}}$. H-InfoNCE improves performance, and AdaSSL variants improve further.

To understand why AdaSSL outperforms baselines in Table 4, we visualize the aggregated marginal distribution of $\mathbf{z}^+$ implied by the learned predictor, $\mathbb{E}_{\mathbf{z}}[p_{\mathrm{model}}(\mathbf{z}^+ \mid \mathbf{z})]$. We define one Monte-Carlo sample of $p_{\mathrm{model}}(\mathbf{z}^+ \mid \mathbf{z})$ for each $\mathbf{z}$ as follows. For InfoNCE, we first encode the input $\mathbf{x} = g(\mathbf{z})$ and then learn a projection from the embedding space to the ground-truth latent space by training a linear regressor from $f(g(\mathbf{z}))$ to $\mathbf{z}^+$. For H-InfoNCE, we pass $f(g(\mathbf{z}))$ through the predictor and project the predicted representations. For AdaSSL-V, we sample from the learned prior $\tilde{\mathbf{r}} \sim p_\theta(\mathbf{r} \mid \mathbf{x})$ and use $\tilde{\mathbf{r}}$ to edit the embeddings with $t(f(\mathbf{x}), \tilde{\mathbf{r}})$ and project the edited embeddings. We repeat this process for random samples of $\mathbf{z} \sim p(\mathbf{z})$ and visualize them in Fig. 5. InfoNCE embeddings produce overly concentrated densities, indicating their inability to accurately capture complex conditional uncertainties. H-InfoNCE partially corrects this, while AdaSSL best fits the ground-truth distribution, suggesting that its improvement arises from more accurate modeling of the conditional uncertainty.

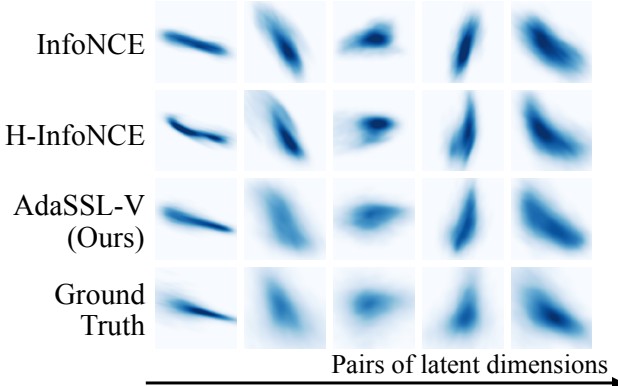

Figure 5: Aggregated marginal distributions $\mathbb{E}_{\mathbf{z}}[p_{\text{model}}(\mathbf{z}^+ \mid \mathbf{z})]$ across latent dimension pairs. InfoNCE produces collapsed densities and H-InfoNCE partially recovers variability, while AdaSSL-V aligns closely with the ground truth. The improvement is most evident in columns two and three, where AdaSSL-V captures both spread and orientation while baselines do not.

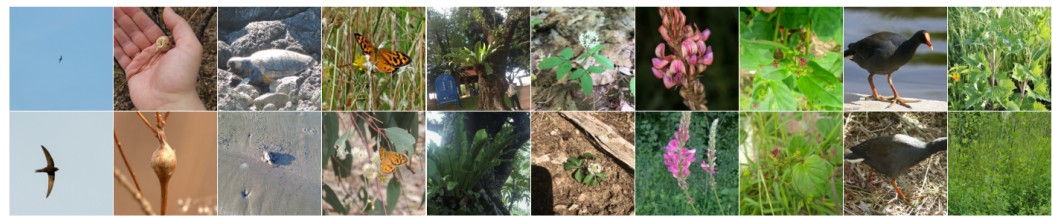

Figure 6: Visualization of images paired by class label from the iNat-1M dataset.

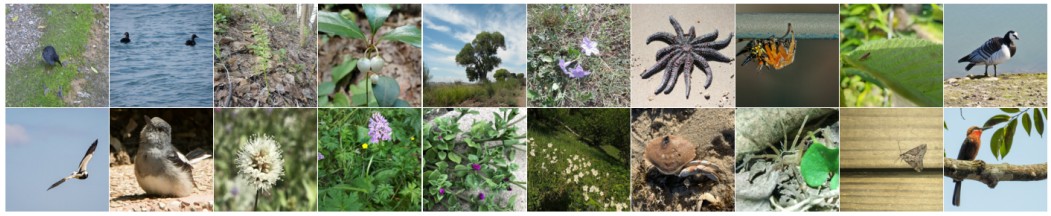

Figure 7: Visualization of images paired by superclass label from the iNat-1M dataset.

## D.2 ROBUSTNESS TO NOISY DATA PAIRINGS ON INAT-1M

In this experiment, we train InfoNCE and AdaSSL-V with a ResNet-50 backbone at 224×224 resolution on a subset of iNat2021 (Van Horn et al., 2021), a large-scale image dataset that contains fine-grained (class) and coarse-grained (superclass) species labels. We call this subset iNat-1M, which contains one million training and 250 000 validation images across 5000 fine-grained classes and 11 superclasses. Fine-grained classification on iNat-1M serves as a proxy task to evaluate whether the model learns diverse features.

Since the goal of AdaSSL is to learn diverse features under target uncertainty, we hypothesize that it is more robust to noisy data pairings than vanilla SSL. The ideal setting is one where we have perfect class labels and we can pair up images within the same class (Fig. 6). To control the amount of noise in the pairings, we corrupt a subset of labels such that these corrupted data is paired with another image from the same super class instead (Fig. 7). This simulates real-world label noise, which is rarely random, but instead structured; it is much more likely for a dog to be mislabeled as a wolf than as an avocado. We gradually increase the percentage of data that are corrupted, and investigate how it affects the performance of online linear probes trained on top of the representations.

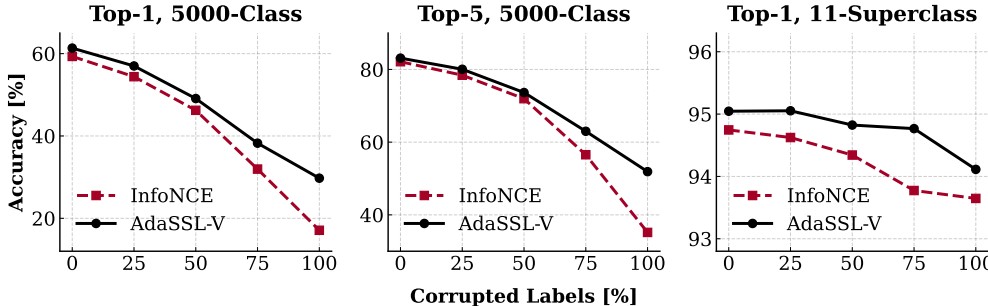

Figure 8: Results on iNat-1M. We gradually increase the percentage of pairings that are corrupted. Performance of online linear probes on the validation set is reported. Left: top-1 accuracy across 5000 classes. Middle: top-5 accuracy across 5000 classes. Right: top-1 accuracy across 11 super-classes. AdaSSL-V's performance decays more gracefully than InfoNCE, showing its robustness to noisy pairings.

Table 5: Ablation of the additional view $\mathbf{x}^{++}$ on CelebA, trained with natural pairs under weak or strong augmentations. Linear $F_1$ scores on representations (encoder output) and embeddings (projector output) are reported. AdaSSL trained without $\mathbf{x}^{++}$ outperforms the baseline, and using $\mathbf{x}^{++}$ improves it further.

| Model | Pairing | WEAK AUGMENTATION | | STRONG AUGMENTATION | |
| | | Repr. | Emb. | Repr. | Emb. |
|---|---|---|---|---|---|
| InfoNCE | Natural | $0.5473 \pm 0.0027$ | $0.3747 \pm 0.0051$ | $0.5784 \pm 0.0008$ | $0.4941 \pm 0.0035$ |
| AdaSSL-S | —"— | $\underline{0.5676} \pm 0.0049$ | $\underline{0.4581} \pm 0.0016$ | $\underline{0.5911} \pm 0.0014$ | $\underline{0.5654} \pm 0.0007$ |
| └without $\mathbf{x}^{++}$ | —"— | $0.5505 \pm 0.0045$ | $0.4466 \pm 0.0026$ | $0.5716 \pm 0.0046$ | $0.5149 \pm 0.0076$ |
| AdaSSL-V | —"— | $\mathbf{0.5784} \pm 0.0025$ | $\mathbf{0.4794} \pm 0.0015$ | $\mathbf{0.6014} \pm 0.0008$ | $\mathbf{0.5706} \pm 0.0034$ |
| └without $\mathbf{x}^{++}$ | —"— | $0.5550 \pm 0.0037$ | $0.4351 \pm 0.0042$ | $0.5869 \pm 0.0011$ | $0.5117 \pm 0.0014$ |

Fig. 8 shows that both InfoNCE and AdaSSL-V suffers decay in fine-grained classification accuracy when we increase the corruption rate. However, AdaSSL-V's accuracy decays more gracefully. We also observe its advantage to be more visible when the corruption rate passes 50%. Interestingly, AdaSSL-V slightly improves InfoNCE even when we do not corrupt any labels. We hypothesize that this could be due to the potential label noise that already exist in the dataset as well as the unavoidable heteroscedastic noise (Proposition 2.1). We observe a slight drop in the superclass classification accuracy and do not observe the gap between AdaSSL and InfoNCE to change much across corruption rates. This is expected because we always pair up data within the same superclass.

### D.3 LEVERAGING AN ADDITIONAL VIEW

Chen et al. (2020a) show that we need strong augmentations to create differences between views in their low-level image statistics. Otherwise, encoding these statistics alone is sufficient to satisfy the SSL objective—a shortcut solution (e.g., standard pairing with weak augmentations in Table 3).

For both AdaSSL-V and AdaSSL-S, the model is expected to learn the factors explaining the differences in the paired views. Allowing $\mathbf{r}$ to encode *any* differences facilitates this shortcut solution. Furthermore, applying pixel-level augmentations may exacerbate this issue by increasing the incentive to learn these discriminative, yet less semantic transformations.

One way to ensure $\mathbf{r}$ learns the desired transformations while remaining invariant to others is to use a surrogate view $\mathbf{x}^{++}$—whose relationship with $\mathbf{x}$ in the underlying content factors $\mathbf{c}$ and $\mathbf{c}^{++}$ mimics that between $\mathbf{c}$ and $\mathbf{c}^{+}$ but differ in low-level statistics—to replace $\mathbf{x}^{+}$. In other words, AdaSSL-V uses $\mathbf{r}$ sampled from $q_\phi(\mathbf{r} \mid f(\mathbf{x}), f(\mathbf{x}^{++}))$ and AdaSSL-S uses $\mathbf{r}$ predicted by $m(f(\mathbf{x}), f(\mathbf{x}^{++}))$. These additional views are usually easy to obtain, e.g., by applying the same random augmentations—that created the differences between $\mathbf{x}$ and $\mathbf{x}^{+}$—to $\mathbf{x}^{+}$. We detail the $\mathbf{x}^{++}$ that we use in each experiment below.

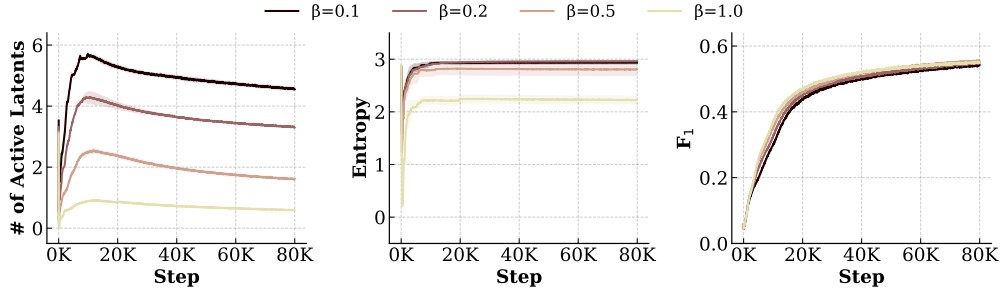

Figure 9: Utilization of $\mathbf{r}$ and performance on CelebA throughout training when AdaSSL-S is applied to InfoNCE. Left: the number of active latent dimensions of $\mathbf{r}$. Middle: the entropy of the probability vector with which the model picks which latent of $\mathbf{r}$ to activate. Right: training $F_1$ score of an online linear probe on representations.

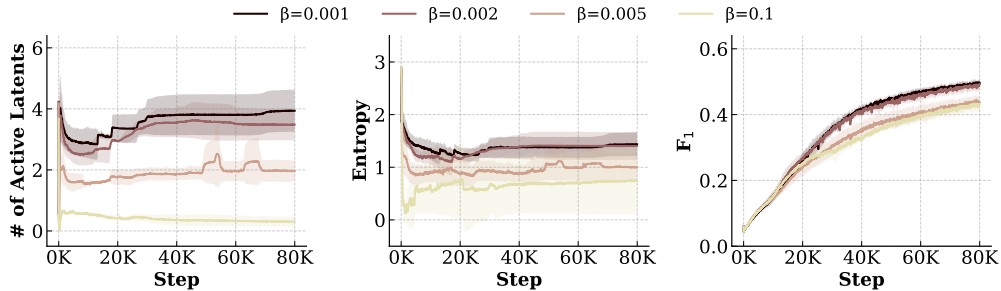

Figure 10: Utilization of $\mathbf{r}$ and performance on CelebA throughout training when AdaSSL-S is applied to BYOL. Left: the number of active latent dimensions of $\mathbf{r}$. Middle: the entropy of the probability vector with which the model picks which latent of $\mathbf{r}$ to activate. Right: training $F_1$ score of an online linear probe on representations.

Notably, this additional view is not always required. When low-level statistics alone are insufficient for instance discrimination, the model is incentivized to learn other features. This is demonstrated in our CRL experiments in §4.3, where we learn to recover *all* data-generating factors from natural pairs without augmentations or $\mathbf{x}^{++}$.

We perform ablation studies of $\mathbf{x}^{++}$ in Table 5 and observe consistent benefits.

### D.4    A DEEPER LOOK INTO ADASSL-S'S BEHAVIOR

Both AdaSSL-V and AdaSSL-S improves InfoNCE on natural pairs, but only AdaSSL-V consistently improves BYOL. We hypothesize that explicitly regularizing the embeddings' information content (as in the denominator of Eq. 2) pressures the model to utilize $\mathbf{r}$ effectively. In contrast, distillation-based methods lack this direct tension. Since AdaSSL-S only regularizes the sparsity, it may require additional regularization on $\mathbf{r}$ that encourages efficient utilization.

In Fig. 9 and Fig. 10, we show statistics of $\mathbf{r}$ throughout training on CelebA with natural pairs and strong augmentations, for AdaSSL-S trained with InfoNCE and BYOL, respectively. For a meaningful comparison, we fix $d_r = 20$ and plot the curves for different values of $\beta$. We show three metrics. First, we show the sparsity of latent usage and plot average number of active dimensions of $\mathbf{r}$ across the batch, i.e., $\mathbb{E}[\|\mathbf{r}\|_0]$. Second, we show the spread of usage across dimensions. To do so, define latent variable $\boldsymbol{\rho}$ in the $d_r$-dimensional probability simplex as the probability with which the model picks each dimension of $\mathbf{r}$ to activate within each batch. We then plot $H(\boldsymbol{\rho})$, the Shannon entropy of $\boldsymbol{\rho}$. Third, we plot the $F_1$ trace on the training set. We expect a successful model to activate a few dimensions of $\mathbf{r}$ each time, but utilize the entire space across different data pairs.

As shown in Fig. 9 and Fig. 10, the strength of regularization controls how many latents are active for both InfoNCE and BYOL. However, BYOL+AdaSSL-S fails to maintain a high spread of

Table 6: Performance of linear probes trained on frozen representations and embeddings on stochastic Moving-MNIST. Evaluation is performed on the online branch of BYOL.

| Model | SETTING A | | | | SETTING B | | | |
| | Representations | | Embeddings | | Representations | | Embeddings | |
| | Acc. [%] | Velocity [$R^2$] | Acc. [%] | Velocity [$R^2$] | Acc. [%] | Velocity [$R^2$] | Acc. [%] | Velocity [$R^2$] |
|---|---|---|---|---|---|---|---|---|
| BYOL | $90.42_{\pm 0.94}$ | $0.8753_{\pm 0.0044}$ | $87.09_{\pm 2.41}$ | $0.1079_{\pm 0.0061}$ | $91.00_{\pm 1.07}$ | $0.8810_{\pm 0.0057}$ | $88.61_{\pm 1.79}$ | $0.1486_{\pm 0.0303}$ |
| BYOL+Future | $88.31_{\pm 1.14}$ | $0.9005_{\pm 0.0063}$ | $78.68_{\pm 0.55}$ | $0.5890_{\pm 0.0242}$ | $88.33_{\pm 1.09}$ | $0.8996_{\pm 0.0059}$ | $78.99_{\pm 0.45}$ | $0.6041_{\pm 0.0186}$ |
| BYOL+GT | $93.09_{\pm 0.24}$ | $0.8814_{\pm 0.0078}$ | $88.95_{\pm 0.56}$ | $-0.0038_{\pm 0.0060}$ | $93.55_{\pm 0.50}$ | $0.8884_{\pm 0.0062}$ | $87.99_{\pm 0.36}$ | $-0.0028_{\pm 0.0045}$ |
| AdaSSL-V$_{\beta=0}$ | $\mathbf{94.18}_{\pm 0.51}$ | $0.8951_{\pm 0.0066}$ | $\underline{90.54}_{\pm 0.54}$ | $0.2867_{\pm 0.0184}$ | $\underline{94.17}_{\pm 0.19}$ | $0.8961_{\pm 0.0028}$ | $\underline{90.34}_{\pm 0.66}$ | $0.2875_{\pm 0.0219}$ |
| AdaSSL-V | $\underline{93.83}_{\pm 0.22}$ | $\mathbf{0.9168}_{\pm 0.0015}$ | $\mathbf{91.28}_{\pm 0.43}$ | $\underline{0.8695}_{\pm 0.0185}$ | $\mathbf{94.31}_{\pm 0.48}$ | $\mathbf{0.9188}_{\pm 0.0006}$ | $\mathbf{92.32}_{\pm 0.73}$ | $\underline{0.8594}_{\pm 0.0035}$ |
| AdaSSL-S | $91.89_{\pm 0.74}$ | $\underline{0.9121}_{\pm 0.0028}$ | $86.00_{\pm 0.33}$ | $\mathbf{0.8901}_{\pm 0.0247}$ | $91.95_{\pm 0.53}$ | $\underline{0.9121}_{\pm 0.0032}$ | $85.53_{\pm 1.90}$ | $\mathbf{0.8750}_{\pm 0.0121}$ |

information across $\mathbf{r}$'s latent dimensions, as reflected by a lower $H(\boldsymbol{\rho})$. We also observe higher instabilities when training AdaSSL-S with BYOL than with InfoNCE. In the latter case, we observe that the model's performance is robust to different $\beta$.

We hypothesize that AdaSSL-V works well with both contrastive and distillation-based SSL because minimizing the KL divergence enforces non-zero variance, which promotes utilization, while using a factorized prior discourages redundancy. Together, the KL regularization encourages learning an efficient representations of $\mathbf{r}$, as we discussed in the disentanglement analysis in Sec. 4.3.

Our findings corroborates the motivation of designing additional regularization terms for sparsity regularization in concurrent work (Garrido et al., 2026), where the authors learn a world model with latent variable regularization on frozen representations.

## D.5 WORLD MODELING

### D.5.1 NUMERICAL RESULTS FOR FIG. 4

We provide full evaluation results on stochastic Moving-MNIST in Table 6. These results further demonstrate AdaSSL's effectiveness in achieving strong performance in both digit recognition and velocity decoding. Our results and ablations in the main text in Fig. 4 uses Setting A because we do not find significant difference between the results.

### D.5.2 FUTURE TRAJECTORY PREDICTION

Given a source (initial) trajectory and its ground-truth latent $\mathbf{r}$ (change in velocity), we predict the embedding of the corresponding target (future) trajectory with the learned predictors of BYOL and AdaSSL-V/S and retrieve the nearest neighbors from a pool of 4096 target trajectory embeddings. To generate this pool, we randomly sample 64 initial trajectories from the validation set and generate 64 future trajectories for each of them. To map the ground-truth $\mathbf{r}$ space to the $\mathbf{r}$ space learned by AdaSSL-S/V, we train a projection (an MLP with one hidden layer of dimension 128) by minimizing the prediction error on the training set using the frozen encoder $f$ and predictor $\eta$.

Table 7 reports mean reciprocal rank (MRR) and hit rate at $k$ (Hit@$k$) which are standard metrics for assessing predictor quality (Kipf et al., 2020; Park et al., 2022; Garrido et al., 2023b; Gupta et al., 2024). AdaSSL consistently outperforms BYOL. Fig. 11 shows the retrieved trajectories and the rank of the ground-truth target in the retrievals. We observe that all methods are able to retrieve the correct digit, while AdaSSL-V and AdaSSL-S predict the direction and velocity of the ground-truth target better than BYOL.

### D.5.3 FUTURE TRAJECTORY SAMPLING

In this section, we evaluate the diversity of the predicted future trajectories on Stochastic Moving-MNIST. We first sample a random source trajectory $\mathbf{x}$ from the validation set and 64 future trajectories from the ground-truth $p(\mathbf{x}^+ \mid \mathbf{x})$. We encode the source and targets using BYOL and AdaSSL-V. Next, we retrieve the nearest neighbor of their predictions and visualize them in Fig. 12. For AdaSSL-V, we use the learned prior to sample multiple $\mathbf{r}$'s and condition the predictor $t$ to predict 10 target embeddings. We then retrieve the nearest neighbors of these targets.

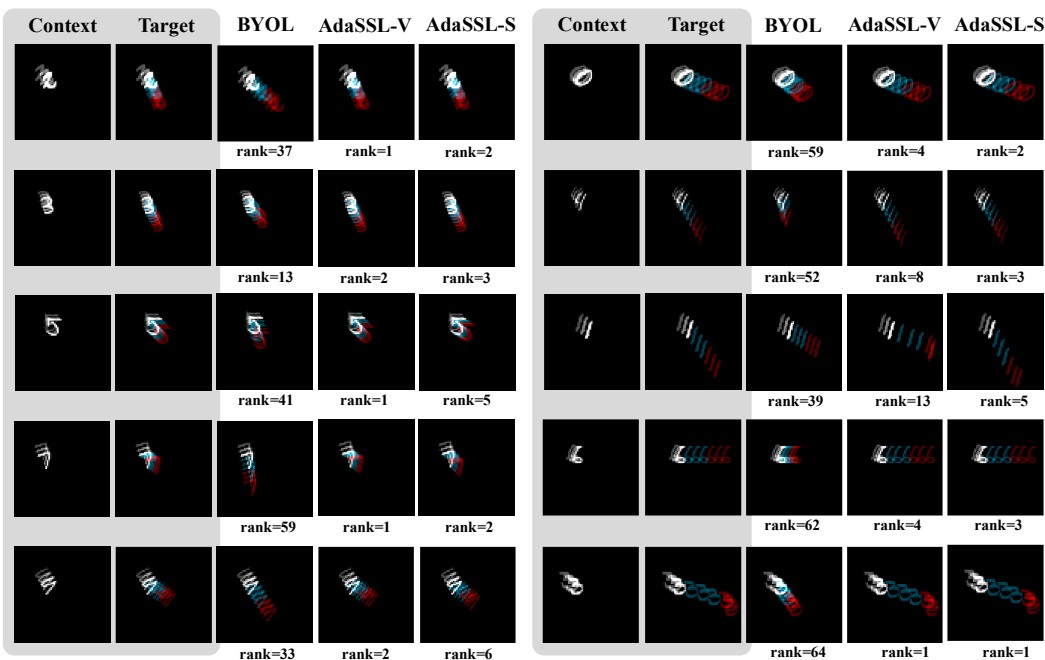

Figure 11: Future prediction visualization. Given a source trajectory and its ground-truth latent **r** (change in velocity), we predict the embedding of the corresponding future trajectory with the learned predictors of BYOL and AdaSSL-V/S and retrieve the three nearest matches from a pool of 4096 future trajectory embeddings. All methods predict the correct digit, while AdaSSL-V and AdaSSL-S predict the directional velocity of the ground-truth target better than BYOL. Retrieval ranks ($\downarrow$) for the ground-truth target are reported below each visualization.

Table 7: Quantitative retrieval results on Stochastic Moving-MNIST. We encode the first three-frame segments of randomly sampled videos and use the learned predictors of BYOL and AdaSSL-S/V to predict one sampled future in the embedding space. We then retrieve the best matched future trajectory from a pool of 4096 video segments (consisting of 64 randomly sampled future trajectories of 64 random initial segments), including the correct segment. We report the mean reciprocol rank (MRR) and hit rate at $k$ (Hit@$k$) for the retrievals.

| Model | MRR ($\uparrow$) | Hit@1 ($\uparrow$) | Hit@5 ($\uparrow$) |
|---|---|---|---|
| BYOL | 0.1758 | 0.1182 | 0.1802 |
| AdaSSL-V | 0.4102 | 0.3071 | 0.5129 |
| AdaSSL-S | **0.5291** | **0.4338** | **0.6287** |

Fig. 12 shows the overlaid ground-truth future trajectories (left) and predictions by BYOL and AdaSSL-V. BYOL produces a single deterministic prediction, whereas the AdaSSL-V predicts diverse plausible futures.

## D.6 RETRIEVAL

In Fig. 13 (left), we perform standard retrieval to accompany our analysis in §4.3. We retrieve the five nearest neighbors of the query image in the embedding space. We observe that both AdaSSL and the baselines are able to retrieve visually similar images. There are still some wrong retrievals in color and spotlight, and rotation is especially hard to learn for all methods.

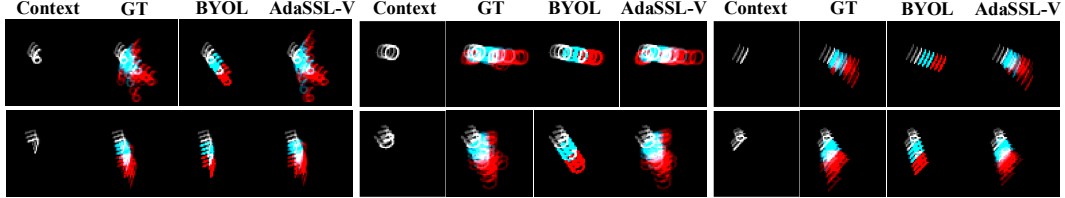

Figure 12: Sampling future trajectories. Given a source trajectory, we visualize the nearest neighbor (via cosine similarity of the embeddings) of the predictions of BYOL (middle) and AdaSSL-V (right). Samples from the ground-truth (GT) transition distribution is shown on the left. AdaSSL-V samples multiple future trajectories from the learned prior while BYOL produces a deterministic prediction.

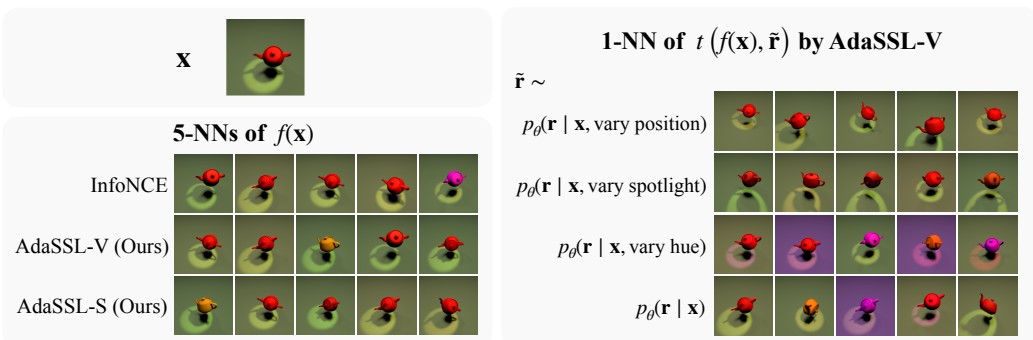

Figure 13: Image retrieval results on 3DIdent. Top left: query image. Bottom left: five nearest neighbors on the embeddings. Right: controllable retrieval by AdaSSL-V.

## E DATA AND IMPLEMENTATION DETAILS

### E.1 IMPLEMENTATION OF $L_0$ PENALTY

For AdaSSL-S, we implement the sparsity regularization following Lachapelle et al. (2022); Brouillard et al. (2020). Specifically, we implement $\mathbf{r}$ as $\mathbf{m} \odot \tilde{\mathbf{r}}$, where $m_i \sim \mathrm{Bern}(\pi_i)$ is a binary mask and $\tilde{\mathbf{r}}$ is the value of the latent. Both the Bernoulli logits and the vector $\tilde{\mathbf{r}}$ are predicted by the MLP $m(f(\mathbf{x}), f(\mathbf{x}^+))$ or $m(f(\mathbf{x}), f(\mathbf{x}^{++}))$.

We encourage sparsity by penalizing the expected number of active gates:

$$\mathbb{E}[\|\mathbf{m}\|_0] = \mathbb{E}\left[\sum_i m_i\right] = \sum_i \mathbb{E}[m_i] = \sum_i \pi_i, \tag{12}$$

which is differentiable w.r.t. $\boldsymbol{\pi}$.

Next, we sample the binary mask $\mathbf{m}$ using the Gumbel-Sigmoid reparameterization of the Bernoulli distribution (Maddison et al., 2017). To enable backpropagation through this discrete decision, we use a straight-through estimator, which backpropagates through the relaxed gates in the backward pass (Bengio et al., 2013b).

### E.2 NUMERICAL EXPERIMENTS IN §4.2 AND §D.1

In the numerical experiments, most of our setup follows prior work (Kügelgen et al., 2021; Zimmermann et al., 2021). We list the similarity functions used by the models in Table 8.

**Complex $p(\mathbf{c}^+ \mid \mathbf{c})$, formally stated.**

$$\boldsymbol{\kappa} \sim \mathcal{N}(0, \Sigma), \quad c_i \mid \boldsymbol{\kappa} \sim \mathcal{N}(\mu(\boldsymbol{\kappa})_i, \sigma(\boldsymbol{\kappa})_i^2), \tag{13}$$

Table 8: Similarity functions used by different models, where $\psi(\cdot) = \frac{f(\cdot)}{\|f(\cdot)\|_2}$ if the model assumes a normalized latent space, in which case InfoNCE and AdaSSL's similarity functions are equivalent to a dot product; otherwise $\psi(\cdot) = f(\cdot)$. The same applies to $\psi_1$ and $\psi_2$, whose subscripts are used to indicate the asymmetry of H-InfoNCE and AdaSSL. Note that in Table 1, H-InfoNCE has $\psi_1 = \psi_2$ because $\mathbb{E}[\mathbf{c}^+ \mid \mathbf{c}] = \mathbf{c}$.

| Model | $s(\mathbf{x}, \mathbf{y})$ |
|---|---|
| InfoNCE | $-\lambda(\psi(\mathbf{x}) - \psi(\mathbf{y}))^\top (\psi(\mathbf{x}) - \psi(\mathbf{y}))$ |
| AnInfoNCE | $-(\psi(\mathbf{x}) - \psi(\mathbf{y}))^\top \mathbf{\Lambda} (\psi(\mathbf{x}) - \psi(\mathbf{y}))$ |
| H-InfoNCE | $-(\psi_1(\mathbf{x}) - \psi_2(\mathbf{y}))^\top \mathbf{\Lambda_x} (\psi_1(\mathbf{x}) - \psi_2(\mathbf{y}))$ |
| AdaSSL | $-\lambda(\psi_1(\mathbf{x}, \hat{\boldsymbol{r}}) - \psi_2(\mathbf{y}))^\top (\psi_1(\mathbf{x}, \hat{\boldsymbol{r}}) - \psi_2(\mathbf{y}))$ |

$$\iota_i \mid \boldsymbol{\kappa} \sim \mathrm{Bern}(\pi(\boldsymbol{\kappa})_i), \quad \mathbf{c}_i^+ \mid \iota_i, \mathbf{c}_i, \boldsymbol{\kappa} \sim \begin{cases} \delta(\mathbf{c}_i^+ = \mathbf{c}_i), & \iota_i = 0 \\ \mathcal{N}\big(\mu(\boldsymbol{\kappa})_i, \sigma(\boldsymbol{\kappa})_i^2\big), & \iota_i = 1 \end{cases}. \tag{14}$$

**Data.** We set $n_c = n_s = 5$ and sample $\Sigma \sim \mathcal{W}^{-1}(n_c + 2, \mathbf{I})$. For anisotropic noise, we sample $\sigma(\mathbf{c})_i^2 \sim \mathrm{InvGamma}(2, 1)$. For heteroscedastic noise, we set $\sigma(\mathbf{c})^2 = \mathrm{softplus}\big(\mathbf{W}_\sigma \mathbf{c} + \mathrm{softplus}^{-1}(1)\big)$. For complex $p(\mathbf{c}^+ \mid \mathbf{c})$, we use $\mu(\boldsymbol{\kappa}) = \mathbf{W}_\mu^\top \boldsymbol{\kappa} + \boldsymbol{b}$, $\sigma(\boldsymbol{\kappa})^2 = \mathrm{softplus}\big(\mathbf{W}_\sigma \boldsymbol{\kappa} + \mathrm{softplus}^{-1}(1)\big)$, and $\pi_i(\boldsymbol{\kappa}) = \mathrm{Sigmoid}\big(\frac{\kappa_i}{\Sigma_{ii}} - 1\big)$. We sample each element of $\mathbf{W}_\mu, \mathbf{W}_\sigma$, and $\boldsymbol{b}$ from $\mathcal{N}(0, 1)$. We parameterize $g_{\mathrm{MLP}}$ as a three-layer MLP with `LeakyReLU` activation (negative slope 0.2) with the same number of units in all layers. We ensure invertibility by using $L^2$-normalized weight matrices that has the lowest condition number among 25 000 uniformly sampled candidates. We use $\mathbf{x}^{++} = g_{\mathrm{MLP}}([\mathbf{c}^+, \mathbf{s}^{++}])$ where $\mathbf{c}^+$ is the same content factor as in $\mathbf{x}^+$ and $\mathbf{s}^{++} \sim \mathcal{N}(0, \mathbf{I})$.

**Architecture.** For the encoder $f$, we use an MLP with four hidden layers of dimensionality $10n$ where $n = n_c + n_s$ is the input dimension. For models that apply $L^2$ normalization to the outputs, we set the output dimensionality to $n + 1$ to accommodate for the missing degree of freedom; otherwise we set it to $n$. For H-InfoNCE$_{\mathrm{Affine}}$, we use an affine layer followed by `softplus` activation to predict $\Lambda_{\mathbf{x}}$. For H-InfoNCE$_{\mathrm{MLP}}$, we use an MLP with three hidden layers of size $10n$ followed by `softplus` activation to predict $\Lambda_{\mathbf{x}}$ and an MLP of the same size to predict $\phi_1(\mathbf{x})$ in Table 4. For AdaSSL, we set $d_r = 5$. We use MLPs with two hidden layers of dimension 64 to parameterize $q_\phi$, $p_\theta$, and $m$ and use a linear $t$ for AdaSSL-V. All MLPs except the encoder use a `BatchNorm` layer followed by `LeakyReLU` with the default negative slope (0.01) after each hidden layer.

**Hyperparameters.** We use the `AdamW` optimizer (Loshchilov & Hutter, 2019) with learning rate $5 \times 10^{-4}$ and weight decay $10^{-4}$ on the parameters except biases. We use a batch size of 2048. For the experiments on complex $p(\mathbf{c}^+ \mid \mathbf{c})$, we apply the loss symmetrically similar to Chen et al. (2020a) because the sampling process of $\mathbf{c}$ and $\mathbf{c}^+$ is symmetric. We train the models for 200 000 steps and observe convergence. For AdaSSL-V, we linearly warmup $\beta$ from 0 to 0.5 for 1000 steps to prevent early KL instabilities. We keep $\beta = 1$ fixed throughout training for AdaSSL-S. For the unimodal $p(\mathbf{c}^+ \mid \mathbf{c})$ experiments, we set $\tau = \mathbb{E}[\sigma_i^2(\mathbf{c})] = 1$ except when the variance is fixed to 0, in which case we set $\tau = 0.1$. For the complex $p(\mathbf{c}^+ \mid \mathbf{c})$ experiments, we set $\tau = 0.1$.

**Evaluation.** We perform evaluation by training a linear regressor on top of the frozen representations on 100 000 unseen data samples and evaluate it on another 100 000 samples.

**Hardware.** Each trial of this experiment required approximately 15-20 hours to run, using eight CPU cores, 4 GB of system memory, and an MIG-partitioned slice of an NVIDIA H100 GPU providing roughly a quarter of the GPU's compute capacity and 20 GB of GPU memory.

### E.3 3DIDENT EXPERIMENTS IN §4.3

**Data.** 3DIdent contains $250\,000$ training images in $\mathcal{D}_{\text{train}}$ and $25\,000$ test images in $\mathcal{D}_{\text{test}}$, which we use for CRL experiments. We sample latent pairs $(\mathbf{z}, \mathbf{z}^+)$ following

$$\mathbf{z} \sim p(\mathbf{z}) \,, \ \tilde{\mathbf{z}} \sim p(\tilde{\mathbf{z}}) \,, \ \iota_i \sim \text{Bern}(0.2) \,, \ \text{z}_i^+ = \text{z}_i \text{ if } \iota_i = 0 \text{ else } \text{z}_i^+ = \tilde{\text{z}}_i \text{ for } i \in [d_z] \,. \quad (15)$$

Since 3DIdent is a finite dataset, after obtaining a latent pair, we find their nearest neighbor in the training set with FAISS (Douze et al., 2024) and use the correspondingly rendered observations as inputs following the original authors (Zimmermann et al., 2021). AdaSSL does not use an additional view in this experiment.

**Data augmentations.** For standard pairs, we use the same set of strong augmentations used for CelebA (§E.4). For natural pairs, we do not perform augmentations. We resize the images to $128 \times 128$ resolution.

**Architecture.** We use a ResNet-18 encoder followed by a two layer MLP projector with hidden size of 128 and output size of 16 as $f$. The hidden layer is followed by `ReLU` activation without `BatchNorm`, and the output layer does not have bias. For AdaSSL, we set $d_r = 16$. We use MLPs with two hidden layers of dimension 128 to parameterize $q_\phi$, $p_\theta$, and $m$. These MLPs use a `BatchNorm` layer followed by `ReLU` activation after each hidden layer. As discussed in §4.3, we ablate the parameterization of $t$ for AdaSSL-V; the MLP parameterization has a hidden layer of dimensionality 128 with `BatchNorm` followed by `ReLU` activations. The VAE-based methods use a ResNet-18 decoder that mirror the encoder.

**Hyperparameters.** We use the `AdamW` optimizer with learning rate $10^{-4}$, weight decay $10^{-5}$ on non-bias parameters, and a batch size of 256. For contrastive learning, we calculate the loss symmetrically following standard practice (Chen et al., 2020a). We train all models for $150\,000$ steps and observe convergence on $\mathcal{D}_{\text{train}}$. All SSL methods use a normalized embedding space, use $\tau = 0.05$, and do not learn $\lambda$ in this experiment. For AdaSSL-V, we perform linear warmup of $\beta$ from 0 to 0.5 for $10\,000$ steps to prevent early KL instabilities. For AdaSSL-S, we fix $\beta = 0.5$. For AdaGVAE, we search within the authors' recommended set of $\beta$'s, $[1, 2, 4, 8, 16]$, but find $\beta = 100$ to give the best disentanglement.

**Evaluation.** We perform evaluation on $\mathcal{D}_{\text{test}}$ with the frozen embeddings and ground-truth latent factors with linear regression and the DCI disentanglement score. We normalize the embeddings for the SSL based models such that they align with the training objective, similar to Zimmermann et al. (2021). We use the posterior mean as the embeddings for VAE-based models and do not normalize them. For the DCI disentanglement score, we use the weights of Lasso regressors as the relative importance matrix.

**Hardware.** Each trial of this experiment required approximately 15-20 hours to run, using eight CPU cores, 32 GB of system memory, and an MIG-partitioned slice of an NVIDIA H100 GPU providing roughly three-eighths of the GPU's compute capacity and 40 GB of GPU memory.

### E.4 CELEBA EXPERIMENTS IN §4.4

**Data.** We split the CelebA dataset into $\mathcal{D}_{\text{train}}$, $\mathcal{D}_{\text{val}}$, and $\mathcal{D}_{\text{test}}$ following an 8-1-1 ratio; this gives us $161\,908$ training images, $20\,346$ images in the validation set and $20\,345$ images in the test set. To create a natural distribution shift, we sample celebrity identity such that the people in $\mathcal{D}_{\text{train}}$ does not appear in $\mathcal{D}_{\text{val}} \cup \mathcal{D}_{\text{test}}$. This gives us 8142 celebrities in $\mathcal{D}_{\text{train}}$ and 2035 celebrities in $\mathcal{D}_{\text{val}} \cup \mathcal{D}_{\text{test}}$. To construct a structured positive pair, we randomly sample two images of the same person. This results in $1\,850\,918$ possible positive pairs. Data pairs examples are visualized in Fig. 14 and the distribution of the number of differed attributes between pairs are shown in Fig. 15, confirming that attributes differ sparsely between positive pairs. During training, we augment the sampled pair using data augmentations and obtain $\mathbf{x}$ and $\mathbf{x}^+$. We use another augmented view of $\mathbf{x}^+$ as $\mathbf{x}^{++}$. This is helpful because our goal is not to learn the low-level style factors, but instead the semantic content factors that differ structurally between $\mathbf{x}^+$ and $\mathbf{x}$. The standard pairing process still use augmented versions of the same image as positive pairs.

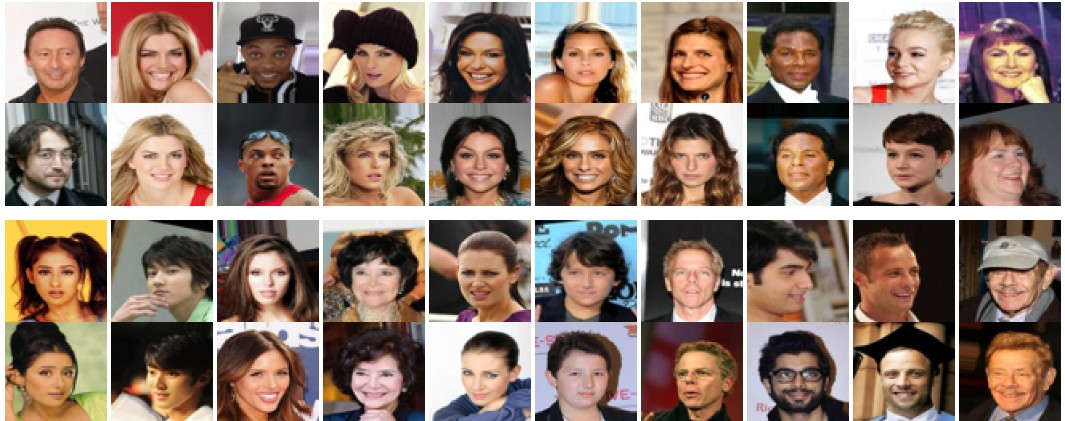

Figure 14: Visualization of images paired by identity from the CelebA dataset.

**Data augmentations.** We investigate the effect of both strong and weak augmentations. For strong augmentations, we apply the standard set of augmentations used in SSL studies (Chen et al., 2020a; Grill et al., 2020). We use `RandomHorizontalFlip` with 0.5 probability, then `RandomResizedCrop` with crops of size within $[8\%, 100\%]$ of the original image and aspect ratio within $[0.75, 1.33]$, which are then resized to $64 \times 64$. Next, with probability 0.8, we randomly apply `ColorJitter` where the brightness, contrast, saturation and hue of the image are shifted by a uniformly random offset. We use parameters $0.4, 0.4, 0.2, 0.1$, respectively. Finally, we apply `RandomGrayScale` with probability 0.2, `GaussianBlur` with probability 0.5, and `Solarization` with probability 0.2. For weak augmentations, we only apply `RandomHorizontalFlip`

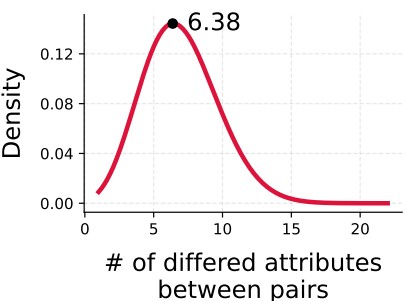

Figure 15: Distribution of the number of differed attributes between pairs of images of the same identity.

with probability of 0.5 and `RandomResizedCrop` with crops of size within $[80\%, 100\%]$ of the original image and aspect ratio within $[0.9, 1.1]$. Notice that this cropping operation is significantly weaker than the one used for strong augmentations.

**Architecture.** We use a ResNet-18 encoder (He et al., 2016) followed by a two layer MLP projector with hidden size of 1024 and output size of 128 as $f$. For InfoNCE, the hidden layer is followed by `ReLU` activation without `BatchNorm`, and the output layer does not have bias following Chen et al. (2020a). For BYOL, the hidden layer uses `BatchNorm` followed by `ReLU` activation, and the output layer does not have bias following Grill et al. (2020). BYOL's predictor design is the same as its projector. For AdaSSL, we set $d_r = 20$. We use MLPs with one hidden layer of dimension 1024 to parameterize $q_\phi$, $p_\theta$, and $m$. We use an MLP with one hidden layer of dimension 512 to parameterize $t$ for AdaSSL-V; this MLP does not have a bias term in the output layer, similar to the predictor in BYOL. These MLPs use a `BatchNorm` layer followed by `ReLU` activation after each hidden layer.

**Hyperparameters.** We use the `AdamW` optimizer with learning rate $2 \times 10^{-4}$ and weight decay $10^{-4}$ on the parameters except biases. We use a batch size of 512. We calculate the SSL loss symmetrically following standard practice (Chen et al., 2020a). We train the models for $80\,000$ steps and observe convergence on $\mathcal{D}_{\text{val}}$. For contrastive learning, all models use a normalized embedding space and use $\tau = 0.1$. The following are the hyperparameters of different methods used in combination with InfoNCE. For AdaSSL-V, we perform linear warmup of $\beta$ from 0 to 0.1 for $10\,000$ steps to prevent early KL instabilities. For AdaSSL-S, we fix $\beta = 0.5$. For LieSSL, we search for the best-performing coefficient of the last regularization term from $[0.01, 0.1, 1, 10]$ because we do not assume access to transformation labels, and choose the best-performing number of basis functions

from [1, 10, 20, 40]. For BYOL, we anneal the EMA momentum from 0.996 to 1 following a cosine schedule. With BYOL, AdaSSL-V uses a linear warmup of $\beta$ from 0 to 0.005 for $10\,000$ steps for weak augmentations and to 0.001 for strong augmentations. For AdaSSL-S, we fix $\beta = 0.002$ for weak augmentations and $\beta = 0.0005$ for strong augmentations.

**Evaluation.** Following standard practice, we train a linear classifier with the `BinaryCrossEntropy` loss for each attribute on top of the frozen representations and embeddings on $\mathcal{D}_{\text{train}}$ until convergence and evaluate it on $\mathcal{D}_{\text{test}}$. We use the $F_1$ score of the minority class as the evaluation metric because the attributes are highly imbalanced. To do that, we compute the $F_1$ score for each attribute then report the mean score over attributes.

**Hardware.** Each trial of this experiment required approximately 15-20 hours to run, using 12 CPU cores, 24 GB of system memory, and an NVIDIA L40S GPU with 48 GB of GPU memory.

### E.5 Video Experiments in §4.4

**Data.** We construct a custom dataset similar to Moving-MNIST (Srivastava et al., 2015; Drozdov et al., 2024), where nine-frame videos are generated stochastically on the fly from sample images in MNIST. For a given image, we first create a black $64 \times 64$ canvas. Afterwards, we resize the original $28 \times 28$ image to $16 \times 16$ and place it on the canvas after uniformly sampling its initial center coordinates from $[8, 16]$. In frames 1-3, the digit moves from this center based on a velocity in the horizontal direction, denoted by $v_{x,1:3}$, and in the vertical direction, denoted by $v_{y,1:3}$. We sample these initial velocities uniformly from $[0, v_0]$ where $v_0 = 3$. Then, with an equal probability, we sample one direction and change its velocity by adding a Gaussian noise proportional to the initial velocity (i.e., heteroscedastic):

$$\iota \sim \text{Bern}(0.5)\,, \quad \begin{cases} v_{x,4:9} \sim \mathcal{N}\left(v_{x,1:3}, \frac{2}{3}v_{x,1:3}\right), v_{y,4:9} = v_{y,1:3}\,, & \iota = 0 \\ v_{y,4:9} \sim \mathcal{N}\left(v_{y,1:3}, \frac{2}{3}v_{y,1:3}\right), v_{x,4:9} = v_{x,1:3}\,, & \iota = 1 \end{cases}. \tag{16}$$

This makes the new velocity in frame 4-9 within $(-v_0, 3v_0)$ with high probability. Generated video samples are shown in Figure 16. We refer to this as *Setting A*.

In *Setting B*, we let $\iota$ depend on the digit input. Concretely, we use equally spaced bins between 0.1 and 0.9 for the ten digits:

$$\iota_k \sim \text{Bern}(p_k)\,, \quad \text{where} \quad p_k = 0.1 + k \cdot \frac{0.9 - 0.1}{10 - 1}\,, \quad k = 0, \ldots, 9\,. \tag{17}$$

This means the distribution of the direction of acceleration varies for different digits.

We partition each sampled video into three-frame segments and use them as $\mathbf{x}$, $\mathbf{x}^+$, and $\mathbf{x}^{++}$ (§D.3). The model predicts $f(\mathbf{x}^+)$ from $f(\mathbf{x})$ (and optionally $f(\mathbf{x}^{++})$ by AdaSSL and BYOL+Future). The goal is to capture both the digit class and the velocity in the three-frame video representations. We partition the $60\,000$ MNIST images into $50\,000$ training images and $10\,000$ validation images and use each set for generating training and validation videos on the fly. Note that we always sample the velocities online, and the model observes different videos in every epoch.

**Architecture.** The encoder $f$ consists of a 3D convolutional encoder, followed by an MLP projector. The 3D convolutional encoder consists of five convolutional layers with $[32, 64, 128, 128, 256]$ channels with `BatchNorm` and `ReLU` activations after each layer. The first two and the last layer have spatial-only kernels of dimensions $[1, 3, 3]$ and the third and fourth layers have temporal convolutions with kernels of dimensions $[3, 1, 1]$. The encoder outputs are average-pooled on the spatial dimensions and then flattened across the temporal dimension resulting in a 768-dimensional representation. The representations are passed to an MLP projector with two hidden layers of size 1024, each followed by `BatchNorm` and `ReLU` activations. The output embeddings have a dimensionality of 128, and are batch-normalized. The projector is followed by an MLP predictor $h$ with two hidden layers of dimensionality 1024 with `BatchNorm` and `ReLU` activations after each hidden layer. The predictor output does not use `BatchNorm` or `ReLU`. For AdaSSL-V, we use a two-dimensional $\mathbf{r}$, which is concatenated to $f(\mathbf{x})$ as the predictor input. We use MLPs with one hidden layer of dimensionality 1024 to parameterize $q_\phi$, $p_\theta$, and $m$. These MLPs use a `BatchNorm` layer

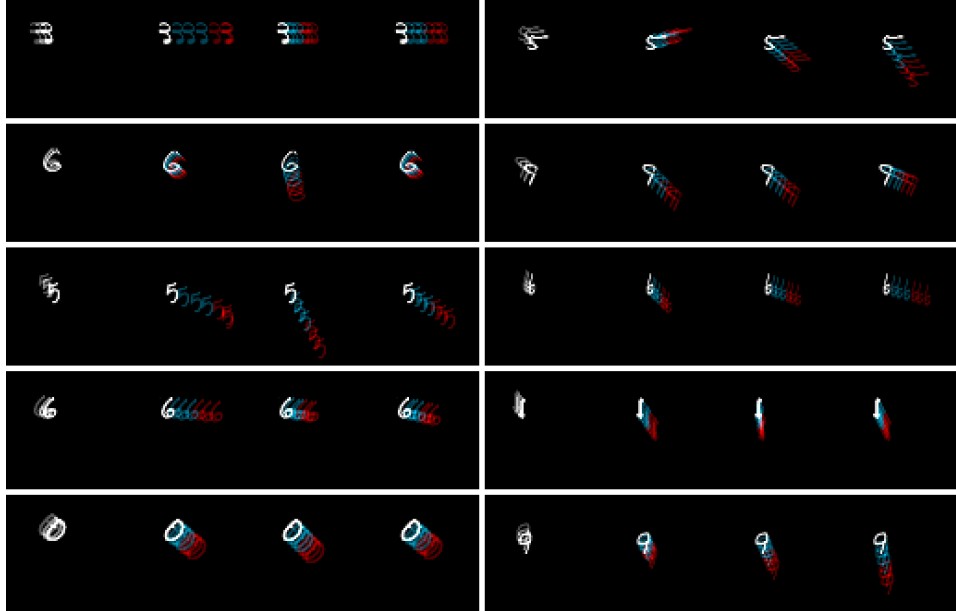

Figure 16: Random samples (nine-frame video sequences) from the stochastic Moving-MNIST dataset. For each example, the first three frames (context) are shown on the left. Then, three different future trajectories of the next six frames (targets) are randomly sampled according to Eq. 16 and visualized to the right of the initial three-frame segment. The third frame is overlaid on all canvases for reference. The motion uncertainty arises from random velocity changes along spatial directions.

followed by `ReLU` activation after each hidden layer. For BYOL+Future, we concatenate the projector embeddings $f(\mathbf{x})$ and $f(\mathbf{x}^{++})$ and use it as the predictor input. BYOL+GT predicts $f(\mathbf{x}^{+})$ from $f(\mathbf{x})$ and $r^{\star}$, the ground-truth difference between the velocities of $\mathbf{x}$ and $\mathbf{x}^{+}$. We experiment with concatenating $r^{\star}$ directly with $f(\mathbf{x})$ or passing it through a learnable linear embedding before concatenation, and find that using an embedding layer slightly improves performance.

**Hyperparameters.**    For all methods, we train the model for $75\,000$ steps with the `AdamW` optimizer using a batch size of $128$. We use an initial learning rate of $10^{-4}$ and decay it following a cosine schedule, following Grill et al. (2020). We use a constant weight decay of $10^{-4}$. For the EMA momentum, we use a constant decay rate of $0.996$. In BYOL+GT, we learn an affine projection to create an embedding for $\mathbf{r}^{\star}$ of dimensionality $32$. For all AdaSSL models, we use a constant regularization coefficient $\beta$, and in our default setting, $d_r = 2$ and $\beta = 0.001$.

**Evaluation.**    To perform evaluation, we train linear probes with `CrossEntropy` (for digit classification) and `MSE` (for velocity regression) losses on top of the frozen video representations and embeddings of the online branch on $\mathcal{D}_{\mathrm{train}}$ until convergence. We then report the digit prediction accuracy and velocity decoding $R^2$ scores on a fixed video test set generated from the $10\,000$ test images of MNIST.

**Hardware.**    Each trial of this experiment required approximately 6-8 hours to run, using six CPU cores, 32 GB of system memory, and an NVIDIA H100 GPU with 80 GB of GPU memory.

### E.6    ɪNAT-1M EXPERIMENTS IN §D.2

**Data.**    iNat2021 (Van Horn et al., 2021) is a large-scale image dataset collected and annotated by community scientists that contains fine-grained (class) and coarse-grained (superclass) species labels. We create a balanced subset from iNat2021 of about half of its size. We first randomly sample 5000 classes that have at least 250 images from iNat2021's training set. We then randomly pick 200 images as $\mathcal{D}_{\mathrm{train}}$ and 50 as $\mathcal{D}_{\mathrm{val}}$. This results in a one-million-image training set and a validation set of 250 000 images. We name this subset of iNat2021 as iNat-1M. During training,

we investigate the model's behavior under different natural pairings. We start with a setting where we pair up each image with an image from the same class, randomly sampled at each step. Since we evaluate the model's performance on fine-grained classes, this setting is similar to supervised contrastive learning (Khosla et al., 2020) because the model is encouraged to cluster images within the same class together. For the rest of the settings, we randomly corrupt the pairing process of each image with probability $\omega \in \{0.25, 0.5, 0.75, 1\}$. Our corruption changes the pairing method for selected images to pairing within the same superclass. We expect AdaSSL to adapt to different corruption ratios better than baselines because it's less restricted on the uncertainty in features that could change under a coarse/corrupted pairing. We augment paired data using data augmentations and obtain $\mathbf{x}$ and $\mathbf{x}^+$ for all methods. We use another augmented view of $\mathbf{x}^+$ as $\mathbf{x}^{++}$.

**Data augmentations.** We resize the images to $224 \times 224$ resolution and use the same set of strong augmentations used for CelebA (§E.4).

**Architecture.** We use a ResNet-50 encoder (He et al., 2016) followed by a two layer MLP projector with hidden size of 1024 and output size of 128 as $f$. The hidden layer is followed by `ReLU` activation without `BatchNorm`, and the output layer does not have bias following Chen et al. (2020a). For AdaSSL, we set $d_r = 8$. We use MLPs with one hidden layer of dimension 1024 to parameterize $q_\phi$, $p_\theta$, and $t$. $t$ does not have a bias term in the output layer. These MLPs use a `BatchNorm` layer followed by `ReLU` activation after each hidden layer.

**Hyperparameters.** We use the `AdamW` optimizer with learning rate $2 \times 10^{-4}$ and weight decay $10^{-4}$ on the parameters except biases. We calculate the SSL loss symmetrically. We use a batch size of 256 and train the models for $200\,000$ steps. For contrastive learning, all models use a normalized embedding space and use $\tau = 0.1$. For AdaSSL-V, we perform linear warmup of $\beta$ from 0 to $\beta_{\text{final}}$ for $10\,000$ steps. We search $\beta_{\text{final}}$ from $\{0.2, 0.4, 0.6, 0.8\}$ and find $\beta_{\text{final}} = 0.8$ performs the best for all settings except on the noisiest setting when $\omega = 1$, where we need weaker regularization, $\beta_{\text{final}} = 0.4$, to accommodate the higher uncertainty.

**Evaluation.** We train two online linear probes with `CrossEntropy` loss on top of the representations, one for predicting the superclass and one for the fine-grained species classes. The gradient of these losses do not propagate back to the encoder. We then report their performance on $\mathcal{D}_{\text{val}}$.

**Hardware.** Each trial of this experiment required approximately 40 hours to run, using 12 CPU cores, 60 GB of system memory, and an NVIDIA H100 GPU with 80 GB of GPU memory.

