# OpenReview forum: "Self-Supervised Learning from Structural Invariance"
_ICLR.cc/2026/Conference — ICLR 2026 Poster_

### Official Review · Reviewer_y8NR · 2025-10-29

**Soundness:** 4
**Presentation:** 4
**Contribution:** 3
**Rating:** 8
**Confidence:** 4

**Summary:**

The paper studies the Self-Supervised Learning problems. It argues that existing methods fail to model the heteroscedasticity in $p(z^+|z)$  for positive pairs. The paper proposes AdaSSL by introducing a latent variable $r$ that captures the stochastic transformation from one view to another. By jointly learning the encoder f(x), the latent transformation r, and an edit function that reconstructs the target embedding, the method adapts to the heteroscedasticity in the above conditional. The paper performs extensive experiments on synthetic and real datasets and shows improved disentanglement and generalization compared to InfoNCE and BYOL variants.

**Strengths:**

- I find the paper interesting and enjoyable to read.
- Learning from pairwise images and the heteroscedasticity in $p(z^+|z)$ is a promising and under-explored area.
- Strong motivations backed by theoretical and empirical analysis.

**Weaknesses:**

- While proposition 2.1 shows heteroscedasticity exists, and 2.2 explains how existing methods fail to account for it. It remains unclear to me (at least intuitively) how modeling the complex conditional $p(z^+|z)$ contributes towards the ultimate SSL objective, i.e., generalization on a wide range of downstream tasks that predict a subset of factors in z.
- It is even harder for me to understand how the proposed approach should be better in downstream tasks compared to baselines with similar motivation (e.g., H-InfoNCE).
- As acknowledged in Section 3.2, the proposed approach is only theoretically justified for contrastive SSL. This introduces a gap due to the popularity and performance of non-contrastive SSL.
- $q_\phi,p_\theta$ are both modeled as factorized Gaussians. This is slightly against the idea that the conditional can be quite complex, as it is up to the edit function $t$ to model the complexity, which can complex model design and reduces learning efficiency.

**Questions:**

- Why is the proposed approach better than H-InfoNCE? How does it encourage disentanglement?
- How learning an efficient representation of r (line 397) leads to a more disentangled feature f(x) in Table 4?
- Why is another view $x^{++}$ introduced in CelebA experiment? What if only using $x$ and $x^+$?

---

> ### Author Response · Authors · 2025-11-21
> **part 1**
>
> Thank you for the positive feedback on our motivation and justification; we are glad that you found our paper enjoyable to read! Please find our response below.
>
> > While proposition 2.1 shows heteroscedasticity exists, and 2.2 explains how existing methods fail to account for it. It remains unclear to me (at least intuitively) how modeling the complex conditional $p(z^+ \mid z)$ contributes towards the ultimate SSL objective, i.e., generalization on a wide range of downstream tasks that predict a subset of factors in z.
>
> **Representation learning and downstream tasks.**
> Please see our joint response above. Please also refer to our discussion with Reviewer TJcB on generalization.
>
> **How modeling $p(z^+ \mid z)$ helps learning more diverse features.**
> In Sec. 2.1, we discuss prior work that has shown that SSL provably recovers the data generating factors under simple conditional distributions [Zim+21, Küg+21, Yao+24, Rus+25]. Intuitively,  representation learning with SSL encourages the diversity of representations, while making sure the encoded features are **shared (invariant) between positive pairs**. Contrastive learning achieves this by explicitly pushing apart representations of different data points, whereas non-contrastive learning does so through architectural tricks (e.g., [Zhu+23] shows that these designs increase the effective rank of the representations).
>
> Our work extends SSL to tackle non-trivial conditional distributions. We continue to leverage the SSL objective to learn diverse features, but by allowing **structured differences** between the latents of positive pairs (i.e., $z$ and $z^+$) through $r$, we “reduce the pressure” for the SSL loss to discard features that are different.
>
>
>
> > It is even harder for me to understand how the proposed approach should be better in downstream tasks compared to baselines with similar motivation (e.g., H-InfoNCE). Why is the proposed approach better than H-InfoNCE?
>
> Note that all InfoNCE, AnInfoNCE, and H-InfoNCE still assume $p(z^+ \mid z)$ to be a unimodal distribution, and simply use the dot product of the embeddings, albeit directionally weighted, to judge whether a data pair comes from the joint p(x, x^+). In comparison, AdaSSL puts less assumptions on $p(z^+ \mid z)$ by using a latent variable model: $p(z^+ \mid z) = \int p(r \mid z)p(z^+ \mid z, r) dr$. Thus, it outperforms H-InfoNCE in our results.
>
> We have added a sentence to clarify this distinction when we introduce H-InfoNCE.
>
> Please also kindly see Table 5 in the paper for the comparison of similarity functions used by different methods.
>
> > As acknowledged in Section 3.2, the proposed approach is only theoretically justified for contrastive SSL. This introduces a gap due to the popularity and performance of non-contrastive SSL.
>
> Please see the joint response above.
>
> > $q_\phi$ and $p_\theta$ are both modeled as factorized Gaussians. This is slightly against the idea that the conditional can be quite complex, as it is up to the edit function $t$ to model the complexity, which can complicate model design and reduce learning efficiency.
>
> Thank you for this question. We agree with your intuition. We hypothesize that there are two layers of complexity that $t$ needs to model.
>
> First, how a cause changes the observation, i.e., $p(z^+ \mid z, r)$ depends on the data pairs. It is crucial to note that, unlike in VAEs, $t$ models the effect of $r$ in the latent space $Z$ rather than the observation space $X$. We hypothesize that this is why a complex editing function is often unnecessary. In our experiments, $r$ is intuitively closely related to the factors we aim to capture in the embeddings. For example, we encode facial attributes and $r$ could learn whether the attributes are different between $z$ and $z^+$; in this case, $t(f(x), r)$ only needs to “flip” the differing binary attribute to get an accurate estimate of $f(x^+)$.
>
> Second, the choice of distribution family for $p$ and $q$ is a design choice. We use factorized Gaussians because it is easy to work with (e.g., tractable KL term) and standard in variational inference. If more prior knowledge is available about how the pairs are constructed, alternative parameterizations could also be considered.
>
> In our experiments, the largest editing function is **an MLP with one hidden layer**, including for the newly added iNaturalist-1M results in the joint response above. That said, we acknowledge that in more complex settings such as long-horizon world modeling, latent causes can induce non-trivial changes in observations, which may require a larger editing function.
>
> We have added this limitation to Appendix E.

---

> > ### Author Response · Authors · 2025-11-21
> > **part 2**
> >
> > > How learning an efficient representation of r (line 397) leads to a more disentangled feature f(x) in Table 4?
> >
> > **Disentanglement in $r$.**
> > Our hypothesis is based on empirical observations provided in $\beta$-VAE [Hig+17], which show that the KL penalty encourages the learned latent variables to be disentangled. Assuming that the data is generated from independent factors, the intuition is that if two latent dimensions both encode the same generative factor, their means will vary together across data points. This creates statistical dependence in the distribution of latent codes across the dataset (i.e., the aggregated posterior $q(z)$), which deviates from the fully factorized prior $p(z)$ and therefore incurs a higher KL penalty.
> >
> > In our case, we assume $p(r \mid x)$ to have independent dimensions. Similarly as above, the KL term is minimized when the additional information that comes from $x^+$ is encoded in a component-wise independent way. We use the term "efficient" because the KL term regularizes the capacity of the latent space, and minimizing it drives the model to find an efficient solution to represent $r$.
> >
> > **Disentanglement in $f(x)$.**
> > Whether $f(x)$ is disentangled depends on the complexity of the editing function. In the simplest case, an additive editing function, i.e., $t(f(x), r) = f(x) + r$, would make the dimensions of $f(x)$ to align with $r$. This leads to disentanglement in $f(x)$, as shown empirically by Table 3.
> >
> > We have updated the manuscript to make it clearer how we arrive at this hypothesis.
> >
> > > Why is another view $x^{++}$ introduced in the CelebA experiment? What if only using $x$ and $x^+$?
> >
> > The reason we use another view is because we apply augmentations to the images. Recall that in our DGP we have $x = g(z)$ and $x^+ = g(z^+)$. Augmentations perturbs $x$ and $x^+$, changing the data pair into $x = g(z) + \epsilon_1$ and $x^+ = g(z^+) + \epsilon_2$. Therefore, if we do not remove these noise, $r$ may learn to encode the difference of these low-level features, i.e., $\epsilon_2 - \epsilon_1$, rather than prioritizing learning the meaningful data generating factors $z$ that we care about. Predicting $r$ from $x$ and $x^{++} = g(z^+) + \epsilon_3$ prevents this; it helps the representations to still be invariant to low-level noise and prevents this invariance to be a confounder in our analysis.
> >
> > Note that we have shown in the CRL experiments (Sec. 4.3 ) that it is unnecessary to use $x^{++}$ when we don’t use augmentations. Additionally, per your suggestion, we performed the same experiment on CelebA for AdaSSL, without using $x^{++}$. The results are shown below in Table D. We find that the performance slightly degrades, but still outperforms the baseline in most cases.
> >
> > We will add this analysis to the manuscript.
> >
> > **Table E. Ablation of $\mathbf{x}^{++}$ on CelebA, trained with natural pairs under weak or strong augmentations. Test $F_1$ scores of linear probes on the representations or embeddings are reported.**
> >
> > | Method | Repr. (weak) | Emb. (weak) | Repr. (heavy) | Emb. (heavy)  |
> > | :---  | :---: | :---: | :---: | :---: |
> > | InfoNCE | 0.5473 $\pm$ 0.0027 | 0.3747 $\pm$ 0.0051 | 0.5784 $\pm$ 0.0008 | 0.4941 $\pm$ 0.0035 |
> > | AdaSSL-V | 0.5550 $\pm$ 0.0037 | 0.4351 $\pm$ 0.0042 | 0.5869 $\pm$ 0.0011 | 0.5117 $\pm$ 0.0014 |
> > | AdaSSL-V (w. $\mathbf{x}^{++}$) | **0.5784** $\pm$ 0.0025 | **0.4794** $\pm$ 0.0015 | **0.6014** $\pm$ 0.0008 | **0.5706** $\pm$ 0.0034 |
> > | AdaSSL-S | 0.5505 $\pm$ 0.0045 | 0.4466 $\pm$ 0.0026 | 0.5716 $\pm$  0.0046 | 0.5149 $\pm$ 0.0076 |
> > | AdaSSL-S (w. $\mathbf{x}^{++}$) | **0.5676** $\pm$ 0.0049 | **0.4581** $\pm$ 0.0016 | **0.5911** $\pm$ 0.0014 | **0.5654** $\pm$ 0.0007 |
> >
> >
> >
> > ---
> >
> > We thank the reviewer again for the helpful discussion points. We hope our response has addressed the raised concerns, and we are happy to clarify any remaining questions.
> >
> > ---
> >
> > # Refs.
> >
> > [Küg+21] Self-Supervised Learning with Data Augmentations Provably Isolates Content from Style. von Kügelgen et al., NeurIPS 2021.\
> > [Yao+24] Multi-view causal representation learning with partial observability. Yao et al., ICLR 2024.\
> > [Zim+21] Contrastive Learning Inverts the Data Generating Process. Zimmermann et al., ICML 2021.\
> > [Rus+25] InfoNCE: Identifying the Gap Between Theory and Practice. Rusak et al., AISTATS 2025.\
> > [Hig+17] $\beta$-VAE: Learning basic visual concepts with a constrained variational framework. Higgins et al., ICLR 2017.

---

> > > ### Author Response · Authors · 2025-11-27
> > > **Quick follow-up**
> > >
> > > Dear Reviewer y8NR,
> > >
> > > As the rebuttal period is nearing its end, we wanted to briefly follow up.
> > >
> > > We have responded to all of your comments above and updated the paper accordingly. We believe these revisions have strengthened the work, and we would be happy to clarify anything further if you have remaining questions.
> > >
> > > Thank you again for your time and effort in reviewing our paper!
> > >
> > > Best,\
> > > Authors

---

### Official Review · Reviewer_NVc3 · 2025-10-30

**Soundness:** 3
**Presentation:** 3
**Contribution:** 2
**Rating:** 4
**Confidence:** 4

**Summary:**

This paper addresses the problem of self-supervised representation learning when the two views of the data used to learn the representation are so-called {\em natural pairs}  instead of handcrafted augmentations, such that the two views are generated from (unknown) latent factors with the dependency between the latent variable encoded by some unknown conditional probability distribution.

**Strengths:**

SSL methods are widely used, and their theoretical study is welcome. The dea of modeling the dependency of the views on some latent variables is interesting.

**Weaknesses:**

Although the topic of the paper is interesting, the presentation is hard to follow without well identified objectives and experiments mostly limited to toy examples.

The presentation should be clarified. For example:
- Section 2 defines the data generation process in terms of latent variables z and z+, but in Section 3 these disappear to be replaced by a latent variable r presumably there to parameterize the predictor, simillar to predictive SSL methods.
- I did not understand the difference between the AdaSSL loss and that typically used in predictive SSL, in part because the dependency of the model on the latent variable r is never defined before giving Eqs. (4) and (5) so I didn't understand how the terms in these equations were computed.
-psi_1 and psi_2  are used in Eq. (8) before they are defined in Eq. (9).
-The function t, which was an arbitraty MLP until then, is defined explicitly in Eq. (11) as a modular editing function.

I could not find any justification as to why it should be possible to recover the latent variables since they are never used, as far as I know, in the actual loss of the AdaSSL variants. This is problematic since the experimental evaluation is for the most part dedicated to this recovery.

Remark: although it is frequently used in non-contrastiva approches to SSL, the EMA formulation in Eq. (3) is, as far as I know, ill defined since the exponential moving average is normally taken over the parameters defining psi over time.

**Questions:**

I understand how, from their probabilistic definition in terms of latent factors, natural data pairs may be different from the "augmented" pairs typically used in SSL. From an intuitive point of view, however, I do not really see how nearby video frames are qualitatively different from image crops, say. Both can be seen as crops, temporal or spatial, of the data. I would appreciate that the authors comment on this point.

Please explain the significance of Prop. 2.1.

As noted by the authors, AdaSSL-V is only justified for contrastive SSL, but it is used for non-contrastive SSL as well. Could you please justifiy this?

---

> ### Author Response · Authors · 2025-11-21
> **part 1**
>
> We thank the reviewer for finding our paper interesting and for the helpful feedback on clarity. We address your concerns below.
>
> > Although the topic of the paper is interesting, the presentation is hard to follow without well identified objectives
>
> Before addressing your concrete feedback on presentation, we’d like to clarify the paper structure here.
>
> **Main objective.**
> We tackle the standard representation problem of learning a diverse representation specifically under natural pairs, as motivated in our intro. **We defined this objective formally in Sec. 2.1.** We have added more intuition to the revised manuscript. Please also see the joint response above for how this relates to downstream tasks.
>
> **Paper structure.**
> We begin by hypothesizing that existing SSL methods are not designed for natural pairs because they assume a simple conditional distribution $p(z^+ \mid z)$ in Sec. 2.2 & 2.3. We validate this hypothesis in controlled numerical experiments in Sec. 4.2.
> Building on these observations, we introduce H-InfoNCE and AdaSSL, two objectives designed to to handle the richer conditional relationships induced by natural pairs. To demonstrate their effectiveness, we first evaluate on 3DIdent, **one of the largest datasets with known ground-truth data generating factors**, and show that our methods recover the latent factors more precisely than existing baselines. Finally, on image and video datasets where ground-truth  data-generating factors are unavailable, we use proxy tasks such as fine-grained classification and velocity decoding to assess whether our representations recover diverse features. Across these settings, our methods consistently improve over standard SSL.
>
> > experiments mostly limited to toy examples
>
> Our goal in this work is to address a fundamental issue in latent SSL objectives. For this reason, we evaluated our method across **diverse data types and tasks**---CRL on numerical data and 3DIdent, fine-grained classification on CelebA, and world modeling on stochastic Moving-MNIST---to demonstrate its versatility.
>
> Nonetheless, to address the reviewer’s concerns regarding scalability, we have added a large-scale experiment on a **1M-image subset of iNaturalist spanning 5,000 classes**, using ResNet-50 as the backbone. Please see the joint response above for details.
>
> > Section 2 defines the data generation process in terms of latent variables z and z+, but in Section 3 these disappear to be replaced by a latent variable r presumably there to parameterize the predictor, similar to predictive SSL methods.
>
> SSL learns from pairs of data $(x, x^+)$. In Sec. 2.1, we adopt the perspective of CRL, and define the DGP that suggests that both $x$ and $x^+$ are deterministically mapped from latent factors $z$ and $z^+$ by $g$. The conditional distribution $p(z^+ \mid z)$ defines the relationship between the data pair. Our goal is to recover a subset of $z$, the data generating factors $z$, from observation $x$.
>
> $r$ is a latent variable we introduced in our method that captures the uncertainty in the conditional distribution $p(z^+ \mid z)$. We can define the joint as $p(z^+, z, r) = p(z)p(r \mid z)p(z^+ \mid z, r)$, where $p(z^+ \mid z, r)$ has low uncertainty because $r$ provides the missing information such as the driver’s intention to turn left or right if $z$ and $z^+$ are the current and future states of a traffic. In other words, $z, z^+$ are the latent state of the world and $r$ is the hidden cause that contributes to the transition from $z$ to $z^+$.
>
> We agree that omitting $r$ in our original DGP can cause confusion. We have made it clear where $r$ lies in the DGP in the revised Sec. 2.1.
>
> > I did not understand the difference between the AdaSSL loss and that typically used in predictive SSL, in part because the dependency of the model on the latent variable r is never defined before giving Eqs. (4) and (5) so I didn't understand how the terms in these equations were computed.
>
> Thanks for pointing this out. The difference between AdaSSL and existing predictive SSL methods is that $r$ is **learned/inferred** from data, instead of given, as we have stated in line 63 of the intro and illustrated in Fig. 1.
>
> The only modification we make to standard SSL methods (InfoNCE, BYOL) is to replace the representation $f(x)$ with $t(f(x), r)$ wherever it enters the loss. This leaves the loss itself unchanged, making the approach fairly general. In InfoNCE, $t(f(x), r)$ is used in the similarity function (Eq. 6, 7); in BYOL, we keep the predictor and feed it $t(f(x), r)$ instead of $f(x)$. No other parts of the architecture or objective are changed.
>
> We have added this clarification under Eq. 9. We have also added a sentence to state that BYOL’s predictor can condition on $r$ in Line 210.

---

> > ### Author Response · Authors · 2025-11-21
> > **part 2**
> >
> > > -psi_1 and psi_2 are used in Eq. (8) before they are defined in Eq. (9).
> >
> > Thanks. We have moved the definitions before Eq. 8 (Now Eq. 9).
> >
> > > -The function t, which was an arbitrary MLP until then, is defined explicitly in Eq. (11) as a modular editing function.
> >
> > In AdaSSL-V, $r$ is a continuous vector and $t$ is always an MLP. In AdaSSL-S, $r$ is sparse, and controls whether each module is active, similar to a mixture of experts. Therefore, $t$ is always a modular function in AdaSSL-S.
> >
> > We have made this clear in the revised manuscript.
> >
> >
> > > I could not find any justification as to why it should be possible to recover the latent variables since they are never used, as far as I know, in the actual loss of the AdaSSL variants. This is problematic since the experimental evaluation is for the most part dedicated to this recovery.
> >
> > In Sec. 2.1, we discuss prior work that has shown that SSL provably recovers the data generating factors under simple conditional distributions [Zim+21, Küg+21, Yao+24, Rus+25]. Intuitively,  representation learning with SSL encourages the diversity of representations, while making sure the encoded features are **shared (invariant) between positive pairs**. Contrastive learning achieves this by explicitly pushing apart representations of different data points, whereas non-contrastive learning does so through architectural tricks (e.g., [Zhu+23] shows that these designs increase the effective rank of the representations). Based on our DGP, because $g$ is deterministic, features learned by the model can only come from $z$.
> >
> > Our work extends SSL to tackle non-trivial conditional distributions. We continue to leverage the SSL objective to learn diverse features, but by allowing **structured differences** between the latents of positive pairs (i.e., $z$ and $z^+$) through $r$, we "reduce the pressure" for the SSL loss to discard features that are different.
> >
> >
> > > Remark: although it is frequently used in non-contrastive approaches to SSL, the EMA formulation in Eq. (3) is, as far as I know, ill defined since the exponential moving average is normally taken over the parameters defining psi over time.
> >
> > We follow the objective stated in the BYOL paper [Gri+20], and note that $\psi_\mathrm{EMA}$ is the EMA of $\psi$. We have added a clarification to suggest that the EMA is “taken over the parameters defining $\psi$ over time.”
> >
> > We are open to other suggestions to improve the precision of our exposition.
> >
> > > I understand how, from their probabilistic definition in terms of latent factors, natural data pairs may be different from the "augmented" pairs typically used in SSL. From an intuitive point of view, however, I do not really see how nearby video frames are qualitatively different from image crops, say. Both can be seen as crops, temporal or spatial, of the data. I would appreciate that the authors comment on this point.
> >
> > The key difference is that changes in the video frames do not act on all the image pixels as a whole, but only the causal entities in it, and are therefore very hard to simulate with augmentations.
> >
> > Crops can indeed change the size of the foreground object, but the background necessarily changes size with it. Differently, when one object moves towards or away from the camera, the background remains fixed. This helps the model to distinguish background and foreground, which may be important when their correlation changes (e.g., waterbird on land). The same reasoning applies to image rotations, which necessarily rotates the background with it.
> >
> > Temporal changes are perhaps even harder to simulate. For example, water in a glass may evaporate over time, without affecting other components of the image.
> >
> > > Please explain the significance of Prop. 2.1.
> >
> > Please see the joint response above.
> >
> > > As noted by the authors, AdaSSL-V is only justified for contrastive SSL, but it is used for non-contrastive SSL as well. Could you please justify this?
> >
> > Please see the joint response above.
> >
> > ---
> >
> >
> > We thank the reviewer again for their feedback. We hope our response and new results have addressed the concerns. We have added the suggested clarifications to the revised manuscript, which we believe further strengthen the paper. Please let us know if you have further questions.
> >
> >
> > ---
> >
> > # Refs.
> >
> > [Zhu+23] Towards a unified theoretical understanding of non-contrastive learning via rank differential mechanism. Zhuo et al., ICLR 2023.\
> > [Gri+20] Bootstrap your own latent: A new approach to self-supervised Learning. Grill et al., NeurIPS 2020.\
> > [Küg+21] Self-Supervised Learning with Data Augmentations Provably Isolates Content from Style. von Kügelgen et al., NeurIPS 2021.\
> > [Yao+24] Multi-view causal representation learning with partial observability. Yao et al., ICLR 2024.\
> > [Zim+21] Contrastive Learning Inverts the Data Generating Process. Zimmermann et al., ICML 2021.\
> > [Rus+25] InfoNCE: Identifying the Gap Between Theory and Practice. Rusak et al., AISTATS 2025.

---

> > > ### Comment · Reviewer_NVc3 · 2025-11-26
> > >
> > > Thanks for the clarifications. About crops vs video frames: it is of course true that video frames are hard to simulate. But it is not necessary to use *augmentations* of the data in SSL. On can use different *views* such as crops, both spatial and temporal in the video case. In LLMs for example as far as I know no augmentation is used. In recent SSL methods for videos such as V-JEPA, my understanding is that spatio-temporal crops are used as well:

---

> > > > ### Author Response · Authors · 2025-11-26
> > > >
> > > > Ah, good point! To clarify, we fully agree with your intuition that spatio-temporal crops are not fundamentally different from the examples we mention. In fact, we do consider them as "natural pairs." When we mention "nearby video frames," we include models that use spatio-temporal crops in the references in Line 40. The temporal aspect is what makes the distinction, as spatial crops alone are still limited for the reasons discussed in our previous response.
> > > >
> > > > **Spatio-temporal crops.**
> > > >
> > > > Our hypothesis is that models trained with spatio-temporal crops still inherit the limitations of standard SSL objectives when the available context (i.e., the unmasked patches) does not contain enough information to predict the target (i.e., masked patches).
> > > >
> > > > For example, V-JEPA reports that (a) masking a large continuous "tube" works better than (b) masking the entire future segment [Bar+24]. One possible reason is that when some information from each target frame is available, prediction becomes easier because the temporal uncertainty is greatly reduced. Results in our Stochastic Moving-MNIST experiments, where AdaSSL outperforms BYOL, show that we can improve SSL under strategy (b).
> > > >
> > > > Even under strategy (a), we hypothesize that prediction uncertainty can still grow as the frame rate decreases. This possibly explains why these models typically use a relatively high frame rate (e.g., only ~0.2 seconds between frames in V-JEPA). This could limit the predictor's ability to learn long-horizon dynamics, and benchmarks generally remain focused on short-horizon actions.
> > > >
> > > > It would be an interesting direction for future work to combine AdaSSL with the specific masking strategies used in these models.
> > > >
> > > >
> > > > **LLMs.**
> > > >
> > > > Our method is not directly applicable to LLMs, but consider this analogy: in next-token prediction, LLMs explicitly model uncertainty by learning a probability distribution over tokens. In contrast, standard SSL objectives do not explicitly account for uncertainty in the prediction target, which is precisely the motivation behind our work.
> > > >
> > > > **References.**
> > > >
> > > > [Bar+24] Revisiting Feature Prediction for Learning Visual Representations from Video. Bardes et al., TMLR, 2024.
> > > >
> > > > ---
> > > >
> > > > Please let us know if this addresses your question.

---

> > > > > ### Comment · Reviewer_NVc3 · 2025-11-27
> > > > >
> > > > > Thanks for the clarification. While I am not completely convinced by your arguments, you have addressed several of my concerns and I have raised my score to 6.

---

> > > > > > ### Author Response · Authors · 2025-11-27
> > > > > >
> > > > > > Thank you for your responsiveness and for reconsidering your score. We appreciate your constructive feedback and are glad that our response addressed several of your concerns.
> > > > > >
> > > > > > Thanks again for the time and effort in reviewing our work.

---

### Official Review · Reviewer_gRec · 2025-10-30

**Soundness:** 3
**Presentation:** 3
**Contribution:** 2
**Rating:** 4
**Confidence:** 4

**Summary:**

This paper presents novel self-supervised learning methods that model the uncertainty in data pairs generated from natural generative processes, using regularized latent variables. For example the uncertainty between consecutive frames in a video. Two variants are presented, one based on variational inference on the other on enforcing sparsity on the latent variable. Experiments on artificial and reel data are conducted to demonstrate the effectiveness of the approach in identifying latent factors of variations and modelling uncertainty.

**Strengths:**

- The paper tackles a key problem in self-supervised learning: modelling conditional uncertainty, which arises in many other problems related to causal prediction. The potential applications are therefore numerous: video prediction, video generation, world modelling and latent action prediction, efficient self-supervised learning ect. The paper could actually do a better job at motivating these applications.

- The ideas presented in the paper are interesting, described in depth, well-motivated, and seem to be good candidate solutions, at least on the toy problems explored in the experimental section.

**Weaknesses:**

- The paper is complex to understand, with lots of formalism (the probabilist framework, Proposition 2.1), complex vocabulary (heteroscedasticity, modular editing, DCI) that is not necessarily introduced, or introduces lots of new vocabulary (CRL, DGP, SSL from structural invariance, Adaptive SSL, natural pairs), all for ideas that are actually fairly simple. I feel like this over–complixification hinders the reading flow and makes it harder to deliver the message it intends to deliver.

- Also, the paper put a lot of emphasis on particular SSL losses such as contrastive vs non-contrastive, as well as several variants of InfoNCE, which does not seem very relevant for this study, and adds too many factors of variation that make the conclusions of the paper less clear and convincing. Section 2.2 and 2.3 are probably unnecessary, and the new variants H-InfoNCE, AnInfoNCE, ect are not well motivated.

- Finally, all this formalism is derived by the JEPA framework and the authors mention that they only take inspiration from JEPA, whereas I see these contributions as instanciations of JEPAs, just with various ways of regularizing the latent variables. In Figure 1, b) and c) are JEPAs.

- The experiments are conducted on toy problems which limits the credibility of the approach. How does it behave on more concrete problems ? I don’t think people care about the artificial numerical problems of section 4.2, these should be more of a tool for you to debug the approach. Section 4.2 is interesting but very artificial, and 4.3 is Moving MNIST which is good again for debugging but nowhere near close to the actual interesting problems.  Finally, the focus is on velocity decoding, which is good for debugging but not as interesting as the problem of prediction. It would be more interesting to show experiments where a model predicts the future trajectory in moving MNIST, and being able to sample several possible future trajectories by sampling from the latent.

- Related to this, the claims made at the beginning of the paper need to be toned down, for example Line 20 “and we empirically show its superiority on identifiability, generalization, fine-grained image understanding, and world modeling on videos”. Superiority against which concurrent method ? And on benchmarks that are too toy.

- The paper ignores the vast literature existing on uncertainty modelling and latent variables. All the work in generative modes, video generative models, video prediction, latent action models.

- In conclusion, the paper is tackling an interesting problem and presents interesting ideas but it is hard to be convinced by the toy experiments. These points would make it much stronger:

- Remove the studies on InfoNCE variants, along with sections 2.2 and 2.3, and focus on AdaSSL. Maybe rename using the JEPA terminology and just name the latent variable regularization methods.

- Remove section 4.2 and focus more on real data experiments.

- Add more motivations in terms of potential applications
- Acknowledge other literature in uncertainty modelling and world modelling.

- Focus the experiments more on video world modelling, and the prediction capability, rather than training probes to recover properties such as velocity.

**Questions:**

- Line 160: Then the solution is just to project and do prediction in the same space ?

- In AdaSSL-V, how could you make more explicit the mechanism that regularizes the latent variable regularized, basically what is L_reg ?

---

> ### Author Response · Authors · 2025-11-21
> **part 1**
>
> We thank the reviewer for recognizing the potential applications our method implies and that our idea is interesting and well-motivated. We also appreciate the feedback on the structure of the paper and actionable suggestions. Please find our response to your comments below.
>
> > The paper is complex to understand, with lots of formalism (the probabilist framework, Proposition 2.1), complex vocabulary (heteroscedasticity, modular editing, DCI) that is not necessarily introduced, or introduces lots of new vocabulary (CRL, DGP, SSL from structural invariance, Adaptive SSL, natural pairs), all for ideas that are actually fairly simple. I feel like this over–complixification hinders the reading flow and makes it harder to deliver the message it intends to deliver.
>
> We appreciate the reviewer's concern about readability.
>
> We want to clarify that heteroscedasticity is a standard term to describe the conditional noise in statistical ML, while CRL and DGP are well-established terms in causal representation learning. SSL can be seen as a special case of multi-view CRL [Yao+25]. We introduce these terms early in the paper (lines 42 and 45). DCI is a common evaluation metric in CRL.
>
> We believe that the probabilistic framework, the formal definitions, and Proposition 2.1 are essential to precisely state the issue of SSL that we are addressing. We follow the suggestion of recent position papers that have encouraged formalism of SSL to push the field forward [Rei+25] as well as the line of work that studies the intersection of SSL and CRL [Zim+21, Küg+21, Yao+24, Rus+25].
> While a less formal presentation is possible, we are concerned that it would sacrifice precision and blur the scientific contribution, making it significantly harder to evaluate or reproduce. We note that we provide intuition and the objective of our study in the intro and problem definition in Sec. 2.
>
> That said, we acknowledge that some parts lacked clarity, and we have added additional intuitions to the problem definition (Sec. 2.1), Proposition 2.1, and the method motivation (Sec. 3.1) to improve readability.
>
> > Also, the paper put a lot of emphasis on particular SSL losses such as contrastive vs non-contrastive, as well as several variants of InfoNCE, which does not seem very relevant…
>
> > Remove the studies on InfoNCE variants, along with sections 2.2 and 2.3, and focus on AdaSSL.
>
> We want to clarify that we **directly build on the line of work that examines how contrastive learning recovers the underlying data generating factors under different assumptions of the conditional distribution** $p(z^+ \mid z)$ [Zim+21, Rus+25]. AnInfoNCE, a recent improvement of InfoNCE that considers anisotropic noise in the conditional, is proposed to address the problem we are tackling directly [Rus+25]. Therefore, we believe they are relevant to our study. H-InfoNCE addresses heteroscedastic noise, while AdaSSL addresses more complex conditionals (e.g., multimodal).
>
> To evaluate generality beyond contrastive SSL, we include BYOL as a representative non-contrastive method whose architecture is shared by state of the arts such as I/V-JEPA. These choices therefore cover both the theoretically analyzed and the practically dominant branches of SSL, demonstrating that AdaSSL applies to both.
>
> Sections 2.2 and 2.3 provide essential background explaining why these major branches of SSL are limited in their ability to model the complex conditional $p(x^+ \mid x)$ that arises in natural pairs. We believe removing them would make it harder to understand why AdaSSL is needed and how it differs from prior work.
>
> > Finally, all this formalism is derived by the JEPA framework and the authors mention that they only take inspiration from JEPA, whereas I see these contributions as instantiations of JEPAs, just with various ways of regularizing the latent variables. In Figure 1, b) and c) are JEPAs.
>
> > Maybe rename using the JEPA terminology and just name the latent variable regularization methods.
>
> We want to clarify that our formalism is not derived from the JEPA framework: the DGP follows conventions in CRL and the loss is derived from the mutual information. The resulting method belonging to the JEPA family of methods is a consequence of our derivation, not the starting point.
>
> Please see the end of Sec. 3.1, where we **explicitly state that our objective matches the JEPA framework**. Please let us know if you have additional concerns.

---

> > ### Author Response · Authors · 2025-11-21
> > **part 2**
> >
> > > The experiments are conducted on toy problems which limits the credibility of the approach. How does it behave on more concrete problems? I don’t think people care about the artificial numerical problems of section 4.2, these should be more of a tool for you to debug the approach. Section 4.2 is interesting but very artificial, and 4.3 is Moving MNIST which is good again for debugging but nowhere near close to the actual interesting problems. Finally, the focus is on velocity decoding, which is good for debugging but not as interesting as the problem of prediction. It would be more interesting to show experiments where a model predicts the future trajectory in moving MNIST, and being able to sample several possible future trajectories by sampling from the latent.
> >
> > > Remove section 4.2 and focus more on real data experiments.
> >
> > > Focus the experiments more on video world modelling, and the prediction capability, rather than training probes to recover properties such as velocity.
> >
> >
> > **Evaluation.**
> > Please see the joint response above.
> >
> > **World modeling analysis.**
> > Based on your suggestion, we performed additional visualizations and analysis on the learned world models on Stochastic Moving-MNIST in Appendix D.3 (summary below).
> >
> > ### A. Predicting the future trajectory (Appendix D.3.2)
> >
> > Given a source trajectory and its ground-truth latent $r$ (change in velocity), we predict the embedding of the corresponding future trajectory with the learned predictors ("world models") of BYOL and AdaSSL-V/S and retrieve the five nearest matches from a pool of 4096 future trajectory embeddings.
> >
> > Table D reports MRR and Hit@k which are standard metrics for assessing predictor quality [Gar+23]. AdaSSL consistently outperforms BYOL. Fig. 9 shows the retrieved trajectories and the rank of the ground-truth target in the retrievals. We observe that all methods are able to retrieve the correct digit, but AdaSSL’s predicted future trajectories better match the targets in velocity.
> >
> > **Table D. Mean reciprocal rank (MRR) and hit rate at k (Hit@k) of the retrievals from 4,096 future trajectories on Stochastic Moving-MNIST.**
> >
> > Method | MRR ($\uparrow$) | Hit@1 ($\uparrow$) | Hit@5 ($\uparrow$) |
> > | :---  | :---: | :---: | :---: |
> > | BYOL | 0.1758 | 0.1182 | 0.1802 |
> > | AdaSSL-V | 0.4102 |  0.3071  |  0.5129 |
> > | AdaSSL-S | **0.5291** | **0.4338** | **0.6287** |
> >
> >
> >
> >
> > ### B. Sampling future trajectories (Appendix D.3.3)
> >
> > We evaluate the diversity of future trajectory predictions. Given a source trajectory, we embed it and sample multiple latents $r$ (using the learned prior for AdaSSL variants) to generate multiple predicted target embeddings. We then visualize their nearest neighbor.
> >
> > Fig. 10 shows the overlaid ground-truth future trajectories and predictions. BYOL produces a single deterministic prediction, whereas the AdaSSL-V predicts diverse plausible futures. Similar qualitative results hold across other benchmarks (Fig. 2, Fig. 7).
> >
> >
> > **Experiment scale.**
> > We have conducted a large-scale experiment on a subset of iNaturalist with 1M training images spanning 5000 classes, using ResNet-50 as the backbone. Please see the joint response above.
> >
> >
> > **Interesting problems.**
> > While there are numerous applications that we can test, a necessary first step toward scientific progress is to perform **controlled experiments** to examine under what conditions do existing methods fail. We do that by carefully changing the conditional distribution (Sec. 4.1). While video tasks are an obvious application of our method, we note that we are tackling a fundamental issue in latent SSL objectives. That’s why we choose to test our method’s versatility on **different types of data and tasks** (CRL on numerical data and 3DIdent, fine-grained classification on CelebA and iNaturalist, and world modeling on stochastic Moving-MNIST).
> >
> >
> > > … the claims made at the beginning of the paper need to be toned down…
> >
> > Thank you for pointing out the imprecision in our wording. We have changed the word “superiority” to “versatility.”
> >
> > > Acknowledge other literature in uncertainty modelling and world modelling.
> >
> > We’d like to point out that, in related work (Line 521), we have acknowledged the literature that performs uncertainty modeling and world modeling in the observation space. We have also extensively performed the literature review on latent SSL that uses action conditioning in the first paragraph of Sec. 5. If the reviewer has any specific references in mind, we’d be happy to include them as well.
> >
> > It is worth noting that the uncertainty modeling in latent SSL is underexplored (we are only aware of [Zim+21, Rus+25]). This is partly because it is a much harder task than SSL in the observation space because latent SSL, which does not require reconstructing the target, can discard features that are different between the positive pairs. This is why we propose to shift the perspective from **feature invariance** to **structural invariance** in the conditional distribution.

---

> > > ### Author Response · Authors · 2025-11-21
> > > **part 3**
> > >
> > > > The potential applications are therefore numerous: video prediction, video generation, world modelling and latent action prediction, efficient self-supervised learning etc. The paper could actually do a better job at motivating these applications.
> > > > Add more motivations in terms of potential applications.
> > >
> > > Thank you for the suggestion.
> > >
> > > We want to point out that, in the paper, we have mentioned our method’s utility in equivariant SSL in line 446, in CRL in Line 454, and in world modeling in Line 519. We have also motivated the need to have such a method in the intro.
> > >
> > > We have now added Appendix E to discuss the potential applications of our method.
> > >
> > >
> > > > Line 160: Then the solution is just to project and do prediction in the same space ?
> > >
> > > The ground-truth latent space of the data generating factors is unknown, and SSL methods typically choose the hypersphere or $R^d$. Prop. 2.1 tells us one needs to consider such mismatches. Please also see our joint response on the significance of Prop. 2.1.
> > >
> > > > In AdaSSL-V, how could you make more explicit the mechanism that regularizes the latent variable regularized, basically what is L_reg ?
> > >
> > > For AdaSSL-V, as stated in Eq. 9,  $L_\mathrm{reg}$ is the KL divergence between the prior $p(r \mid x)$ and the variational posterior $q(r \mid x, x^+)$. The modeling of these distributions follows a standard practice in variational inference [Kin+14, Soh+15]. Specifically, both $p$ and $q$ are factorized Gaussians. $p$ is an MLP that takes in $f(x)$, and predicts the mean and log variance of $p$. Similarly, $q$ is an MLP that takes in $f(x)$ and $f(x^+)$, and predicts the mean and log variance of $q$. There is a closed form solution for the KL between factorized Gaussians, and we use that as the loss.
> > >
> > > We have added an explicit statement to specify that $D_\mathrm{KL}$ refers to the KL divergence in the revised manuscript.
> > >
> > >
> > > ---
> > >
> > > We thank the reviewer again for their feedback. We hope that our response can help clarify some confusions. We also hope the additional results, clarifications, and discussions for future work have strengthened our paper. Please let us know if you have further questions.
> > >
> > >
> > > ---
> > >
> > > # Refs.
> > > [Rei+25] Position: An Empirically Grounded Identifiability Theory Will Accelerate Self-Supervised Learning Research. Reizinger et al., ICML 2025.\
> > > [Yao+25] Unifying causal representation learning with the invariance principle. Yao et al., ICLR 2025.\
> > > [Zim+21] Contrastive Learning Inverts the Data Generating Process. Zimmermann et al., ICML 2021.\
> > > [Rus+25] InfoNCE: Identifying the Gap Between Theory and Practice. Rusak et al., AISTATS 2025.\
> > > [Kin+14] Auto-Encoding Variational Bayes. Kingma et al., ICLR 2014.\
> > > [Soh+15] Learning Structured Output Representation using Deep Conditional Generative Models. Sohn et al., NIPS 2015.\
> > > [Küg+21] Self-Supervised Learning with Data Augmentations Provably Isolates Content from Style. von Kügelgen et al., NeurIPS 2021.\
> > > [Yao+24] Multi-view causal representation learning with partial observability. Yao et al., ICLR 2024.\
> > > [Gar+23] Self-supervised learning of Split Invariant Equivariant representations. Garrido et al., ICML 2023.

---

> ### Author Response · Authors · 2025-11-27
> **Quick follow-up**
>
> Dear Reviewer gRec,
>
> As the rebuttal period is nearing its end, we wanted to briefly follow up.
>
> We have responded to all of your comments above and updated the paper accordingly. We believe these revisions have strengthened the work, and we would be happy to clarify anything further if you have remaining questions.
>
> If you feel that our responses adequately address your concerns, we kindly ask you to reconsider your rating in light of the rebuttal.
>
> Thank you again for your time and effort in reviewing our paper!
>
> Best,\
> Authors

---

### Official Review · Reviewer_TJcB · 2025-10-31

**Soundness:** 3
**Presentation:** 2
**Contribution:** 3
**Rating:** 6
**Confidence:** 4

**Summary:**

The paper proposes a model for self supervised learning that extends previous approaches to include an auxiliary variable that capture the variation between representation of related data, e.g. augmentations or naturally occurring variations. The results suggest indicate that the model is better able to adapt to data generated under more flexible assumptions, such as heteroskedasticity, relative to baselines.

**Strengths:**

The paper is relatively clear and well explained (see weaknesses for exceptions). The motivation seems sound and the proposed solutions are rationally justified. Experimental results suggest the proposed models are effective.

**Weaknesses:**

A main weakness of the paper is its occasional lack of clarity. For the most part the paper is easy to follow, which makes the following very noticeable. The paper would improve greatly if these areas were addressed.
* 33 - this does not seem to be about "distribution shift", i.e. a change to the distribution, but rather that artificial augmentations do not span the full variation in the distribution of natural images
* Prop 2.1 - this is unclear, e.g. in line 156 we already know h maps to the unit sphere so why restate? It is unclear what this proposition adds since it appears to be specific to Gaussian distributions under a mapping between different topologies. This is a contrived scenario so it is unclear how it is necessarily relevant or "unavoidable" in practical scenarios. If the point is that the distributions of $z^+|z$ may vary over $z$ for real world datasets, which sounds highly plausible, Prop 2.1 doesn't appear to prove that and seems redundant.
* 199: Eq 4 does not appear to support the "intuition" that follows, e.g. $r$ could convey no information and Eq 4 holds, so "*should* help" is untrue.
* 209: The use of $p$ and $q$ suggest that $p$ is a ground truth posterior distribution over $r$ that $q$ learns to approximate similarly to a VAE, but a VAE is quite different since a ground truth posterior $p(z|x)$ is defined by the model's likelihood and prior, which $q$ provably learns to approximate. Here there is no defined prior over $r$ or ground truth posterior $p$, so the notation seems spurious (it would seem more accurate to refer to $r$ as an "auxiliary" variable (as in Khemakhem et al. (2020)) and "conditional" $q$.
* 234: it is unclear how $r$ is included in InfoNCE or how the projector compares to the MLP used in BYOL. This is a key part of describing the model and should be very clear.
* 308: the distinction between embeddings and representations is unclear from the text.
* 4.2: this section is very difficult to read, e.g.
    - "$f$ is the frozen encoder trained on $p(z)$" presumably means $f$ was trained under the SSL algorithm given $x, x^+$ pairs sampled under the described generative process? If so, that is hard to parse and should be made clearer.
    - (a) appears to describe generating data under the described method and training a regressor to predict the true generative $c$ component of $z$ from the representation $f(x)$. It is much less clear what (b) and (c) describe (c appears to be about robustness of the regressor?)
    - since $\Sigma$ is unstated, the significance of $5I$ in the "OOD" cases has no context (presumably 5 is higher variance than $p(z)$)?
* "OOD experiments" - the theoretical background does not appear to suggest robustness under distribution shift, so there is no clear explanation for the improved results and "only flexible models generalize OOD" seems unjustified. It is fair to note the improved OOD results, but it seems unexplained.
    - 366 - as above, the emphasis on OOD sees out of keeping with the rest of the paper. The model is designed to learn more flexible latent conditionals, this is shown in column 1 of Table 2, which seems the main result justifying the approach. No explanation has been given why this model would be expected to perform better OOD.
* 377 - "identifiability" - what does this refer to?

CelebA results: while the paper considers an adaptation to the InfoNCE loss, I believe that InfoNCE does not achieve state of art performance, so the results are not well contextualised in terms of current model performance.

**Questions:**

see weaknesses

---

> ### Author Response · Authors · 2025-11-21
> **part 1**
>
> Thank you for acknowledging the soundness of our motivation and the proposed solution! We appreciate the detailed feedback on the clarity of our paper. Please find our response below.
>
> > 33 - this does not seem to be about "distribution shift", i.e. a change to the distribution, but rather that artificial augmentations do not span the full variation in the distribution of natural images
>
> We agree that the rotation example does not directly illustrate a distribution shift. Our point is that real-world shifts often arise from changes in the distribution of underlying natural factors (e.g., lighting changes when moving from indoor to outdoor scenes), making invariance or equivariance to such factors important.
>
> We have rewritten this sentence to clarify this point.
>
> > Prop 2.1 - this is unclear, e.g. in line 156 we already know h maps to the unit sphere so why restate?
>
> Thanks. We have removed the redundant assumption.
>
> > It is unclear what this proposition adds…
>
> Please see the joint response for the significance of Prop 2.1.
>
> > 199: Eq 4 does not appear to support the "intuition" that follows, e.g. $r$ could convey no information and Eq 4 holds, so "should help" is untrue.
>
> We thank the reviewer for pointing out the ambiguity. Eq. 4 is indeed an identity and does not by itself ensure that $r$ encodes useful information. Our intention in presenting it is to clarify how $r$ is meant to be used by the objective: the SSL term is intuitively aligned with improving $I(x,r; x^+)$, pushing $r$ to encode information that reduces the uncertainty in predicting $x^+$, while the regularizer prevents shortcuts by discouraging $r$ from directly copying $x^+$.
>
> We have revised the text under Eq. 4 & 5 to make this intuition precise.
>
>
>
>
> > 209: The use of  $p$ and $q$ suggest that $p$  is a ground truth posterior distribution over $r$  that $q$ learns to approximate similarly to a VAE, but a VAE is quite different since a ground truth posterior $p(z \mid x)$ is defined by the model's likelihood and prior, which $q$ provably learns to approximate. Here there is no defined prior $r$ over  or ground truth posterior $p$, so the notation seems spurious…
>
> Thank you for raising this concern. The confusion stems from the fact that we introduced the latent variable $r$ later in the method section, and did not make it explicit in our original description of the data generating process (DGP).
>
> We have now defined the full joint as $p(x^+, x, r) = p(x^+ \mid  x, r) p(r \mid x) p(x) $, which is analogous to the conditional VAE formulation (see Fig. 1b of [Soh+15](https://papers.nips.cc/paper_files/paper/2015/file/8d55a249e6baa5c06772297520da2051-Paper.pdf)). Under this decomposition, the conditional of interest is $p(x^+ \mid x) = \int p(x^+ \mid x, r)p(r \mid x) dr$ and a prior over $r$ is defined through $p(r \mid x)$.
>
> We have added this joint factorization to Sec. 2.1 when we describe our DGP.
>
> > 234: it is unclear how $r$ is included in InfoNCE or how the projector compares to the MLP used in BYOL. This is a key part of describing the model and should be very clear.
>
> Thanks for the feedback. The only modification we make to standard SSL methods (InfoNCE, BYOL) is to replace the representation $f(x)$ with $t(f(x), r)$ wherever it enters the loss. This leaves the loss itself unchanged, making the approach fairly general. In InfoNCE, $t(f(x), r)$ is used in the similarity function (as shown in Eq. 6 & 7); in BYOL, we keep the predictor and feed it $t(f(x), r)$ instead of $f(x)$. No other parts of the architecture or objective are changed.
>
> We have added this clarification under Eq. 9.
>
> > 308: the distinction between embeddings and representations is unclear from the text.
>
> We state the distinction between embeddings (output of the encoder+projector) and representations (output of the encoder) in the experimental setup paragraph in Sec. 4.1. Our method operates on the embeddings $f(x)$.
>
> > "$f$ is the frozen encoder trained on $p(z)$ " presumably means $f$ was trained under the SSL algorithm given $x,  x^+$ pairs sampled under the described generative process? If so, that is hard to parse and should be made clearer.
>
> Thank you for the feedback. Your interpretation is correct and we have rewritten it:
>
> "We then train $f$ on the $(x, x^+)$ pairs under the SSL algorithms and freeze it. During evaluation, we train a linear regressor to predict $c$ from $f(x)$, where $x = g([c, s])$."
>
> > 377 - "identifiability" - what does this refer to?
>
> Identifiability refers to the study of under which conditions the ground truth latent factors can be inferred, or “identified”, from the data [Rei+25], whereas CRL is the practical problem of recovering the data generating factors from high-dimensional data. While we do not theoretically show that our method provably recovers all underlying data generating factors, we empirically demonstrate its promise in the CRL experiment.
>
> We have clarified our results are empirical in the revised manuscript.

---

> > ### Author Response · Authors · 2025-11-21
> > **part 2**
> >
> > > (a) appears to describe generating data under the described method and training a regressor to predict the true generative $c$ component of $z$ from the representation $f(x)$. It is much less clear what (b) and (c) describe (c appears to be about robustness of the regressor?)
> >
> > In all three cases, we train a regressor to predict $c$ from the frozen representation $f(x) = f(g(z))$; the difference lies only in the distributions used at training and evaluation time of the regressor.
> >
> > - (a) Both training and testing use $z \sim p(z)$.
> > - (b) Both training and testing use $z \sim \mathcal{N}(0, 5I)$. This tests whether the representation supports accurate prediction of $c$ under covariate shift.
> > - (c) The regressor is trained on $z \sim p(z)$ but evaluated on $z\sim \mathcal{N}(0, 5I)$. This corresponds to a practical scenario where distribution shifts happen after deployment when both the encoder and regressor are frozen. This setting requires not only a robust regressor but also that **the representation be well aligned across the two distributions**.
> >
> > An intuitive graphical illustration of the difference between (b) and (c) in binary classification is Fig. 1c of [[Rua+22]](https://arxiv.org/pdf/2201.00057). A regressor can perform well on (b) but fail on (c) if the representation alignment between training and testing distributions is poor.
> >
> > We have added this clarification to the manuscript.
> >
> >
> > > since $\Sigma$ is unstated, the significance of $5I$ in the "OOD" cases has no context (presumably 5 is higher variance than $p(z)$)?
> >
> > Thanks for pointing this out. We mention that $p(z)$ contains correlated latents but only define $\Sigma$ properly in the appendix.
> >
> > In our setup, $\Sigma$ is sampled from an Inverse-Wishart distribution $\mathcal{W}^{-1}(n_c+2, I)$, whose expectation is $I$. Thus, the OOD cases indeed use a latent distribution with larger marginal variance. Also, each sampled $\Sigma$ typically contains nonzero correlations, whereas the OOD setting uses uncorrelated latents.
> >
> > We have added a brief description of $\Sigma$ where it is first introduced and results over three random draws of $\Sigma$.
> >
> >
> > > "OOD experiments" - the theoretical background does not appear to suggest robustness under distribution shift, so there is no clear explanation for the improved results and "only flexible models generalize OOD" seems unjustified. It is fair to note the improved OOD results, but it seems unexplained.
> >
> > > 366 - as above, the emphasis on OOD seems out of keeping with the rest of the paper. The model is designed to learn more flexible latent conditionals, this is shown in column 1 of Table 2, which seems the main result justifying the approach. No explanation has been given why this model would be expected to perform better OOD.
> >
> > Thank you for raising this question. We hypothesize that there are two factors that contribute to generalization:
> >
> > **A. Independent variation in $p(c^+ \mid c)$ helps disentangle factors.**
> > Independent variation in the conditional $p(c^+ \mid c)$ provides signals that distinguish latent dimensions that may be correlated under $p(c)$. For example, observing the same bird in different environments introduces variation along the background dimension while keeping the foreground largely fixed. This can help the model to encode these factors separately rather than learning the correlations.
> >
> > In Table 1, when $\mathrm{Var}(c^+ \mid c) = 0$, i.e., $c^+ = c$, vanilla InfoNCE performs almost perfectly in-distribution, potentially because it can leverage correlations and achieve a low SSL loss without encoding all latent factors. However, it degrades under OOD evaluation, whereas the best models trained with $\mathrm{Var}(c^+ \mid c) \neq 0$ generalize better. This suggests independent variation in the conditional may help better latent recovery and, in turn, generalization.
> >
> > **B. A flexible model benefits more.**
> > Our sentence that suggests only flexible models generalize OOD is indeed misleading. The reason is likely that they capture more data-generating factors (i.e., approximate $g^{-1}$ more closely) under complex conditionals, and this benefit is more noticeable on the OOD results.
> >
> > We have adjusted our wording accordingly in the manuscript.
> >
> >
> >
> > > …InfoNCE does not achieve state of art performance, so the results are not well contextualised in terms of current model performance.
> >
> > Please see the joint response above.
> >
> > ---
> >
> > We thank the reviewer for their detailed feedback, which helped us improve our work. We hope that our rebuttal has addressed all of the raised concerns. We remain available for further discussion or clarification.
> >
> > ---
> >
> > # Refs.
> >
> > [Soh+15] Learning Structured Output Representation using Deep Conditional Generative Models. Sohn et al., NIPS 2015.\
> > [Rei+25] Position: An Empirically Grounded Identifiability Theory Will Accelerate Self-Supervised Learning Research. Reizinger et al., ICML 2025.\
> > [Rua+22] Optimal Representations for Covariate Shift. Ruan et al., ICLR 2022.

---

> ### Author Response · Authors · 2025-11-27
> **Quick follow-up**
>
> Dear Reviewer TJcB,
>
> As the rebuttal period is nearing its end, we wanted to briefly follow up.
>
> We have responded to all of your comments above and updated the paper accordingly. We believe these revisions have strengthened the work, and we would be happy to clarify anything further if you have remaining questions.
>
> Thank you again for your time and effort in reviewing our paper!
>
> Best,\
> Authors

---

### Author Response · Authors · 2025-11-21
**General response (part 1)**

We thank all reviewers for their thoughtful feedback. We are encouraged that the reviewers find our motivation sound **(TJcB: "sound"; gRec: "key problem", "well-motivated"; NVc3: "interesting"; y8NR: "promising and under-explored", "strong motivations")** and our method interesting and rationally-justified **(gRec: "novel", "described in depth", "good candidate solutions"; y8NR: "interesting", "enjoyable to read", "extensive experiments backed by theoretical and empirical analysis"; TJcB: "relatively clear and well explained", "rationally justified")**.

Below we address the common questions and respond to the remaining concerns individually.



# Significance of Prop 2.1 (Reviewers TJcB, NVc3)

**Intuition of Prop 2.1.**
We show that when the ground-truth latent space $Z$ has a different geometry from the embedding space $E$, the conditional distribution $h(z^+)\mid h(z)$ becomes heteroscedastic in these settings. This arises **even when the true latent conditional $z^+\mid z$ is homoscedastic**: the curvature mismatch forces the encoder to introduce input-dependent uncertainty.

We demonstrate this for two concrete cases, $Z=R^d, E=S^{d-1}$ (Prop. 2.1) and $Z=S^{d-1}, E=R^d$ (Prop. B.2). While we leave the fully general statement for future work, the geometric intuition is simple: mapping between spaces with different curvature stretches or compresses local neighborhoods differently depending on the location of $h(z)$, which directly induces heteroscedasticity in $h(z^+)\mid h(z)$.

**Significance for SSL.**
This matters for SSL because existing objectives implicitly assume that the variability in $h(z^+)\mid h(z)$ can be modeled with a fixed-variance similarity function (Sec. 2.2) or predictor (Sec. 2.3). When the geometry of $Z$ and $E$ differ---which is likely, since $Z$ is unknown while $E$ is chosen as either $S^{d-1}$ or $R^d$---this assumption is violated. We show that existing methods struggle even under the simple case of $Z=R^d$.

Empirically, when $Z$ and $E$ do not match, existing SSL methods struggle to recover the latent factors (Table 1), consistent with [Zim+21] ((Table 4, rows 2 & 4). Modeling this input-dependent noise (via H-InfoNCE) alleviates the issue.

We clarify the connection between Prop. 2.1 and our problem in Sec. 2.4 of the revised manuscript.


# Representation learning and downstream tasks (Reviewers gRec, NVc3, y8NR)

We adopt the causal representation learning (CRL) view that a central goal of representation learning is to recover the data-generating factors $z$ from observations $x$ [Sch+21, Ben+13, Zim+21]. Under this view, a "diverse" representation is one that captures the full latent variability in $z$, because all semantic information in $x$ originates from $z$. This objective is formalized in Sec. 2.1, and we study this in the setting of natural pairs as motivated in the intro.

We argue that learning a diverse representation is important for general-purpose representations for different downstream tasks, which may need some unknown subset of $z$ for inference [Ben+13]. For example, a world model cannot generate meaningful future trajectories of cars if its representation does not contain velocity.

We follow the standard evaluation protocols in representation learning with SSL: training a linear probe and evaluating the model’s performance through its ability to predict the data-generating factors [Che+20, Gri+20]. When we don’t have access to these factors, we use proxy tasks, such as fine-grained classification on CelebA and iNaturalist (results shown below).

We have revised Sec. 2.1 to make explicit how our formal objective connects to downstream performance.



# Clarity of the paper


We thank the reviewers for their suggestions regarding clarity. We have addressed each point in our individual responses and incorporated the corresponding revisions into the manuscript, with changes highlighted in **red**.

---

> ### Author Response · Authors · 2025-11-21
> **General response (part 2)**
>
> #  The choice of SSL objectives (Reviewers TJcB, NVc3, y8NR)
>
> **Justification for the selection of SSL objectives.**
> As discussed in Sec. 2.2, SSL methods fall broadly into contrastive and non-contrastive approaches.
> For contrastive methods, we compare with InfoNCE and AnInfoNCE because (a) prior work explicitly analyzes how they model $p(z^+ \mid z)$ [Zim+21, Rus+25], and (b) InfoNCE-based contrastive learning remains state-of-the-art in many applications [Xu+24, Wan+24, Beh+24].
> To evaluate generality beyond contrastive SSL, we also include BYOL as a representative non-contrastive method whose architecture is shared by state of the arts such as I/V-JEPA. These choices therefore cover both the theoretically analyzed and the practically dominant branches of SSL.
>
> **Intuition for why our method applies to non-contrastive SSL.**
> Both contrastive and non-contrastive objectives encourage the representations of positive pairs to be close. As a result, the model tends to encode only the features of $z^+$ that are predictable from $z$; unpredictable components (e.g., $z_i$ that has a high variance in $p(z_i^+ \mid z)$) are suppressed because the optimal solution under such uncertainty is to regress toward the mean. Conditioning on $r$ reduces this uncertainty, enabling the model to retain information in $f(x^+)$ that would otherwise be discarded.
>
> **Empirical results with BYOL.**
> Our video experiments in Sec. 4.4 confirm that our method improves BYOL. We additionally report CelebA results in Table A: consistent with the InfoNCE observations, natural pairs offer large gains under weak augmentation, and AdaSSL-V improves BYOL across all settings, achieving performance close to BYOL+GT under weak augmentation (0.58 vs. 0.59). Due to compute limits, we include AdaSSL-V here and plan to add AdaSSL-S in future work. Per reviewer gRec’s suggestion, we also added retrieval analysis for BYOL+AdaSSL-S/V on Stochastic-Moving-MNIST to Appendix D.3.
>
> **Table A. BYOL results on CelebA. The model is trained with either weak or strong augmentations. Test $F_1$ scores of linear probes on the representations or embeddings are reported. BYOL+GT uses the GT attribute difference between pairs and serves as a soft upper bound.**
>
>
> | Method | Pairing | Repr. (weak) | Emb. (weak) | Repr. (heavy) | Emb. (heavy)  |
> | :---  | :---: | :---: | :---: | :---: | :---: |
> | BYOL | Standard | 0.2989 $\pm$ 0.0025 | 0.1832 $\pm$ 0.0037 | 0.5368 $\pm$ 0.0013 |  0.5043  $\pm$ 0.0037 |
> | BYOL | Natural | 0.5465 $\pm$ 0.0018 |  0.4263  $\pm$ 0.0019 |  0.5608 $\pm$ 0.0004 | 0.5019 $\pm$ 0.0013 |
> | AdaSSL-V | Natural | **0.5816** $\pm$ 0.0035 | **0.5067** $\pm$ 0.0051 | **0.5702** $\pm$ 0.0017 | **0.5302** $\pm$ 0.0018 |
> | BYOL+GT | Natural | 0.5948 $\pm$ 0.0042 | 0.5730 $\pm$ 0.0016 | 0.5984 $\pm$ 0.0024 | 0.5872 $\pm$ 0.0003 |
>
>
>
>
> # Scale of the experiment (Reviewers gRec, NVc3)
>
> Our method already works across **multiple data types (numerical data, images, videos)**, and here we additionally demonstrate scalability to a large dataset. We added an experiment on a 1M-image subset of iNaturalist 2021 [GM21], a standard large-scale benchmark with **5,000 fine-grained species classes**. Each class belongs to one of the 11 superclasses. We train ResNet-50 encoders at 224×224 resolution.
> This setting also lets us study how pairing quality affects downstream performance. We consider
> - fine-grained pairs from the same class.
> - coarse-grained pairs from the same superclass.
> - a mixed setting with 50% fine and 50% coarse pairs to mimic realistic noisy labels, where wrong labels are not random, but from the correct superclass.
>
> Results in Tables B and C shows:
> - Fine-grained pairs: AdaSSL-V matches InfoNCE as expected, as this setup is close to supervised contrastive learning.
> - Coarse or noisy pairs: AdaSSL-V is more robust. When training only on superclass pairs, AdaSSL-V doubles the top-1 and top-5 accuracy of InfoNCE, indicating that it can better discover fine-grained structure without fine-grained labels.
>
> These findings show that our method scales effectively to large image datasets and remains robust under degraded or noisy pairing.
>
>
> **Table B. Top-1 validation accuracy on 5000-class iNaturalist with ResNet-50. Linear probe performance on representations are reported.**
>
> | Method | Fine-grained | 50% fine, 50% coarse | Coarse-grained |
> | :---  | :---: | :---: | :---: |
> | InfoNCE | 46.86 | 29.38 | 8.30 |
> | AdaSSL-V | **47.40** | **36.43** | **21.09** |
>
> **Table C. Top-5 validation accuracy on 5000-class iNaturalist with ResNet-50. Linear probe performance on representations are reported.**
>
> Method | Fine-grained | 50% fine, 50% coarse | Coarse-grained |
> | :---  | :---: | :---: | :---: |
> | InfoNCE | **72.58** | 54.24 | 20.45 |
> | AdaSSL-V | 72.00 | **62.08** | **40.81** |

---

> > ### Author Response · Authors · 2025-11-21
> > **References**
> >
> > # Refs.
> >
> >
> > [Sch+21] Towards causal representation learning. Schölkopf et al., Proceedings of the IEEE, 2021.\
> > [Ben+13] Representation Learning: A Review and New Perspectives. Bengio et al., TPAMI 2013.\
> > [Che+20] A simple framework for contrastive learning of visual representations. Chen et al., ICML 2020.\
> > [Gri+20] Bootstrap your own latent: A new approach to self-supervised Learning. Grill et al., NeurIPS 2020.\
> > [Zim+21] Contrastive Learning Inverts the Data Generating Process. Zimmermann et al., ICML 2021.\
> > [Xu+24] Demystifying CLIP Data. Xu et al., ICLR 2024.\
> > [Wan+24] Text Embeddings by Weakly-Supervised Contrastive Pre-training. Wang et al., 2024.\
> > [Rus+25] InfoNCE: Identifying the Gap Between Theory and Practice. Rusak et al., AISTATS 2025.\
> > [Beh+24] LLM2Vec: Large Language Models Are Secretly Powerful Text Encoders. BehnamGhader et al., COLM 2024.\
> > [GM+21] iNat Challenge 2021 - FGVC8. Grant Van Horn and Mac Aodha. https://kaggle.com/competitions/inaturalist-2021, 2021. Kaggle.

---

### Author Response · Authors · 2025-12-02
**Summary of rebuttal**

Dear AC and SAC,

Thank you for your effort in assessing our work amidst the current turmoil.

We would like to express our gratitude to all reviewers for their positive feedback on our motivation and method, as well as their constructive comments. Below, we summarize how we addressed each reviewer's comments. Before reverting to the original scores, **Reviewer NVc3** had already engaged in the discussion and increased their score.

- **Reviewer TJcB (Score 6).** The reviewer's main feedback was on our "occasional lack of clarity". In our rebuttal, we answered each of the reviewer's points and updated the manuscript accordingly. Although the reviewer did not reply, they mentioned that the paper would "improve greatly" if these points were addressed. We also clarified our choice of SSL objectives in the general response.

- **Reviewer gRec (Score 4).** **(A)** The reviewer suggested removing some of the formalism in our work, as well as our studies on InfoNCE and AnInfoNCE. We clarified that our problem setup follows seminal works at the intersection of CRL and SSL and directly extends prior analyses on the identifiability of InfoNCE and AnInfoNCE. We also added intuitive explanations to address the raised points, improving readability.
**(B)** The reviewer further recommended removing controlled experiments and the linear-probe evaluation and focusing on applications. We explained that our method addresses a fundamental SSL problem beyond videos and therefore requires controlled analyses across different data types. We further note that linear-probe evaluation is standard in SSL. Following the reviewer’s suggestions, we also added analyses of AdaSSL’s video world-modeling capabilities, demonstrated its scalability on a larger dataset (iNaturalist-1M) and a more challenging task (5,000-way classification), and added a section to the paper discussing potential applications. The reviewer has not yet replied back.

- **Reviewer NVc3 (Score 4 → 6).** The reviewer requested several clarifications. We explained the distinction between the data-generating factors $z$ and the latent "cause" variable $r$, the role of Proposition 2.1, and why our method also applies to non-contrastive SSL (supported by new BYOL experiments). We also clarified why temporal crops qualify as natural transformations, as they modify the underlying causal entities. The reviewer raised their score to a 6 following these clarifications, as documented in our discussion thread.

- **Reviewer y8NR (Score 8).** The reviewer asked for intuition on how modeling the conditional distribution between positive pairs aids representation learning. We clarified that a flexible conditional distribution "reduces the pressure" to discard features that vary across pairs, enabling the model to retain more diverse information.
The reviewer also raised some technical questions. We explained that H-InfoNCE assumes a unimodal distribution, whereas AdaSSL can handle more complex (e.g., multimodal) ones. We added a new figure (Fig. 1) to illustrate this. We further clarified the complexity required of our editing function, our hypothesis for why AdaSSL learns disentangled representations, and the rationale for using an additional view under augmentations (supported by new ablations).


We believe our rebuttal has answered all reviewers' concerns and position AdaSSL as a novel and general method for self-supervised learning under uncertainty in the positive pairs. We hope the AC considers our response, new experiments, and the paper's improved clarity in their final assessment.


Best,\
Authors

---

### Meta-Review · Area_Chair_4QjN · 2026-01-08

**Summary:**

I continue to believe that this is a strong paper with significant merit. However, I have carefully considered the feedback from the other reviewers, particularly regarding the paper's accessibility. While I personally found the core concepts understandable, the consensus among **Reviewers TJcB**, **NVc3**, and **y8NR** indicates that the manuscript presents hurdles for some readers.

Specifically, **Reviewer TJcB** noted occasional lacks of clarity, **Reviewer NVc3** found the presentation difficult to follow, and **Reviewer y8NR** identified specific sections as hard to understand. I appreciate the authors' efforts to clarify these concepts; however, the fact that three independent reviewers struggled suggests that the current version would benefit significantly from structural improvements to ensure easier reading and broader accessibility.

Regarding the context and evaluation, **Reviewer gRec** raised valid points concerning the connection between this work and Joint-Embedding Predictive Architectures (JEPAs). Although the paper cites relevant works, I agree that a more explicit discussion of these relationships—and including other missing literature—would better contextualize the paper's originality. Furthermore, the concern regarding insufficient experiments is well-taken; additional empirical validation would strengthen the paper's contributions.

Finally, regarding the claims made in the paper: while I feel **Reviewer gRec’s** assertion that the paper is "overclaiming" might be slightly overstated itself, I do find their advice on improving the writing valuable. Refining the text to be more precise would help avoid potential misinterpretations and satisfy the demand for rigor without diminishing the paper's actual value.

Furthermore, I believe a nuanced interpretation of Reviewer y8NR’s evaluation is necessary. Although Reviewer y8NR assigned the highest possible rating, her/his written comments explicitly acknowledge areas that require improvement. There is a perceptible gap between the perfection implied by the numerical score and the constructive criticism offered in the text.

**Reviewer Concerns:**

In my view, the authors have addressed some of the reviewers’ concerns; however, a number of important issues remain unresolved, especially those related to the clarity and readability of the manuscript.

**Reviewer Scores:**

Based on a fair and consistent review process, I do not anticipate major changes in most reviewers’ ratings; however, outcomes may be less predictable under different circumstances.

---

### Decision · Program_Chairs · 2026-01-26

Accept (Poster)